# Steroids Bearing Heteroatom as Potential Drugs for Medicine

**DOI:** 10.3390/biomedicines11102698

**Published:** 2023-10-03

**Authors:** Valery M. Dembitsky

**Affiliations:** Centre for Applied Research, Innovation and Entrepreneurship, Lethbridge College, 3000 College Drive South, Lethbridge, AB T1K 1L6, Canada; valery.dembitsky@lethbridgecollege.ca or dvmioch@gmail.com

**Keywords:** heteroatom steroids, metallocene steroid conjugates, antineoplastic, anti-inflammatory, antifungal, antibacterial, antiviral

## Abstract

Heteroatom steroids, a diverse class of organic compounds, have attracted significant attention in the field of medicinal chemistry and drug discovery. The biological profiles of heteroatom steroids are of considerable interest to chemists, biologists, pharmacologists, and the pharmaceutical industry. These compounds have shown promise as potential therapeutic agents in the treatment of various diseases, such as cancer, infectious diseases, cardiovascular disorders, and neurodegenerative conditions. Moreover, the incorporation of heteroatoms has led to the development of targeted drug delivery systems, prodrugs, and other innovative pharmaceutical approaches. Heteroatom steroids represent a fascinating area of research, bridging the fields of organic chemistry, medicinal chemistry, and pharmacology. The exploration of their chemical diversity and biological activities holds promise for the discovery of novel drug candidates and the development of more effective and targeted treatments.

## 1. Introduction

Natural and synthetic steroids are complex lipophilic molecules that play various specialized and regulatory roles in multicellular organisms [1,2,3]. Synthetic steroids and related lipophilic compounds can contain diverse heteroatoms and functional groups, serving as tissue-specific regulators of gene transcription within specific domains. Consequently, they exert a broad range of physiological effects [1,2,3,4,5,6]. These steroids possess unique structural features where one or more heteroatoms, such as oxygen (O), nitrogen (N), boron (B), sulfur (S), aluminum (Al), arsenic (As), astatine (At), germanium (Ge), silicon (Si), selenium (Se), tin (Sn), tellurium (Te), and halogens (F, Cl, Br, and I), are incorporated into the steroid framework. This incorporation of heteroatoms imparts distinct chemical and biological properties to these compounds, enabling a wide range of potential therapeutic applications. Heteroatom steroids refer to steroid molecules that contain one or more heteroatoms within their structure. These steroids exhibit a wide range of biological activities, including the following: antineoplastic, anti-inflammatory, antifungal, antibacterial, antimicrobial, antiviral, immunomodulatory, and other properties. It is important to note that the specific biological activities and properties of heteroatom steroids can vary depending on the structure of the steroid, the presence and position of heteroatoms, and other factors [5,6,7,8,9]. The exploration of heteroatom steroids has led to the development of innovative synthetic strategies and the discovery of new compounds with enhanced therapeutic profiles. Their ability to target specific biological pathways and interact with molecular targets opens up new possibilities for drug design and delivery systems. Furthermore, the investigation of heteroatom steroids has provided valuable insights into the relationship between chemical structure and biological activity. This knowledge contributes to the understanding of drug-receptor interactions and the optimization of drug efficacy and safety.

Extensive research conducted over the past five decades on enzyme kinetics and the interplay between steroid precursors and their metabolic products has fostered the prevailing notion that numerous enzymes participate in the conversion of synthetic steroids and their derivatives into active steroid hormones [7,8,9,10,11]. Since the mid-1950s, thousands of distinct steroid compounds have been synthesized, many of which have found extensive application in clinical medicine and pharmacology [12,13,14,15,16,17,18,19].

This review centers on the pharmacological characteristics of selected semi-synthetic and synthetic steroid hormones and related lipophilic compounds, which exhibit distinct biological activities. These activities have been documented using experimental in vitro or in vivo *studies*, as well as computer-based evaluations. Quantitative Structure-Activity Relationship (QSAR) analysis is a computational modeling technique employed to elucidate the correlation between the structural properties of chemical compounds and their biological activity [20,21,22]. While QSAR modeling is essential for determining the biological activity of drugs, it does possess certain limitations [23,24,25,26]. Nevertheless, the information derived from QSAR analysis provides approximate reliability ranging from 70 to 97 percent, as supported using numerous publications in the scientific literature [27,28,29,30].

## 2. Steroids Bearing Nitrile Group

Nitrile-containing secondary metabolites are commonly discovered and isolated from various sources, such as bacteria, algae, sponges, plants, fungi, and insects [31,32,33,34]. 

While animals do not naturally produce nitrile-containing metabolites [35], one prevalent class of such metabolites includes cyanogenic glycosides, cyanolipids, and aromatic nitriles, which feature the -C≡N group [36,37,38,39,40]. Cyanolipids, initially isolated from plants of the soapberry family Sapindaceae (or Soapberry) in 1920, are widely distributed in nature, yet steroids with a nitrile group are not commonly found [38].

Steroids that incorporate a nitrile group are referred to as cyanosteroids [41]. The synthesis of anabolic cyanosteroids was first accomplished during the 1940s–1950s [42,43,44]. Currently, over 400 cyanosteroids have been synthesized, demonstrating notable antibacterial, antifungal, and anticancer properties [45,46,47,48,49]. 

### The Influence of Nitrile Group(s) Positioning in Steroids and Their Biological Activity

One of the most renowned and extensively utilized semi-synthetic progestogens in medicine is 17α-cyanomethylestra-4,9-dien-17β-ol-3-one (**1**), commonly known as dienogest. Its molecular structure is depicted in Figure 1, while Table 1 illustrates its biological activity, and a 3D diagram in Figure 2 illustrates the distribution of its biological activities. Dienogest, a cyanosteroid with exceptional pharmacological properties, was first synthesized in 1979 under the guidance of Professor Kurt Ponsold in Jena, Deutsche Demokratische Republik [50]. In combination with ethinylestradiol and marketed as Valette, dienogest is primarily used as a contraceptive and for the treatment of severe menstrual bleeding in the United States, European countries, and Russia [51,52,53,54].

Among dienogest derivatives, 11β-hydroxydienogest (**2**), 11β- hydroperoxy-17-hydroxy-3-oxo-19-norpregna-4,9-diene-21-nitrile (**3**), and dienogest impurity C (**4**) are notable examples. All these compounds possess antiandrogenic activity and are utilized in combination with other steroids to inhibit ovulation [52,53,54,55,56]. The compounds 1α-cyano-5α-cholestan-3-one (**5**) and (**6**) have been synthesized, and their rotatory dispersion was studied, but their biological activity remains unexplored [57]. Another synthetic compound, 2-cyano-3,12-dioxoolean-1,9-dien-28-oic acid (**7**) or CDDO, exhibits a wide range of potential actions. This cyano-hormone demonstrates inhibitory effects on the proliferation of various human tumor cell lines and displays neuroprotective, proliferative, and anti-inflammatory activities [58].

An additional compound, 2α-cyano-4,4′,17α-trimethylandrost-5-en-17β-ol-3-one (**8**), known as cyanoketone, acts as an inhibitor of the production of gonadal and adrenal steroids, including androgens, estrogens, and progesterone [59,60,61]. Other 2-cyano-3-oxo steroids (**9**, **10**, and **11**) have shown endocrinological and pharmacological activities, such as adrenal and pituitary inhibition, electrolyte modification, hypotensive effects, and coronary dilation [62,63].

Several cyanosteroids have been synthesized, including 3β-cyanosteroids such as 3-cyano-5-methoxy-5α-cholest-3-en-6-one (**12**), 3,5-dicyano-5α-cholest-3-en-6-one (**13**), as well as 5β-cyanosteroids like 5-cyano-5α-cholest-3-en-6-one (**14**) and (**15**). However, their biological activity remains unexplored [64].

Among the synthetic compounds, 5-cyano progesterone (**16**) shows potential as an inhibitor of the enzyme 5-α-reductase both in vitro and in vivo [65]. Furthermore, 6-cyano-steroids such as 6α-cyano-16α-hydroxy-cortisone (**17**), 6α-cyano-16α-hydroxy-prednisolone (**18**), and (**19**) have been synthesized as prospective agents with remarkable anti-endocrine activity [66].

Figure 3 depicts a 3D graph illustrating the predicted and calculated activity of cyanosteroids (**1**, **2**, **4**, and **20**). These steroids possess properties suitable for the treatment of menstrual irregularities and menopausal disorders and act as potent ovulation inhibitors and contraceptive agents. Similarly, Figure 4 showcases a 3D graph representing the predicted and calculated anti-inflammatory activity of cyanosteroids (**17**, **18**, and **19**). These steroids exhibit promising anti-inflammatory properties. Furthermore, Figure 5 displays a 3D graph demonstrating the activity of cyanosteroid (**23**) associated with the treatment of heart failure and related diseases.

An intriguing series of cyanosteroids, characterized by the presence of a cyano group at the α- or β- configuration in position seven of the B ring of the steroid molecules (**20**–**23**), has drawn attention. These synthesized steroids exhibit potential hormone-like properties, displaying activities such as anti-estrogenic, anti-inflammatory, glycogenic, thymolytic, eosinopenic, catabolic, and anti-androgenic effects [67,68].

Over 60 years ago, a rare class of 9-cyano steroids (**24**, structure see in Figure 6) was synthesized by Albert Ercoli and Rinaldo Gardi from Italy. However, the biological activity of these compounds remains unexplored [69]. Another intriguing group is the 17β-cyano estrogens (**25**), which represent a unique set of hormones that may serve as potential inhibitors of 16α-hydroxylase [70].

Alphaxalone, a neuroactive steroid with a 17β-acetyl group, is known for its potent anesthetic activity in humans. The 17β-carbonitrile analog of alphaxalone (**26**) was synthesized as a promising candidate with potent anesthetic and receptor actions [71]. The 16-cyano steroids (**27**–**29**) constitute a group of biologically active hormones exhibiting anti-atherosclerotic activity [72]. Likewise, the 18-cyano steroids (**30** and **31**) from the pregnane series have demonstrated pharmacological activity, particularly as anti-atherosclerotic agents [73,74].

A noteworthy carbon-bridged cyanosteroid (**31**) was synthesized from 3β-acetoxy-17-cyano-5,14,16-androstatriene using the Diels-Alder reaction. However, the biological activity of this steroid remains uninvestigated [75]. The 20-cyano steroids (**32**, **33**, and **34**) were obtained from 17-keto androstanes via 20-keto androstanes using a Witting reaction, but their activity has not been studied [76]. Raggio and Watt described the synthesis of 3β-hydroxy-9α-cyano-cholest-7-ene (**35**) and 3β-acetoxy- 7α-hydroxy- 9α-cyano-5α-cholestane (**36**) as potential inhibitors of cholesterol biosynthesis [76]. An anabolic 9α-cyano-19-nortestosterone analog (**37**) was synthesized, although its biological activity was not assessed [77,78]. Figure 7 presents a 3D plot showcasing the activity of cyanosteroids (**32**, **35**, and **36**) as inhibitors of cholesterol synthesis, while Figure 8 displays a 3D plot illustrating the activity of cyanosteroids (**38** and **39**) as inhibitors of 5α-reductase.

The synthetic compounds 17α-aza-3-cyanosteroid with a phenyl substituent on the nitrogen (**48**) and 17-aza-17α-homo-3-cyano-androst-3,5-en-16,17α-dione (**49**) exhibited moderate inhibition of 5α-reductase type II [79]. The synthesis of 6-cyano-16-methylene-17α-hydroxypregna-4,6-diene-3,20-dione 17-acetate (**50**) and its 3-OEt analog (**51**) has been described, but their biological activity has not been tested [80]. Various authors have synthesized several biologically active cyanosteroids and their derivatives (**42**–**47**), although the extent of their activity evaluation is limited [81,82,83,84]. Additionally, carbon-bridged steroids of cyanosteroids (**48**, **49**, and **50**) and a rare dicyanosteroid (**51**) have been synthesized, yet their biological activity remains unstudied [85].

Synthetic anabolic cyanosteroids primarily exhibit cytotoxic activities, although the predicted biological activity indicates a wide spectrum of potential effects. Table 1 and Table 2 present a variety of activities, including antineoplastic, antiviral, antidiabetic, anti-ischemic, and others. These results are of significant interest to physicians, pharmacologists, and the pharmaceutical industry.

## 3. Steroids Bearing Nitro Group

To date, approximately one thousand metabolites containing the nitro group have been isolated from various natural sources. However, steroids incorporating this group within their structure have not been discovered [86,87,88,89,90,91,92]. The most well-known group of nitro-containing natural compounds is aristolochic acid and its derivatives, which are found in species of the Aristolochia family (Aristolochiaceae) [93,94,95]. It is worth noting that Streptomyces and Penicillium species produce a diverse array of antibiotics, including simple nitroaromatic metabolites, as well as more complex compounds like siderophores or cyclic heptapeptides [89,96,97,98,99,100].

The nitro group can be found in aliphatic and aromatic hydrocarbons, fatty acids, carboxylic acids, terpenoids, heterocyclic compounds, and peptides [92,93,94,95,96,97,98,99,101,102]. Beyond its charge-imparting ability, the nitro group possesses unique properties that make it a crucial functional group in chemical synthesis. Acting as an excellent electron acceptor, the nitro group is utilized in numerous organic reactions, and its ease of transformation into various functional groups adds to the significance of nitro compounds in the synthesis of complex molecules [86,87,88,89,90,95,96]. In recent years, several review articles have explored the toxicity, mutagenicity, biosynthesis, and biodegradation of nitro-containing compounds [103,104,105,106,107]. Of particular interest is the wide range of nitro-steroids, wherein the nitro group is positioned at various locations within the steroid skeleton. This variability adds to the structural diversity and potential functional properties of these compounds.

### The Influence of Nitro Group Positioning in Steroid Molecules and Their Biological Activity

Surprisingly, no naturally occurring steroids containing a nitro group have been discovered thus far. However, there are approximately 300 known synthetic nitrosteroids [86,87,88,108,109,110]. Many of these compounds exhibit noteworthy activities such as antitumor, antibacterial, antifungal, and more [111].

The selected nitrosteroids can be categorized into six groups based on the position of the nitro group within the steroid structure. The first group comprises steroids with a nitro group in the second position (**52**–**61**), as depicted in Figure 9, with their biological activity detailed in Table 3. The second group consists of anabolic steroids featuring a nitro group in the third position (**62**–**71**). The third group encompasses steroids, with a nitro group in the fourth position (**72**–**81**). The fourth group includes steroids, where the nitro group is in the sixth position (**82**–**90**), as shown in Figure 10, with their biological activity presented in Table 4. The fifth group consists of steroids, with a nitro group in the seventh position (**91**–**93**). Lastly, the sixth group encompasses steroids with a nitro group located at positions **11**, **16**, **17**, **20**, or **21** within the steroid structure (**94**–**100**).

For steroids (**52**–**54**, structures see in Figure 9 and Figure 10, and 3D graph shown in Figure 11), their activity has not been determined. However, the 2-nitro-3-oxo steroid (**55**) has demonstrated significant antifungal activity, inhibiting the growth of communicable pathogenic fungi such as *Trichophyton mentogrophytes* and *Microsporum gypseuin* [112]. 2-Nitro-3-oxo steroids are widely utilized as fungicides for the treatment of dermatophytosis, a superficial fungal skin disease. Several compounds, including 2-nitro-cholestan-3-one, 2-nitro-cholestan-3β-ol, 2-nitro-cholestene, and 2-nitro-cholestane, have been synthesized [113]. Synthetic 2-nitrosteroids (**58**–**64**) have shown antitumor activity [111]. Additionally, 2-nitro-steroids (**61**) exhibited a wide range of biological activities [114]. Other 2-nitrosteroids have also been synthesized [111,115,116,117]. The dominant activities associated with 2-nitrosteroids include antineoplastic, immunosuppressant, bone disease inhibition, analeptic, dermatologic, and various other properties (refer to Table 3).

It is worth noting that 2-nitrosteroids exhibit various unique properties in addition to their fundamental characteristics. Some compounds in this class demonstrate anesthetic, analeptic, and antipruritic properties. Furthermore, certain steroids can be employed as dermatologic agents for the treatment of skin, nail, or hair diseases. Contraceptive properties are also exhibited by specific 2-nitrosteroids (refer to Table 3).

A series of unsaturated steroids (**62**–**66**, 3D graph shown in Figure 12) featuring a nitro group at the 3 position were synthesized and demonstrated in vitro as inhibitors of human and rat prostatic steroid 5α-reductase [111,118,119]. Another series comprised 5α-steroids with nitro groups at positions 3β, 3α, 4β, 4α, 6β, 6α, 7β, 7α, and 17β, as well as 4β and 4α-nitro-5β-cholestane, and related compounds. Their properties were studied and reported [111,116,120,121]. Dominant activities associated with 3-nitrosteroids include analeptic, neuroprotective, antineoplastic, bone disease treatment properties, and other biological activities presented in Table 4. Additionally, 3-nitrosteroids possess specific properties. For example, some steroids can serve as preventive agents for prostate disorders, including benign prostatic hyperplasia, as well as for the treatment of bone diseases (refer to Table 4).

The 4-substituted estrones, as well as a series of 6α- and 6β-substituted estrones, have been identified as aromatase inhibitors [122]. Several 4-nitrosteroids (**72**–**81**), including compounds like 20,21-dihydroxy-4-nitropregn-4-en-3-one (**78**) and 17β-cyclopropyloxy-4-nitroandrost-4-en-3-one (**80**), exhibit aromatase inhibitory activity, as well as inhibition of steroid C17-20 lyase and 5α-reductase [123]. Three 4-nitrosteroids (**78**–**80**, 3D graph is shown in Figure 13) were synthesized and identified as potential inhibitors of 4-methyl sterol oxidase [124]. The primary activities associated with 4-nitrosteroids include anti-eczematic, anti-hypercholesterolemic, antifungal, antineoplastic, and anti-osteoporotic properties. Other activities of 4-nitrosteroids are presented in Table 3. Notably, 4-nitrosteroids possess special properties such as anti-hypercholesterolemic and anti-psoriatic effects (refer to Table 3).

Ringold and colleagues [125] synthesized a series of 6-nitrosteroids and demonstrated that many of these compounds exhibited anti-inflammatory activity. For 6-nitrosteroids (**82**–**84**), characteristic activities include antineoplastic, neuroprotective, ovulation inhibition, prostate disorder treatment, muscular dystrophy treatment, and other activities, as shown in Table 4. The activity of 6-nitrosteroids (**82**–**93**, 3D graph for **89** is shown in Figure 14) presented in Table 4 has not been previously studied. Additionally, 6-nitrosteroids possess unique properties such as respiratory analeptic effects, muscular dystrophy treatment, ovulation inhibition, and hypogonadism treatment (refer to Table 4).

In the case of 7-nitrosteroids (**91**–**93**), it has been demonstrated that they possess inhibitory effects on gonadotropin and ovulation [126]. The dominant activities associated with 7-nitrosteroids, as shown in Table 5, include anti-neoplastic properties, ovulation inhibition, anti-seborrheic effects, cardiovascular analeptic activity, antipruritic effects, and other activities.

Various nitrosteroids have been synthesized, and their activities have been partially studied [111,126,127,128,129,130,131,132,133,134,135,136,137]. For instance, 11-nitrosteroids exhibit antineoplastic and anesthetic activity, while 16-nitrosteroids display anti-inflammatory and anesthetic activities. 17-nitrosteroids are characterized by anti-inflammatory and anti-secretory activities, 20-nitrosteroids demonstrate respiratory analeptic effects and activities related to prostate disorders, and 21-nitrosteroids exhibit anti-allergic properties and ovulation inhibition. The activities of nitrosteroids (**94**–**100**) are presented in Table 4.

**Figure 13 biomedicines-11-02698-f013:**
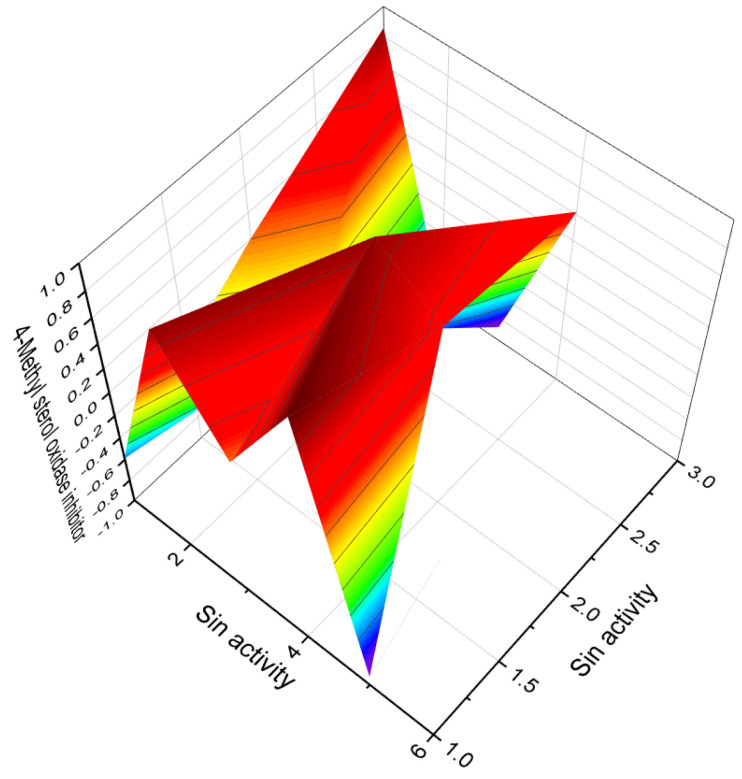
3D Graph illustrating the predicted and calculated activity of nitro-steroids (**78**, **79**, and **80**) with over 92% confidence as inhibitors of 4-methyl sterol oxidase. 4-Methylsterols are intermediate compounds in sterol biosynthesis and serve as precursors for the synthesis of cholesterol, ergosterol, and phytosterols. The enzyme sterol-C4-methyl-oxidase is responsible for catalyzing the demethylation of C4-methylsterols within the cholesterol synthesis pathway. C4-methylsterols are involved in the activation of meiosis and are found in high concentrations in the testes and ovaries, playing a crucial role in the process of meiosis activation. Nitrosteroids serve as inhibitors of this process, as indicated by previous studies [138,139,140].

**Figure 14 biomedicines-11-02698-f014:**
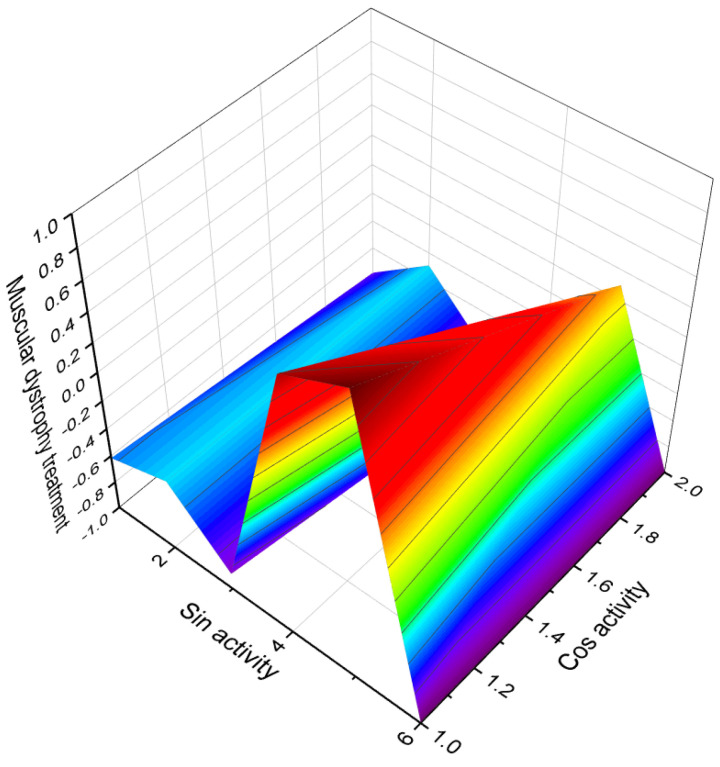
3D Graph depicting the predicted and calculated activity of nitro-steroid (**89**) with over 94% confidence as a potential treatment for muscular dystrophy. Muscular dystrophies encompass a group of hereditary genetic disorders characterized by progressive muscle weakness. The identification of nitro-steroid (**89**) as a candidate for treating muscular dystrophy is of significant practical interest to pharmacologists, as it raises the possibility of utilizing steroids for the treatment of this condition. Further research and investigations are necessary to explore the therapeutic potential of steroids in addressing muscular dystrophy.

Nitrosteroids are lipid compounds with high biological activity, and they belong to the class of steroid hormones that play a crucial role in regulating metabolism and various physiological functions in the human body. Synthetic hormones, including nitrosteroids, have demonstrated superior action compared to their natural counterparts. These compounds exhibit potent anti-cancer and antibacterial properties and possess unique characteristics that are distinct from nitrosteroids alone. The data presented on nitrosteroids are of significant interest to academic scientists and pharmaceutical companies involved in the production of anabolic steroids. Each group of nitrosteroids exhibits distinct features that set them apart from other steroid groups. The comprehensive information on the biological activities of 2-, 3-, 4-, 6-, 7-, 11-, 16-, 17-, 20-, and 21-nitrosteroids presented in this study is expected to stimulate researchers in the field of medical chemistry to focus on the synthesis and exploration of specific nitro steroid groups.

## 4. Steroids Bearing Halogen Atom(s) (F, Cl, Br, or I)

Halogenated steroids form a distinct subset of natural lipid molecules and can be found in various sources such as plants, fungi, marine invertebrates, and seaweeds. Initially, halogenated natural products were considered rare and potentially toxic compounds. However, their number has significantly increased over time and currently exceeds 6000 identified compounds [141,142,143,144,145,146,147,148,149,150,151,152,153,154,155,156,157,158,159,160]. In plants, chlorine-containing steroids are prevalent, while marine organisms contain steroids with chlorine, bromine, and iodine atoms. Interestingly, no naturally occurring fluorine-containing triterpenoids have been discovered [141,142,146,151,160].

The synthesis of halogenated compounds, particularly steroids, is of both academic and pharmaceutical interest [161,162,163]. Synthetic anabolic hormones, increasingly used in sports medicine, have gained substantial commercial significance [164,165].

The halogenated steroids discussed in this review are categorized into four groups. The first group comprises steroids incorporating a fluorine atom within their molecular structure. The structures of fluorine-containing steroids are depicted in Figure 1, and their biological activity is outlined in Table 1. The second group includes steroids containing a chlorine atom. The structures of chlorine-containing sterols are shown in Figure 2, alongside their respective biological activities presented in Table 2. The third group focuses on bromine-containing steroids, as illustrated in Figure 3, with corresponding biological activity details provided in Table 3. Lastly, the fourth group explores iodine-containing steroids, with their structures and biological activities presented in Figure 4 and Table 4, respectively.

### 4.1. Steroids Bearing Fluorine Atom(s)

Steroids bearing a fluorine atom represent a small group of natural compounds primarily found in plants [160,166,167]. It is noteworthy that currently, all fluoride-containing steroids are of synthetic origin [168,169]. Many of these lipid compounds belong to the class of anabolic steroids and are commonly used in sports medicine and bodybuilding [170,171]. 

One well-known fluorinated steroid is halotestin (**101**), which is a 17α-alkylated derivative of testosterone. Halotestin, shown in Figure 15, belonged to the class of active anabolic-androgenic hormones and was synthesized in the late 1950s [172,173]. This steroid is a potent inhibitor of 11β-hydroxysteroid dehydrogenase type 2 and has demonstrated anticancer activity against breast cancer [174,175].

Another series of interest is the 2-fluoro-5α-androstane-3,17-diones, including compound (**102**, structure see in Figure 15), which was synthesized by Allinger and colleagues [176]. The configuration and conformation of the synthesized steroids were determined; however, their biological activity was not investigated. Additionally, other 2-fluoro androstanes have been synthesized as inhibitors of gonadotropin in pregnancy and for the treatment of menstrual disorders [177].

Various steroids bearing a fluorine atom have been synthesized for research purposes, although their activity has not been extensively studied [169,178,179]. For example, 3β-fluoro-5-pregnen-20-one (**103**) and other compounds such as (**104**–**110** and **112**) were synthesized to develop efficient synthesis methods, but their biological activity remains unknown [169,178,179].

One specific steroid, 17-fluoro-progesterone (**107**), was synthesized as a hormone for the treatment of menopause or perimenopause without complications in the uterus of warm-blooded animals [180]. However, the activity of this compound has not been reported extensively.

Additionally, a rare 2,2,4-trifluoro-steroid (**111**) was synthesized over forty years ago, but its activity has not been investigated [181]. Furthermore, several fluorinated cholesterol derivatives, including fluorine-containing steroids (**113**–**116**), have been described by various researchers, but the activity of many of these compounds has not been published [182,183,184].

By examining the data presented in Table 5, we can observe that fluorinated steroids primarily belong to the class of anabolic steroids. These compounds demonstrate a range of biological activities, including anti-inflammatory, anti-allergic, anti-asthmatic, and gynecological disorder treatments, as well as potential antineoplastic properties (structures given in Figure 15).

**Table 5 biomedicines-11-02698-t005:** Biological activities of steroids bearing fluorine atom(s) (**101**–**116**) [175].

No.	Dominated Biological Activity (Pa) *	Additional Predicted Activities (Pa) *
**101**	Anabolic (0.983)	Anti-inflammatory (0.967)
	Antiallergic (0.959)	Inflammatory Bowel disease treatment (0.779)
	Anti-asthmatic (0.946)	Respiratory analeptic (0.776)
**102**	Antineoplastic (0.961)	Ovulation inhibitor (0.678)
	Gonadotropin inhibitor (0.889)	Endometriosis treatment (0.668)
	Prostate disorders treatment (0.746)	Contraceptive (0.617)
	Prostate cancer treatment (0.717)	Gynecological disorders treatment (0.597)
**103**	Prostate disorders treatment (0.780)	Anti-infertility, female (0.750)
	Prostatic (benign) hyperplasia treatment (0.692)	Menopausal disorders treatment (0.631)
**104**	Antiallergic (0.849)	Erythropoiesis stimulant (0.816)
	Anesthetic general (0.842)	Prostate disorders treatment (0.780)
	Anti-asthmatic (0.738)	Choleretic (0.682)
**105**	Anti-inflammatory (0.988)	Antipruritic (0.902)
	Antiallergic (0.980)	Autoimmune disorders treatment (0.887)
	Anti-asthmatic (0.962)	Immunosuppressant (0.839)
**106**	Anesthetic general (0.900)	Antineoplastic (0.805)
	Cardiotonic (0.746)	Contraceptive (0.765)
**107**	Prostate disorders treatment (0.992)	Antineoplastic (0.916)
	Gynecological disorders treatment (0.962)	Prostatic (benign) hyperplasia treatment (0.899)
	Menopausal disorders treatment (0.949)	Anesthetic general (0.824)
**108**	Prostate disorders treatment (0.992)	Antiacne (0.932)
	Gynecological disorders treatment (0.951)	Antineoplastic (0.930)
	Menopausal disorders treatment (0.876)	Prostatic (benign) hyperplasia treatment (0.873)
**109**	Respiratory analeptic (0.950)	Antineoplastic (0.905)
	Anti-inflammatory (0.922)	Anti-secretoric (0.875)
**110**	Anti-inflammatory (0.925)	Antineoplastic (0.820)
	Antiallergic (0.902)	Prostate disorders treatment (0.720)
	Anti-asthmatic (0.823)	Diuretic (0.656)
**111**	Antineoplastic (0.914)	Ovulation inhibitor (0.663)
	Prostate disorders treatment (0.710)	Menopausal disorders treatment (0.625)
**112**	Antiallergic (0.961)	Antiarthritic (0.915)
	Anti-asthmatic (0.955)	Male reproductive dysfunction treatment (0.903)
	Anti-inflammatory (0.937)	Inflammatory Bowel disease treatment (0.824)
**113**	Antineoplastic (0.888)	Anti-hypercholesterolemic (0.840)
	Anesthetic general (0.865)	Hypolipemic (0.653)
**114**	Dermatologic (0.917)	Antiallergic (0.902)
	Anti-inflammatory (0.916)	Anti-asthmatic (0.883)
	Anti-secretoric (0.858)	Antiarthritic (0.880)
**115**	Anesthetic general (0.865)	Antineoplastic (0.865)
	Respiratory analeptic (0.821)	Prostatic (benign) hyperplasia treatment (0.825)
**116**	Anesthetic general (0.928)	Antineoplastic (0.903)
	Respiratory analeptic (0.862)	Prostatic (benign) hyperplasia treatment (0.871)

* Only activities with Pa > 0.5 are shown.

The 3D plots shown in Figure 16 and Figure 17 depict the predicted and calculated activities of specific fluorinated steroids, namely (**101**, **105**, **110**, and **112**) and (**107** and **108**), respectively. These plots visually represent the multiple beneficial activities exhibited by these compounds. Further details about the specific activities can be found in the accompanying text or in the notes provided beneath the respective figures.

### 4.2. Steroids Bearing Chlorine Atom(s)

Steroids bearing chlorine atom(s) are a well-known group of natural compounds predominantly found in plants and marine invertebrates [142,146,152,156,157]. Extensive reviews have been published recently, providing further details on these chlorinated lipid metabolites [156,157,158,185,186]. 

In the realm of synthetic chlorinated steroids, several anabolic hormones have gained popularity among athletes. These include 4-chlorotestosterone, also known as clostebol (**117**, structure see in Figure 18, and activity see in Table 6), norclostebol (**118**), oral turinabol (**119**), and oxyguno (**120**) [187,188,189,190,191,192]. Clostebol possesses anabolic effects and has been used in sports to enhance physical performance, although it is prohibited by the International Olympic Committee [187]. Norclostebol, also known as lentabol (**118**), is a derivative of testosterone and exhibits 6.6 times higher anabolic potency and 40% higher androgenic activity compared to testosterone. It is a potent anabolic compound with minimal side effects [188]. Oral turinabol (**119**), or 4-chlorodehydromethyl-testosterone, is a powerful anabolic steroid developed in East Germany in 1962. It has maintained a high safety rating over many decades, suitable for use by both men and women, including children, as it promotes weight gain with minimal side effects [189,190]. Oxyguno (**120**), or 4-chloro-17α-methyl-etioallochol-4-ene-17β-ol-3,11-dione, is an anabolic steroid with approximately 7% of the androgenic effect of testosterone and at least 850% of the anabolic effect of testosterone. It was synthesized in East Germany in the 1960s and was utilized by athletes from the German Democratic Republic (GDR) [191,192].

Among the chlorinated steroids, there are several noteworthy compounds. Hexadrone (**121**), also known as 6-chloro-androst-4-ene-3-one-17β-ol, is a new generation legal androgen that promotes muscle mass increase (up to 8–12 pounds) and enhances athlete strength. It is a potent prohormone with strong anabolic and androgenic properties, exhibiting a ratio of 300:1. Notably, it does not cause water retention in the body [193]. 6β-Chloro-androst-4-en-17β-ol-3-one (**122**) is an epimer of hexadrone, possessing milder properties compared to hexadrone [193].

Promagnon (**123**), or 3-chloro-17-methylandrostenediol, is a methylated derivative of the anabolic steroid testosterone. It is designed to stimulate muscle growth and enhance muscle density, vascularity, and hardness [194]. The biological activities of these anabolic chlorinated hormones (**117**–**123**) are provided in Table 2.

Additionally, the synthesis of chlorinated steroids has led to the discovery of compounds with specific properties. For example, a series of 17-chloro steroids were synthesized as potential anti-influenza drugs, and one of the compounds (**124**) exhibited pronounced anti-inflammatory activity [195]. A 10-chloro-19-norcardiosteroid (**125**) was synthesized, but its activity has not been determined [196]. Various chloro- and bromo-estrogens, including 2,4-dichloro-estriol (**126**), were synthesized and demonstrated estrogenic activity [197]. Several chlorinated steroids (**127**–**130**) were synthesized by different researchers, although their activity has not been determined [198,199,200,201]. Figure 19 presents a 3D plot illustrating the anti-inflammatory activity of chlorinated steroids (**127**, **129**, and **130**).

### 4.3. Steroids Bearing Bromine Atom(s)

Brominated natural compounds are primarily synthesized in marine invertebrates and algae [142,146,156]. They are also found in *Acorospora* lichen [202,203] but are not commonly found in plants. Brominated steroids are a rare class of natural lipids and are primarily found in marine sponges. For example, nakiterpiosinone and nakiterpiosin, two related C-nor-D-homosteroids, have been isolated from the sponge *Terpios hoshinota*. These compounds show potential as anticancer agents, particularly in tumors resistant to existing antimitotic agents and dependent on Hedgehog pathway responses for growth [204,205].

A series of 6-bromo compounds (**131** and **132**, as shown in Figure 20, with biological activity presented in Table 7) were synthesized over 60 years ago as therapeutic agents with progestational activity [206,207]. 2-Bromo- (**133**) and 4-bromoestrones (**134**) and their corresponding derivatives have been synthesized using the Suzuki-Miyaura cross-coupling method, although their activity has not been reported [208]. Farrar [209] reported the synthesis of 9-bromo-11β-hydroxy steroids (**135** and **136**), but their biological activity has not been studied.

Brominated triterpenoids (**137**, **138**, and **139**) were synthesized from betulinic acid, which was isolated from *Bischofia javanica*. These compounds have shown strong activity against topoisomerase IIa and demonstrated activity against HeLa cells [210]. Additionally, a series of 16-bromo steroids (**140**, **141**, and **142**) were synthesized and exhibited significant anti-inflammatory and glucocorticoid activity, as well as some arthritic, allergic, and asthmatic activities [211,212,213,214].

### 4.4. Steroids Bearing Iodine Atom(s)

Iodinated natural metabolites are exclusively synthesized by marine organisms and algae [142,146]. They are present in various lipid compounds, fatty acids, peptides, and other complex molecules [148,159,160,215,216,217]. While iodinated steroids have not been found in nature, there are over 100 known synthetic steroids containing iodine.

The biological activity of iodine-containing steroids has been studied to a lesser extent compared to their analogs containing fluorine, chlorine, or bromine. For example, Fried [218] investigated the effect of 9α-halo-(F, Cl, Br, I)-11-hydroxyprogesterones on the inhibition of glycogen biosynthesis in rat liver. The study showed that the activity of halogenated steroids increases from 9α-iodo-11-hydroxyprogesterones (0.1) to 9α-fluoro-11-hydroxyprogesterones (0.85), and adrenocortical activity increases from 0.1 (iodine), 0.3 (bromine), 4.7 (chlorine) to 10.7 (fluorine) for the same compounds.

2-Iodo-androst-4-ene-3,17-dione (**143**, structure see in Figure 21, and biological activity is shown in Table 8) and a series of similar 2-iodo-steroids have been successfully synthesized, but their biological activity has not been determined [219]. Similarly, 2,4-diiodoestrones (**144**) were synthesized using the Suzuki-Miyaura cross-coupling method, but no specific activity data have been reported [208]. In the pursuit of developing methods for synthesizing iodine-containing steroids, 21-iodo-20-keto-pregnanes (**145** and **146**) have been synthesized using various approaches, but their activity has not been investigated [220,221].

The 16-iodo steroids, including steroids **147**, **150**, and **151**, exhibit diverse biological activities. For instance, steroids **147** and **151** demonstrate significant anti-inflammatory and glucocorticoid activities [222]. On the other hand, 16α-estradiol, also known as 16α-iodo-E2 (**150**), exhibits notable estrogenic activity and is used in the study of the estrogen receptor [223]. It plays a role in regulating cell growth via estrogen signaling and shows potential as a target for thyroid cancer treatment [224]. As mentioned earlier, 9-iodo steroids inhibit glycogen biosynthesis in rat liver, although to a lesser extent compared to their fluoride-containing counterparts. This observation is also applicable to steroid **148** [218].

7α-iodo-5α-dihydrotestosterone (**149**) exhibits high androgenic activity and serves as an effective receptor-mediated diagnostic imaging agent [225]. The synthesis of the iodinated steroids series (**152**–**158**) has been documented, although their biological activity has not been determined [226,227,228,229]. Additionally, an antitumor steroid known as 19-iodocholest-5-en-3β-ol palmitate (**159**) has been synthesized from cholesterol [230].

This section presents the biological activity of halogenated steroids (F, Cl, Br, and I). A total of sixty halogenated steroids have demonstrated confirmed and predicted activities such as anti-inflammatory, estrogenic, anabolic, gynecological disorders treatment, anti-arthritic, antineoplastic, and other activities. This research is highly significant for pharmacologists, physicians, biochemists, and the agricultural industry. The 3D plot in Figure 22 visualizes the activity of iodinated steroids (**154**, **158**, and **159**).

## 5. Steroids Bearing Epithio Group

Epithio-containing steroids represent a small group of compounds that contain the epithio (sulfur) functional group attached to the steroid skeleton. These steroids have been synthesized and studied for their biological activities [231,232,233,234,235]. The synthesis and biological activities of epithio steroids are still relatively limited compared to other functional groups. Further research is needed to explore their potential therapeutic applications and understand their mechanisms of action. One example of an epithio-containing steroid is 17α-ethioandrostenone (**160**, for structure, see Figure 23, and 3D graph see in Figure 24), which is a synthetic compound with androgenic activity [231]. Another example is 2α,3α-epithio-5α-androstan-17β-ol (**161**), which has been studied for its anti-inflammatory and analgesic properties [232]. These epithio steroids have been extensively studied for their anabolic properties and their potential applications in enhancing athletic performance. They have also shown promise in the treatment of various medical conditions. However, it is important to note that the use of these compounds should be approached with caution and under the supervision of medical professionals, as their misuse or abuse can lead to adverse health effects [233,234,235].

Epithio steroids containing the thiirane group at positions 2 and 3 are classified as anabolic steroids and have gained popularity in sports medicine and pharmacology [236,237,238]. Some well-known epithio steroids used in sports pharmacology and medicine include epistane, epitiostanol (known for its potent anti-estrogenic and antitumor effects), hemapolin, mepitiostane, epivol, epivol black, and straight epi (2,3α-epithio-17α-methyletioallocholane-17β-ol) [237,238,239,240,241].

Figure 23 shows the structures of 2,3-epithio steroids (**160**–**168**). These steroids belong to the large group of anabolic steroids and have gained significant interest from pharmacologists and lipidomic networks. Notable examples of 2,3-epithio steroids include epitiostanol (2α,3α-epithio-5α-androstan-17α-ol) and epistane (17α-methyl-2α,3α-epithio-5α-androstan-17β-ol), both of which were synthesized in the 1960s. Epitiostanol was used in the treatment of breast cancer, while epistane was utilized to promote lean muscle mass and fat reduction [242,243,244,245]. These 2,3-epithio steroids exhibit various physiological activities, as indicated in Table 9. Additionally, 3α,4α-epithio-5α-androstan-17β-ol (**169**) has shown pharmacological activity, specifically pituitary gonadotrophin inhibiting activity [243].

Several 5,6-epithio steroids (**170**–**173**) have been identified to possess antiseptic, germicidal, and fungicidal properties, making them valuable for use in skin compounds to prevent occupational dermatitis and protect the skin [246]. Withanolides, steroidal lactones isolated from the Indian plant *Withania somnifera* and related species, have demonstrated antimicrobial, anticancer, antiproliferative, anti-inflammatory, and antiarthritic activities [247,248,249]. 

Thiirane withanolide derivatives (**172** and **173**) have been synthesized and reported to exhibit anticancer activity [250]. Additionally, 7α,8α-epithio-7-dehydrocholesterol (**174**) has shown respiratory analeptic, cholesterol antagonist, and anti-hypercholesterolemic activities (Table 2). Korneno et al. synthesized 11,12-epithio steroids of the pregnane series (**175**–**177**), although their specific activity was not determined initially [243]. Subsequent studies have revealed that these steroids possess anti-hypercholesterolemic activity. 16α,17α-epithio-progesterone (**178**, structure shown in Figure 25 and biological activity shown in Table 10) has been synthesized as a potential inhibitor of DOCA (deoxycorticosterone acetate) [251]. Epithio steroids (**179** and **180**) contain the epithio group at positions 22 and 23 or 24 and 25 in the hydrocarbon tails of cholesterol and lanosterol, respectively, and are expected to exhibit cholesterol antagonist properties. 

**Table 9 biomedicines-11-02698-t009:** Biological activities of steroids bearing sulfur atom (**160**–**177**) [252].

No.	Dominated Biological Activity (Pa) *	Additional Predicted Activities (Pa) *
**160**	Antineoplastic (0.964)	Anti-secretoric (0.948)
	Antineoplastic (breast cancer) (0.898)	Cytostatic (0.798)
	Estrogen antagonist (0.860)	Erythropoiesis stimulant (0.760)
**161**	Antineoplastic (0.966)	Anti-secretoric (0.952)
	Estrogen antagonist (0.832)	Cytostatic (0.781)
	Prostatic (benign) hyperplasia treatment (0.673)	Bone diseases treatment (0.663)
**162**	Antineoplastic (0.966)	Anti-secretoric (0.952)
	Estrogen antagonist (0.832)	Prostate disorders treatment (0.736)
	Anabolic (0.748)	Prostatic (benign) hyperplasia treatment (0.673)
**163**	Antineoplastic (0.932)	Anti-secretoric (0.863)
	Bone diseases treatment (0.729)	Anti-hypercholesterolemic (0.759)
	Estrogen antagonist (0.660)	Hypolipemic (0.676)
**164**	Antineoplastic (0.955)	Anti-secretoric (0.938)
	Estrogen antagonist (0.807)	Erythropoiesis stimulant (0.643)
	Anabolic (0.665)	Cytostatic (0.676)
**165**	Antineoplastic (0.971)	Anti-secretoric (0.861)
	Estrogen antagonist (0.761)	Male reproductive dysfunction treatment (0.808)
**166**	Antineoplastic (0.960)	Anti-secretoric (0.965)
	Estrogen antagonist (0.915)	Cytostatic (0.724)
	Anabolic (0.857)	Bone diseases treatment (0.716)
**167**	Estrogen antagonist (0.946)	Anti-secretoric (0.967)
	Antineoplastic (0.939)	Immunosuppressant (0.795)
	Anabolic (0.823)	Cytostatic (0.787)
**168**	Antineoplastic (0.970)	Immunosuppressant (0.729) Anti-secretoric (0.677)
	Prostate disorders treatment (0.729)	
	Estrogen antagonist (0.686)	
**169**	Cardiotonic (0.925)	Antineoplastic (0.868)
	Antiarrhythmic (0.858)	Anti-secretoric (0.854)
**170**	Cholesterol antagonist (0.933)	Hepatoprotectant (0.850)
	Anti-hypercholesterolemic (0.929)	Hepatic disorders treatment (0.761)
**171**	Anesthetic general (0.847)	Antineoplastic (0.780)
	Anti-hypercholesterolemic (0.715)	Immunosuppressant (0.751)
**172**	Hepatoprotectant (0.940)	Anti-eczematic (0.939)
	Hepatic disorders treatment (0.935)	Antifungal (0.795)
	Cytostatic (0.934)	Antineoplastic (0.874)
**173**	Antie-czematic (0.929)	Antineoplastic (0.872)
	Cytostatic (0.926)	Antineoplastic (breast cancer) (0.649)
**174**	Respiratory analeptic (0.963)	Cholesterol antagonist (0.946)
	Anesthetic general (0.913)	Anti-hypercholesterolemic (0.930)
	Analeptic (0.876)	Hypolipemic (0.781)
**175**	Cholesterol antagonist (0.932)	Anesthetic general (0.923)
	Anti-hypercholesterolemic (0.900)	Respiratory analeptic (0.919)
	Cardiotonic (0.886)	Analeptic (0.872)
	Atherosclerosis treatment (0.838)	
**176**	Cardiotonic (0.941)	Respiratory analeptic (0.959)
	Antiarrhythmic (0.844)	Analeptic (0.877)
	Erythropoiesis stimulant (0.801)	Atherosclerosis treatment (0.805)
**177**	Anesthetic general (0.897)	Cardiotonic (0.883)
	Atherosclerosis treatment (0.774)	Antiarrhythmic (0.806)

* Only activities with Pa > 0.5 are shown.

The 3D plot in Figure 26 depicts the activity of epithio steroids (**169**, **176**, and **178**). Two steroids (**181** and **182**) were synthesized as potential aromatase inhibitors, and subsequent studies confirmed their activity as inhibitors of breast cancer [252]. The study of their activity was confirmed much later than their synthesis [252], and the same activity was observed for steroids (**183** and **184**). Aromatase inhibitors are a class of drugs used to treat breast cancer in postmenopausal women and men, as well as gynecomastia in men. 

Thiiranyl steroids (**185** and **186**) have been synthesized, and their biological activities are shown in Table 3 [253]. 10-thiiranyl-4-estrene-3,17-dione (**181**) is known as an inhibitor of estrogen synthetase from human placental microsomes [254]. Thiiranyl steroid (**186**) was synthesized as an inhibitor of lanosterol 14α-demethylase (P45014DM) and showed inhibition of cholesterol biosynthesis [73]. The activities of these steroids are presented in Table 9.

Thiirane-containing steroids, which are derivatives or analogs of α- and/or β-androstanes, have been found to exhibit interesting properties and activities. In these steroids, the thiirane (epithio) group is located at position 3 or 17 of the steroid structure. Compound (**189**) is an example of a steroid with the epithio group at position 17, while compounds (**187**) and (**188**) have the epithio group at position 3. These compounds have been investigated for their biological activities and have shown promising results. The activities of thiirane-containing steroids are diverse and varied. Table 9 presents a summary of the activities associated with these compounds. 

Thiirane-containing steroids have also shown a range of other activities, which could include but are not limited to antiviral properties, antidiabetic effects, anti-ischemic (prevention of insufficient blood supply) properties, treatment of phobic disorders, and regulation of lipid metabolism. The position of the thiirane group within the steroid structure appears to play a significant role in determining the compound’s activity. 

This suggests that modifications to the core structure of steroids by incorporating the thiirane group can lead to significant changes in their biological properties. Thiirane steroids have demonstrated potential as antineoplastic agents, meaning they can inhibit the growth of tumors or cancer cells. This makes them of particular interest in cancer treatment. Thiirane steroids have shown activity in the field of dermatology, indicating potential for the treatment of skin-related conditions or disorders. Some thiirane steroids have exhibited effects relevant to gynecological disorders, suggesting possible applications in women’s health. The broad spectrum of biological activities exhibited by synthetic thiirane steroids makes them of great interest to chemists, physicians, pharmacologists, and the pharmaceutical industry. Further research and development in this area may uncover new therapeutic applications and potential drug candidates.

## 6. Steroids Bearing Boron Atom(s)

Organoboron chemistry is a specialized field within both organic and inorganic chemistry that focuses on the study of chemical compounds containing boron (**B**) and carbon atoms, denoted as C-B or C-O-B bonds. With over 100,000 known compounds, organic boron compounds exhibit a wide range of biological activities [255,256,257,258,259,260,261,262,263,264]. The field of boron medical chemistry, a branch of organic chemistry, specifically investigates the biological properties of organoboranes [256,257,261]. Organoboron compounds demonstrate diverse biological activities, making them intriguing targets for research and development. They have shown promising potential as enzyme inhibitors, displaying antibacterial, anticancer, and other activities [256,257,258]. Their unique chemical properties, including their ability to form stable covalent bonds with target molecules, contribute to their effectiveness as pharmaceutical agents.

Boronosteroids (**190**–**201**), as depicted in Figure 27, have been synthesized in various laboratories and exhibit a wide range of activities. The biological activity of these steroids is summarized in Table 11. 

They possess distinct features, including antitumor, antieczema, and antihypertensive properties. Additionally, these steroids function as apoptosis agonist agents, promoting programmed cell death. To illustrate the distribution of dominant activity against skin diseases, Figure 28 presents the percentage distribution based on the example of a borosteroid bearing a boron atom (**190**). This graph provides insights into the compound’s efficacy against skin diseases, emphasizing its potential therapeutic applications in this context.

Furthermore, Figure 29 presents a 3D plot showcasing the activity patterns of several boron-containing steroids (**192**, **193**, **194**, **195**, **196**, and **197**). This graph represents the predicted and calculated activity of these compounds, providing a visual representation of their potential effectiveness in various applications. The synthesis and evaluation of borosteroids contribute to our understanding of their unique biological properties and their potential as therapeutic agents. Their diverse activities, including their antitumor, antieczema, and antihypertensive properties, make them intriguing candidates for further research and development.

The study of boronosteroids and their biological activity holds significance for medicinal chemistry, as it provides insights into the design and development of novel therapeutic agents. Researchers explore their potential applications in drug discovery, disease treatment, and other areas of biomedical research. Furthermore, the broad spectrum of biological activities exhibited by borosteroids highlights their potential for diverse therapeutic interventions. Ongoing research in boron medical chemistry aims to elucidate the mechanisms of action and further optimize the properties of these compounds for specific therapeutic purposes. In summary, organoboron chemistry encompasses the study of borosteroids in relation to their biological activity. The significant number of known organic boronosteroids and their wide-ranging biological effects provide a foundation for further exploration and exploitation of their potential in drug discovery and other biomedical applications.

## 7. Organometallic Steroids: A Unique Class of Synthetic Compounds

The field of synthetic steroids has evolved extensively, with over 8000 known synthetic steroids and their derivatives documented to date [265]. Within this diverse landscape, organometallic steroids or steroids containing semi-metals stand out as a remarkable class of chemical compounds. These compounds, which are not found in nature, are exclusively synthesized, resulting in a vast array of chemical structures [265,266,267,268,269,270,271,272,273,274]. The concept of organometallic steroids was introduced in the mid-1950s by pioneering groups of scientists [274]. Over time, approximately 1000 synthetic organometallic steroids and their derivatives have been identified [265,274].

To provide a systematic overview, the organometallic steroids and related compounds can be categorized into seven distinct groups: *aluminum steroids*, *arsenosteroids, astatosteroids, germylated steroids, silasteroids, selenium steroids, tellurium steroids*, *and tin steroids*. These groupings allow for a more comprehensive understanding of the diverse nature of organometallic steroids and facilitate further exploration of their chemical properties and biological activities. The study and characterization of these unique compounds contribute to expanding our knowledge of synthetic steroids and offer potential applications in various fields. As synthetic compounds, they provide opportunities for innovation, particularly in medicinal chemistry, materials science, and other related disciplines. The division into distinct groups underscores the rich structural diversity and the potential for distinct properties and activities within each group. Further research in the field of organometallic steroids promises to unveil new insights and discoveries, leading to the development of novel compounds with potential therapeutic applications and other valuable uses.

### 7.1. Steroids Bearing Aluminum Atom

Steroids bearing an aluminum atom, or aluminum-containing steroids, are not commonly found or well-documented in scientific literature. Aluminum (**Al**) is not typically incorporated into the structure of steroids, which are organic compounds based on a specific carbon framework.

Aluminum (Al) is a paramagnetic metal known for its high thermal conductivity, electrical conductivity, and resistance to corrosion. When exposed to oxygen, aluminum forms a strong oxide film on its surface, which protects it from further interaction with the environment [275,276]. Although aluminum has various industrial applications, such as in construction, packaging, and electrical transmission lines, its incorporation into organic compounds, particularly steroids, is not a common occurrence.

Aluminum was first obtained by the Danish physicist Hans Christian Oerstedin in 1825 [277]. Since then, aluminum has become widely used in various industries due to its beneficial properties. However, it is important to note that aluminum compounds and aluminum-containing substances need to be handled and used appropriately to prevent potential health risks associated with excessive exposure or ingestion. In summary, while aluminum is a versatile metal with useful properties, steroids bearing an aluminum atom are not commonly encountered in scientific research or pharmaceutical applications. The focus of aluminum utilization typically lies in its industrial applications rather than its incorporation into organic compounds like steroids.

Organoaluminium chemistry is a specialized field within organometallic chemistry that focuses on the study of organic compounds containing aluminum and the chemical bonds formed between carbon and aluminum atoms [278,279]. Numerous articles have been published over the past 50 years exploring the properties and characteristics of organic aluminum compounds. In the last decade, several monographs have been published specifically dedicated to the chemistry of organo-aluminum compounds, consolidating the knowledge and incorporating citations from previous publications [280,281,282].

Aluminum-containing steroids (**202**–**209**), as depicted in Figure 30, represent a unique class of chemical compounds that have been synthesized and extensively studied. Their biological activity has been investigated and reported in various research articles [280,281,282,283,284]. These aluminum-containing steroids exhibit distinctive properties and activities. Table 12 presents the summarized activity profiles of these compounds. Notably, they have demonstrated antiprotozoal and anti-hypercholesterolemic activities with a level of certainty ranging from 78% to 92%. To visualize the activity patterns of selected steroids bearing an aluminum atom (**204**, **205**, **208**, and **209**), a 3D plot has been generated, as shown in Figure 31. This graph provides a graphical representation of the relationship between the predicted activity and the calculated activity of these compounds. The synthesis and characterization of aluminum-containing steroids have contributed to our understanding of their unique biological properties. These findings have implications for potential therapeutic applications and further research in this area.

The discovery of aluminum-containing steroids with potent antiprotozoal activity opens new possibilities for the development of novel antimalarial drugs. Further research and investigation into these compounds may shed light on their mechanism of action and potential as therapeutic agents against protozoan infections.

### 7.2. Steroids Bearing Arsenic Atom

Arsenic is a chemical element with the symbol (**As**) and an atomic number 33. It belongs to the group 15 elements in the periodic table and is classified as a metalloid. Arsenic is well-known for its toxic properties and has a long history of both beneficial and harmful uses [285,286]. Arsenic compounds, including hydrocarbons, lipids, phospholipids, fatty acids, and sugars, are indeed found in nature [285,286,287,288,289]. Arsenolipid analogs of phosphatidylcholine, sphingomyelin, and fatty acids have been identified in various organisms such as fish, crustaceans, lichens, mollusks, sponges, and different species of marine and freshwater invertebrates, as well as brown and green algae [290,291,292]. The presence of arsenic compounds in these organisms reflects their ability to metabolize and incorporate arsenic into their biological processes. The exact functions and roles of these arsenic-containing compounds in these organisms are still under investigation. It’s important to note that the concentration and distribution of arsenic compounds in these organisms can vary depending on factors such as environmental conditions, diet, and species-specific metabolic pathways. The understanding of the occurrence and significance of arsenic compounds in nature continues to evolve via ongoing research and analysis [286,287,288,289,290]. The discovery of arsenic compounds in diverse organisms highlights the complexity of arsenic biochemistry and its potential interactions with various biological systems. Further investigation is needed to elucidate the specific mechanisms of arsenic metabolism, the functions of these compounds, and their potential ecological implications [290,291,292].

Several studies have demonstrated that arsenolipids function as inhibitors of glycerol kinase, bovine carbonic anhydrase, and promyelocytic leukemia. They have also shown inhibitory effects on the growth of certain types of cancer cells [293,294]. These findings highlight the potential pharmacological and therapeutic applications of arsenolipids. In contrast to arsenolipids, naturally occurring *arsenosteroids* have not been identified. This absence in nature is likely attributed to challenges associated with the isolation and identification of these compounds. However, synthetic arsenosteroids (**210**–**215**, as depicted in Figure 32) constitute a small group of compounds that have been successfully synthesized and characterized. The biological activities of these compounds are summarized in Table 13, with all arsenosteroids exhibiting anti-cancer activity. The development and study of synthetic arsenosteroids offer valuable opportunities for advancing anti-cancer research and drug discovery. The unique properties of these compounds make them intriguing candidates for further investigation in the field of oncology.

To visualize the activity patterns of arsenic-containing steroids, a 3D plot has been generated, as depicted in Figure 33. This graph represents the predicted and calculated activity of the steroids bearing an arsenic atom (**210**, **211**, **212**, and **213**). The plot provides insights into the potential efficacy of these compounds as anti-cancer agents.

### 7.3. Steroids Bearing Astatine Atom

Astatine (**At**) is a chemical element with the symbol At and atomic number 85. It belongs to the halogen group on the periodic table and is the rarest naturally occurring element on Earth. Astatine is highly radioactive and has no stable isotopes, with its isotopes decaying rapidly over time [295,296]. Due to its radioactivity, astatine is challenging to study, and very limited quantities of the element are available for research purposes. As a result, our understanding of astatine’s properties and applications is still evolving. While its practical applications are limited, it has found utility in medical research, particularly in targeted cancer therapies. Further research is necessary to expand our knowledge of astatine and its potential applications in various fields.

Astatosteroids are steroids containing astatine that were first synthesized approximately 40 years ago [297]. Certain astatosteroids, including 2- and 4-astatoestradiol and 6-At-cholesterol (**216**, **217**, **218**, **219**, and **220** as depicted in Figure 34), have been successfully synthesized with high radiochemical yields using the reaction of ^211^At/I_2_ and the corresponding chloromercury compounds. Their stability in vitro has been evaluated under various conditions in comparison to analogous iodine compounds [295]. More recently, 6-astatomethyl-19-norcholest-5(10)-en-3β-ol (**220**) was synthesized with a yield of 60–70% [298]. The biological activity of astatosteroids has been investigated, as presented in Table 14. Notably, these compounds have exhibited distinct biological properties, including antineoplastic, anti-seborrheic, anti-secretoric, and anti-hypercholesterolemic activities. To visually represent the activity patterns of the astatine-containing steroids, a 3D plot has been generated, as shown in Figure 35. This graph provides insights into the predicted and calculated activity of the steroids bearing an astatine atom (**216** and **217**), aiding in the assessment of their potential efficacy.

The study and synthesis of astatosteroids contribute to expanding our understanding of their unique properties and potential applications in various fields. Their distinct biological activities make them intriguing candidates for further research and development in areas such as oncology, dermatology, and endocrinology. Further research is needed to investigate the mechanisms of action, stability, and potential therapeutic applications of astatosteroids. These studies will deepen our understanding of these compounds and their potential in addressing various medical conditions and diseases.

### 7.4. Steroids Bearing Germanium Atom

Germanium organic compounds refer to chemical compounds that contain both germanium and carbon atoms. Germanium is a chemical element with the symbol (**Ge**) and atomic number 32. While germanium is primarily known for its applications in the electronics and semiconductor industry, it can also form organic compounds via covalent bonding with carbon [299]. Organogermanium compounds are classified as organometallic compounds, specifically within the broader field of organometallic chemistry. These compounds play a significant role in organic synthesis, catalysis, and material science. The incorporation of germanium into organic molecules can impart unique properties and functionalities to these compounds [300].

Organogermanium compounds exhibit a diverse range of biological activities, including anti-tumor, antiviral, immunomodulating, neurotropic, cardiovascular, and radioprotective properties. These compounds have been studied extensively, and literature reports highlight their broad spectrum of biological activity. Notably, unlike silicon compounds, organogermanium compounds are practically non-toxic [301,302,303]. Additionally, germanium-containing heterocycles have been explored for their potential biological activities, including antimicrobial, antiviral, and anticancer properties. The anti-tumor activity of organogermanium compounds is particularly intriguing, as they have demonstrated potential as therapeutic agents against various types of cancer. Their antiviral activity suggests a role in combating viral infections, while their immunomodulating properties indicate the ability to modulate and regulate the immune system’s response.

Several germinated steroids (or *germanosteroids*) with a substitution at position 16 have been synthesized via the addition of trichlorogermane to a conjugated Δ16-double bond. Notably, the 16α-trichlorogermyl-3β-acetoxy-pregnan-20-ones (**221** and **222**, as depicted in Figure 36) and the 16α-trimethylgermyl-progesterones (**223**–**228**) have been synthesized and found to exhibit remarkable stability [304,305,306]. The structural details of these germinated steroids can be observed in Figure 36, while Table 15 provides an overview of their biological activity. The synthesis and characterization of these compounds have shed light on their potential applications in the field of medicinal chemistry.

The 16α-trichlorogermyl-3β-acetoxy-pregnan-20-ones (**221** and **222**) and the 16α-trimethylgermyl-progesterones (**223**–**228**) have demonstrated notable stability, indicating their ability to withstand various conditions without significant degradation or transformation. The biological activity of these germinated steroids holds promise for their potential therapeutic applications. Further research is needed to fully elucidate their mechanisms of action and explore their efficacy in various biological contexts. The synthesis and study of germinated steroids contribute to expanding our knowledge of their unique properties and potential applications. These compounds open up new avenues for drug discovery and development, with implications for the field of endocrinology, reproductive health, and related disciplines. Continued research in this area will deepen our understanding of germinated steroids, their stability, and their potential as therapeutic agents. The exploration of their biological activity offers valuable insights into their potential applications and paves the way for the development of novel treatments and interventions in the field of medicine.

An uncommon, germinated steroid (**229**) has been synthesized from a Δ16-allopregnene-20-one [307]. The specific biological activity of this compound has not been reported in the literature. However, the predicted biological activities of germinated steroids are presented in Table 16, providing insights into their potential therapeutic applications. While the specific biological activity of the germinated steroid (**229**) remains to be reported, its synthesis and characterization contribute to expanding our understanding of these unique compounds. Further research is needed to explore their mechanisms of action, evaluate their stability, and assess their potential in various therapeutic applications. Among the distinctive biological properties exhibited by germinated steroids, notable characteristics include antineoplastic activity, indicating their potential as agents against cancer, as well as anti-seborrheic and dermatologic activities, suggesting their relevance in addressing skin conditions. These activities highlight the diverse therapeutic potential of germinated steroids.

To visualize the predicted activity patterns of steroids bearing a germanium atom (**227**, **228**, and **229**), a 3D plot has been generated, as depicted in Figure 37. This graph offers a visual representation of the compounds’ potential biological activity, aiding in the assessment of their efficacy in various contexts.

The understanding of organogermanium compounds continues to evolve, offering opportunities for the development of novel therapeutic agents and materials with unique properties. Further studies are necessary to fully uncover the potential of these compounds and harness their benefits for the advancement of medicine and related fields. The study of germinated steroids continues to advance, providing opportunities for the discovery of new bioactive compounds and the development of innovative treatments for various diseases, including cancer and dermatological conditions.

### 7.5. Steroids Bearing Silicon Atom(s)

Silicon and organosilicon compounds are chemical compounds that contain silicon atoms. Silicon is a chemical element with the symbol (**Si**) and atomic number 14. It is a metalloid that is widely abundant in the Earth’s crust and plays a crucial role in various fields, including materials science, electronics, and organic chemistry [299,308,309]. Organosilicon compounds, on the other hand, are organic compounds that contain silicon-carbon (Si-C) bonds. These compounds are formed by replacing hydrogen atoms in organic molecules with silicon atoms. The introduction of silicon imparts unique properties and characteristics to the organic compounds, such as increased thermal stability, improved resistance to oxidation, and enhanced compatibility with silicon-based materials. The unique properties of silicon and organosilicon compounds make them indispensable in modern technology, manufacturing, and scientific research. Their versatility, stability, and compatibility with a wide range of materials continue to drive innovations and advancements in various industries [308,309].

Silicon-containing steroids, commonly referred to as *silasteroids*, have been synthesized with the intention of exploring their potential as oestrogenic agents, anti-estrogenic agents, and antifertility agents [310,311,312,313]. Silasteroids (**230**–**235**, as depicted in Figure 38) with a silicon atom in position 6 represent a significant portion of the synthesized steroids [314,315]. These compounds have been investigated for their biological activity, as summarized in Table 16.

Silasteroids exhibit notable properties, particularly in terms of their antineoplastic and psychotropic activities. The antineoplastic activity of these compounds suggests their potential use in combating cancer. Additionally, their psychotropic activities indicate potential effects on the central nervous system and mental processes. The exploration of these properties opens up possibilities for further research and potential applications in the fields of oncology and neuropsychiatry. The structures and biological activities of silasteroids (**230**–**236**) provide valuable insights into their potential therapeutic applications. However, it is important to note that further research is necessary to fully understand their mechanisms of action, optimize their properties, and assess their potential in clinical settings.

Steroids containing silicon at positions 10 (**237**) and 13 (**238**) have been synthesized [316,317,318], but their specific biological activities have not been extensively studied. However, both compounds exhibit antineoplastic activity and possess a Pa value greater than 0.97, indicating high predicted activity. Further investigation is needed to fully understand their biological effects.

Silasteroids (**230**–**239**) are characterized by a carbon-silicon (C-Si) bond, while in silasteroids (**240** and **241**), a carbon-oxygen-silicon (C-O-Si) bond is present. This distinction significantly influences the biological activity of these steroids. The biological activities of other silasteroids are summarized in Table 16, showcasing a range of potential therapeutic properties.

To visualize the predicted activities of steroids bearing a silicon atom (**230**, **231**, **232**, **236**, and **239**), a 3D graph has been generated, as depicted in Figure 39. This graph provides insights into the potential efficacy and activity patterns of these compounds, aiding in the assessment of their biological effects. 

Silasteroids represent a unique class of synthetic steroids with distinct properties and potential applications. Ongoing research in this field aims to uncover the full range of their biological activities, elucidate their mode of action, and explore their therapeutic potential in various disease areas. The study of silasteroids contributes to the advancement of medicinal chemistry and drug discovery, offering opportunities for the development of novel treatments and interventions. Further investigations will deepen our understanding of these compounds and potentially lead to new therapeutic strategies in the field of oncology and mental health.

The study of silasteroids expands our knowledge of the diverse chemical structures and biological activities that can be achieved using silicon incorporation in steroids. Further research is required to unravel the precise mechanisms of action, evaluate their stability, and investigate their potential applications in various disease treatments.

Continued exploration of silasteroids and their biological activities holds promise for the development of novel therapeutic agents in the field of oncology and beyond. Further studies will deepen our understanding of these compounds and potentially lead to the discovery of new treatment modalities and targeted interventions.

### 7.6. Steroids Bearing Selenium Atom

Steroids bearing a selenium (**Se**) atom have attracted significant attention due to the unique properties and potential biological activities associated with selenium. Selenium is a chemical element in the 16th group of the periodic table and was discovered by Jöns Jacob Berzelius, a Swedish chemist, in 1817 [319].

Selenium is considered an essential trace element for human health, playing a crucial role in various biological processes. It is involved in regulating metabolism and is necessary for the proper functioning of several enzymes and proteins in the body [320]. Selenium deficiency can have adverse effects on human health, emphasizing the importance of monitoring its presence in the diet and ensuring adequate intake [321]. Steroids bearing a selenium atom offer a unique avenue for exploring the potential therapeutic applications of selenium. These compounds have been studied for their biological activities and their influence on various physiological processes. Selenium-containing steroids have shown promise in areas such as antioxidant activity, anti-inflammatory effects, and anticancer properties. The incorporation of selenium into steroids can modulate their biological properties and potentially enhance their therapeutic efficacy.

The *Allium* and *Brassica* families, along with Brazil nuts, mushrooms (shiitake and white mushrooms), beans, chia seeds, brown rice, sunflower, sesame, flax seeds, cabbage, and spinach, are known to contain significant concentrations of selenium and organoselenium compounds [322,323]. These natural sources provide a dietary means of obtaining selenium. In recent years, numerous books have been published that delve into the chemistry, biology, and medical applications of organoselenium compounds [324,325]. Additionally, there are several excellent reviews in the scientific literature that explore the biological roles and functions of these compounds [326]. 

The biological activities and potential health benefits of organoselenium compounds have garnered significant interest. These compounds have been studied for their antioxidant properties, anticancer effects, antimicrobial activity, and potential roles in various physiological processes. The unique properties of organoselenium compounds make them valuable targets for drug discovery and development. The exploration of organoselenium compounds provides insights into their chemical reactivity, biological effects, and potential therapeutic applications [324,325,326]. Ongoing research in this field aims to uncover new compounds, elucidate their mechanisms of action, and assess their potential for addressing various diseases and health conditions.

Selena steroids represent a significant group of essential metalloids that have been synthesized and studied over the past 50 years, with approximately 300 compounds reported [327,328,329]. These compounds can be tentatively categorized into four groups based on their structural features.

The first group comprises selena steroids where the selenium atom is incorporated into the heterocyclic core of the molecules (**242**–**244**, as shown in Figure 40). These compounds exhibit notable antineoplastic and anti-seborrheic activities. Moreover, they show potential for the treatment of Alzheimer’s disease, as indicated in Table 17.

The incorporation of selenium into the heterocyclic core of these selenasteroids introduces unique properties and potential. The selenasteroids can be further classified into additional groups based on the position of the selenium atom within the steroid molecule. The second group comprises selenasteroids in which the selenium atom is located at the second and third positions of the steroid structure (**245**–**249**). These compounds exhibit prominent activities such as antineoplastic effects, anti-hypercholesterolemia properties, and anti-inflammatory properties. The third group includes selenasteroids, with the selenium atom positioned at the sixth position of the core molecule (**250**–**256**). For selenasteroids in this group (**257**–**262**, as illustrated in Figure 40), the main activities observed are respiratory analeptic effects, anesthetic properties, and anti-hypercholesterolemic activities. Moreover, these compounds have the potential to be used as chemopreventive and hepatoprotectant agents. 

The fourth group encompasses selenasteroids, in which the selenium atom is situated in the hydrocarbon tail of the steroid structure (**263**–**267**, structures see in Figure 41). The specific activities associated with these selena steroids vary, and further research is needed to fully understand their properties and potential applications. To visualize the predicted activities of steroids bearing a selenium atom, a 3D plot has been generated, as depicted in Figure 42. This plot provides an overview of the potential biological activities of steroids bearing selenium atoms (**244**, **245**, and **250**), aiding in the assessment of their therapeutic potential. The classification of selenasteroids into distinct groups based on the position of the selenium atom allows for a better understanding of their structural diversity and associated activities. Further studies are required to unravel the precise mechanisms of action and optimize the properties of these compounds for potential therapeutic applications. The exploration of selenasteroids expands our knowledge of the diverse range of biological activities that can be achieved using selenium incorporation. Continued research in this field holds promise for the development of novel therapeutic agents and the advancement of medicine in areas such as oncology, cardiovascular health, and respiratory disorders.

The study of selena steroids exemplifies the interdisciplinary nature of medicinal chemistry and chemical biology, showcasing the exploration of essential metalloids and their potential applications in disease treatment and management. Continued investigations into selena steroids will advance our knowledge and may lead to the discovery of novel treatments and interventions in the fields of oncology, dermatology, and neurodegenerative diseases. Continued research in this area holds promise for discovering new therapeutic agents and advancing our knowledge of the vital roles played by selenium and selena steroids in human health. The investigation of their biological activities highlights their potential as anticancer agents, particularly against neoplastic conditions, and suggests their relevance in addressing seborrheic disorders. Furthermore, their potential use in the treatment of Alzheimer’s disease showcases the diverse range of applications for selena steroids.

### 7.7. Steroids Bearing Tellurium Atom

Tellurium is a metalloid with toxic properties and was discovered by Franz-Joseph Müller von Reichenstein in 1782 [330]. Analyzing the scientific literature on tellurium, I concluded that very little attention is paid to this element, and may partially reflect the very low abundance of this element in nature. In fact, recent discoveries and well-established observations clearly show that this assumption is wrong, and the growing importance of the unique properties of tellurium compounds is evident from the variety of their known and potential uses in both inorganic and organic chemistry [331,332,333,334,335].

Steroids bearing a tellurium (**Te**) atom represent a unique class of compounds that have been of interest to researchers. Due to the toxicity of tellurium, the synthesis and study of tellurium-containing steroids require careful handling and safety precautions. The incorporation of tellurium into steroid molecules (**268**–**280**, as shown in Figure 43) introduces unique chemical properties and potential biological activities, which make these compounds of interest in medicinal chemistry and related fields [336].

However, it is important to note that the research on steroids bearing tellurium atoms is limited, and their specific biological activities and potential applications are still being explored. Further studies are needed to fully understand the properties, reactivity, and potential therapeutic uses of tellurium-containing steroids. The study of tellurium-containing steroids is a niche area of research, and ongoing investigations aim to expand our knowledge and explore their potential applications. It is important to conduct further research to understand the properties, mechanisms of action, and potential toxicological effects of these compounds before considering their practical applications [337,338,339,340]. To visually illustrate the predicted activities of steroids bearing a tellurium atom, a 3D plot has been generated, as depicted in Figure 44. This plot provides an overview of the potential biological activities of steroids bearing tellurium atoms (**271**, **272**, **274**, **275**, and **276**), aiding in the assessment of their therapeutic potential.

For tellurasteroids (**268**–**272**), where tellurium is incorporated into the steroid skeleton, several fundamental properties have been observed. These compounds exhibit antioxidant, anti-inflammatory, antineoplastic, and antiprotozoal activities, making them potential candidates for various therapeutic applications. Additionally, they show promise as anti-parkinsonian, anti-Alzheimer’s disease, and anti-neurodegenerative agents. The biological activities of other tellura steroids (**276** and **277**) are detailed in Table 18. These compounds exhibit a range of activities that include antioxidant, anti-inflammatory, antineoplastic, and potentially other properties that are yet to be fully explored. In the case of tellura steroids (**279** and **280**), their characteristic properties include anti-inflammatory, antioxidant, and anticancer activities. These compounds hold potential for the treatment of inflammatory conditions and have shown promising effects as antioxidants and in combating cancer, as indicated in Table 18.

The investigation of tellura steroids showcases their diverse range of biological activities and potential applications in various fields, including anti-inflammatory, antioxidant, and anticancer therapies. Further research is warranted to elucidate the underlying mechanisms of action and optimize the properties of these compounds for potential clinical use. It is important to note that the specific structure and activity of each compound may vary, and further studies are needed to fully understand their potential therapeutic applications. The presented data serve as a starting point for the exploration of tellura steroids and their potential roles in addressing various diseases and health conditions.

The biological activity of tellurasteroids holds promise for various applications, ranging from potential therapeutic agents to drug discovery and development. The exploration of their properties and mechanisms of action is of great interest to researchers in the pursuit of novel treatments and interventions. While our research may not directly delve into organotellurium chemistry, the significance of tellura steroids and their potential impact on medicine and pharmacology cannot be overlooked. Continued investigations in this area may lead to the discovery of new bioactive compounds and further our understanding of the role of tellurium-containing compounds in various biological processes [332,336,337].

### 7.8. Steroids Bearing Tin Atom or Organotin Steroids

Organostannanes, also known as organotin compounds, are a class of chemical compounds that contain carbon-tin (C-Sn) bonds. These compounds are characterized by the presence of one or more organic groups (such as alkyl or aryl groups) bonded to a tin atom [341,342,343]. Organostannanes have a wide range of applications in various fields, including organic synthesis, catalysis, and materials science. They can serve as reagents in organic reactions, where the tin atom undergoes transformations while the organic groups remain intact. Organostannanes are particularly useful in cross-coupling reactions, where they can be coupled with other organic halides or pseudohalides to form new carbon-carbon or carbon-heteroatom bonds [343,344,345,346]. The synthesis of stannanes was initially achieved by Edward Frankland and later by Carl Jacob Löwig in the 1850s [347,348]. Organotin compounds find applications in various fields, including organic synthesis of organometallic components in the industry. Additionally, these compounds have been explored in medicine and pharmacology despite their known toxicity. Alkyl stannanes exhibit potent bactericidal and fungicidal properties, making them less commonly used in agriculture and related sectors [349,350,351,352,353].

Furthermore, numerous organotin compounds have shown promising potential in terms of their anticancer and anti-tuberculosis activities [354,355]. These findings highlight the diverse biological activities associated with organotin compounds. By understanding the chemistry, applications, and biological activities of organotin compounds, researchers can further explore their potential in drug development and other relevant areas.

The concept of organometallic steroids emerged in the mid-1950s and was pioneered by several groups of scientists [356,357]. In our study, we have specifically focused on 18 stable organotin steroids, which exhibit intriguing properties in the field of medicine and pharmacology [358]. These compounds have demonstrated notable anti-tumor, antiviral, and antibacterial activities [344,354,355,359,360]. By incorporating organotin moieties into steroids, we aim to explore their potential as novel therapeutic agents. The combination of organic and metal components offers exciting possibilities for enhanced biological activities and targeted drug delivery systems. Our findings highlight the promising prospects of organotin steroids in various therapeutic applications.

Organotin steroids, characterized by the presence of C-Sn and/or C-O-Sn chemical bonds, represent a distinctive and non-natural class of chemical compounds. These compounds are exclusively synthesized and exhibit a vast array of chemical structures [358,359,360]. Their unique nature and diverse structures make them of significant interest in various research domains, as well as in the medical and pharmaceutical industries [361]. The utilization of organotin and organometallic steroids in research offers valuable insights into their potential applications and therapeutic benefits. These compounds serve as important tools for studying structure-activity relationships, elucidating biological mechanisms, and developing novel pharmaceutical agents. Their wide-ranging applications span fields such as medicinal chemistry, drug discovery, and pharmacology.

Organotin steroids (**281**–**298**), as depicted in Figure 45, were synthesized via the reaction of various organotin reagents with steroids [358,359,360]. These synthetic steroids have exhibited remarkable properties, including inhibiting the growth of malignant tumors, as well as being employed as insecticides, larvicides, bactericides, and fungicides [341,342,358,359,360,361,362,363]. In summary, organotin steroids, with their non-natural origin and diverse structures, hold considerable significance in research, as well as in the medical and pharmaceutical industries. Their utilization aids in advancing scientific knowledge, exploring new therapeutic possibilities, and driving innovation in drug development. Table 19 provides an overview of the structures and biological activities of the organotin steroids (**281**–**298**). These compounds have shown diverse biological effects, highlighting their potential In therapeutic applications.

To visualize the activity patterns of selected tin-containing steroids (**282**, **283**, **284**, and **289**), a 3D plot has been generated, as illustrated in Figure 46. This plot allows for the analysis and comparison of their predicted and calculated activities, providing valuable insights into their efficacy. Furthermore, Figure 47 presents the percentage distribution of the dominant antitumor activity specifically for the organotin steroid (**292**). This graph provides a visual representation of the compound’s effectiveness against tumor growth.

The synthesis and evaluation of organotin steroids offer opportunities for the development of novel therapeutic agents and their use in various applications. Their multifaceted biological activities make them intriguing targets for further research and investigation, with the potential for significant contributions to the fields of medicine, agriculture, and pharmaceuticals.

Furthermore, the versatility and synthetic accessibility of organotin steroids contribute to their widespread use in the medical and pharmaceutical sectors. Researchers explore their potential as therapeutic agents, drug delivery systems, and imaging agents, among other applications. The unique properties bestowed by the organotin moiety open avenues for innovation and the development of new chemical entities with enhanced biological activities and targeted properties.

## 8. Metallocene Steroid Conjugates

Metallocene steroid conjugates represent a unique class of compounds that combine the structural features of metallocenes and steroids. Metallocenes are organometallic compounds consisting of a transition metal atom sandwiched between two cyclopentadienyl (Cp) ligands. Steroids, on the other hand, are naturally occurring or synthetic compounds that share a common tetracyclic ring system [364,365,366,367]. The synthesis of metallocene steroid conjugates involves the covalent attachment of a metallocene moiety to a steroid scaffold, resulting in the formation of a single molecule with combined properties. These conjugates have attracted significant attention due to their potential for modulating biological activities and offering unique chemical features [366,367,368,369].

The incorporation of metallocene moieties into steroid frameworks introduces novel properties and functionalities, expanding the scope of potential applications. Metallocene steroid conjugates have shown promise in various areas, including medicinal chemistry, drug delivery systems, and catalysis [368,369,370]. In medicinal chemistry, these conjugates have demonstrated enhanced biological activities compared to their parent steroids, including improved pharmacokinetics, increased receptor affinity, and altered drug distribution. The presence of the metallocene moiety can confer unique properties such as redox activity, enzyme inhibition, or metal coordination, allowing for targeted therapeutic interventions [364,365,366,367,368,369,370].

Metallocene steroid conjugates also find applications in drug delivery systems, where the metallocene moiety can serve as a platform for controlled release mechanisms or as a targeting moiety to specific cellular receptors or tissues. Their ability to combine the pharmacological properties of steroids with the distinctive features of metallocenes opens new possibilities for targeted and personalized therapies [364,365,366,367,368,369,370,371,372,373,374].

### 8.1. Ferrocene Steroid Conjugates

Ferrocene steroid conjugates (**299**–**323**, as shown in Figure 48) are a fascinating class of compounds that combine the structural motifs of ferrocene and steroids. Ferrocene, a metallocene consisting of an iron atom sandwiched between two cyclopentadienyls (Cp) ligands, exhibits unique electronic and redox properties [372,373,374,375,376]. Steroids, on the other hand, possess a tetracyclic ring system and are known for their diverse biological activities. The synthesis of ferrocene steroid conjugates involves the covalent attachment of a ferrocene moiety to a steroid scaffold, resulting in the formation of a single molecule that combines the properties of both components. These conjugates have garnered significant interest due to their potential in various fields, including medicinal chemistry, materials science, and catalysis [365,374,375,376,377].

Ferrocene was first synthesized at Duquesne University in 1951 in Pittsburgh, Pennsylvania, by chemists Kealy and Paulson, and such compounds are also known as sandwich compounds [378]. Originally, synthesized ferroquine was interesting as a commercial antimalarial drug containing a ferrocene group, and now it has successfully passed clinical trials and is used in practical medicine [379,380,381]. The anticancer properties of ferrocene derivatives were first studied in the late 1970s of the XX century [382,383,384,385]. Ferrocene androgens have shown potential in blocking the development of prostate cancer. By binding to the androgen receptor in place of testosterone, ferrocene androgens can enhance the antitumor effect and counteract the androgen-induced hormonal effects [379,386]. 

In a study by Manosroi et al. [387], ferrocene testosterone was synthesized by introducing a ferrocenyl group into the C-2 position of the testosterone skeleton (**299**, as depicted in Figure 48). The biological activity of this hormone was evaluated, and it demonstrated high cytotoxic activity against HeLa cells. Another compound, ferrocenyl dihydrotestosterone (**300**), was synthesized, but its specific biological activity has not been determined [388]. In a study conducted by the same authors, two ferrocenic steroidal compounds (**301** and **302**) were investigated for their activity against the human cervical adenocarcinoma cell line (HeLa) and compared to doxorubicin [387]. The growth inhibition of cells, as measured by the GI_50_ value (the concentration required for 50% cell growth inhibition), was determined for each compound. Interestingly, both ferrocenic steroids (**301** and **302**) exhibited GI_50_ values comparable to those of doxorubicin, a widely used anticancer drug. The GI_50_ values for the ferrocenic steroids were found to be 0.27 and 0.25 μg/mL, respectively. These results indicate that both ferrocenic steroidal compounds possess a significantly potent anti-proliferative activity against HeLa cells, surpassing the activity of testosterone or methyl testosterone alone.

A novel compound, estradiol, linked by a sulfide bridge with ferrocenyl at position 7α (**303**), has been investigated for its estrogenic effects and cytotoxic activity against breast cancer cell lines. This compound has shown estrogenic effects in MCF-7 cells and exhibited cytotoxic activity against MDA-MB-231 breast cancer cells at low micromolar concentrations [389].

Furthermore, two ferrocenoyl esters, namely ferrocenoyl 17β-hydroxyestra-1,3,5(10)-trien-3-olate (**305**, also known as estradiol ferrocenoylate) and ferrocenoyl 3β-estra-1,3,5(10)-trien-17-one-3-olate (**306**, also known as estrone ferrocenoylate), have been evaluated for their anti-proliferative activity. Both compounds exhibited significant anti-proliferative effects against MCF-7 cells, a breast cancer cell line. Additionally, the steroid compound (**306**) demonstrated activity against colon cancer HT-29 cells [390].

Among the studied ferrocenes, compounds (**304**–**308** and **314**–**316**) exhibited lower anti-proliferative activity. However, one notable compound, ferrocenoyl steroid (**310**), demonstrated significant anti-proliferative activity in MCF-7 cells [385,389,391,392,393,394,395]. In addition, derivatives of testosterone (**312**) and dihydrotestosterone (**313**) containing the ferrocenyl group at the C-17 position of the steroid skeleton exhibited potent anti-proliferative activity against PC-3 prostate cancer cells. The IC_50_ values for these compounds were measured as 4.7 and 8.3 mM, respectively [396].

While ferrocene-containing steroid (**311**) was synthesized, its pharmacological activity has not been investigated in depth [397,398]. Various ferrocenyl cholesterol derivatives, such as compounds (**317**, **318**, **322**, and **323**), have been synthesized by different researchers, but their specific biological activities have not been determined [399,400,401]. Among these derivatives, the ferrocenoyl derivative of (3β,5Z,7E,22E)- 9,10-secoergosta-5,7,10(19),22-tetraen-3-olate (**319**) exhibited the lowest anti-proliferative activity [390].

In a study conducted by Balogh et al. [402], a steroidal ferrocenyl chalcone (**321**) was synthesized. Additionally, 16-(ferrocenylmethyl)-amino estratriene (**320**) demonstrated anti-microbial activity against *Staphylococci* [403]. The structures of ferrocene steroid conjugates can be observed in Figure 48 and Figure 49, and their corresponding biological activities are summarized in Table 20.

The synthesis and structural characterization of ferrocene steroid conjugates (**310**, **324**–**331**) were carried out, and their molecular structures were determined using single-crystal X-ray diffraction techniques. In particular, the positioning of the ferrocene moiety in relation to the steroid framework was investigated. Steroid conjugates (**329**) and (**330**) showed the ferrocene group positioned in the β-face of the steroid, while in the case of (**310**), it was positioned between the α- and β-faces.

The anti-proliferative activity of these ferrocene steroid conjugates was evaluated using colon cancer HT-29 and breast cancer MCF-7 cell lines (as shown in Table 20). The conjugates displayed varying degrees of anti-proliferative activity, ranging from moderate to high, against both cell lines. Notably, 16-ferrocenylidene-3α-hydroxy androstan-17-one demonstrated the most significant activity against HT-29 cells [396]. To visualize the activity of the ferrocene steroid conjugates, a 3D graph (Figure 50) was constructed, illustrating the relative activity of compounds such as (**311**), (**317**), (**320**), and (**321**).

These findings highlight the potential of ferrocene androgens as promising candidates for prostate cancer treatment. The introduction of the ferrocenyl group into androgenic compounds opens new avenues for developing targeted therapies and improving the efficacy of existing treatments. Continued research in this field will provide further insights into the biological activities and potential clinical applications of ferrocene androgens.

In medicinal chemistry, ferrocene steroid conjugates have shown promising biological activities, such as anticancer, anti-inflammatory, and antimicrobial properties. The incorporation of the ferrocene moiety into the steroid framework can modulate the pharmacokinetics, enhance cellular uptake, and improve the therapeutic efficacy of the conjugates. The unique redox properties of ferrocene also offer opportunities for redox-based therapies and targeted drug delivery systems.

Continued research in this area holds great potential for the development of novel therapeutic agents, advanced materials, and efficient catalytic systems. The versatility and unique properties of ferrocene steroid conjugates make them intriguing candidates for further exploration and offer exciting opportunities for advancements in diverse scientific fields.

### 8.2. Titanocene Steroid Conjugates

Titanocene steroid conjugates are a class of compounds that combine a titanocene moiety with a steroid framework. Titanocene, a transition metal complex containing titanium, has attracted interest in medicinal chemistry due to its potential anticancer properties [376,377,404]. The synthesis and characterization of titanocene steroid conjugates have been explored in several studies. These conjugates typically involve the attachment of titanocene to various positions of the steroid backbone, such as the C-17 or C-21 position, via appropriate linkers [405,406,407]. The presence of the titanocene moiety introduces unique structural and chemical features that can modulate the biological activity of the conjugates.

The anticancer activity of titanocene steroid conjugates has been investigated in various cancer cell lines. These conjugates have shown promising cytotoxic effects and have demonstrated the ability to inhibit cancer cell growth via different mechanisms, including DNA binding and disruption of cellular processes [407,408,409]. The specific targeting of hormone-responsive cancers, such as breast and prostate cancers, has also been explored by incorporating steroid components into the conjugates [409].

Eight titanocene steroid conjugates (**332**–**339**) have been synthesized, and their structures have been characterized in previous studies [364,365,393,394,395]. The anti-proliferative activity of these conjugates was evaluated in colon cancer HT29 and breast cancer MCF-7 cell lines. Among the titanocene cholesterol derivatives, complex (**336**) exhibited significant activity, while others showed IC_50_ values over 200 μM. On the other hand, titanocenyls containing sex steroid derivatives displayed higher cytotoxicity, with IC_50_ values below 50 μM against the MCF-7 cell line. Conjugates containing cholesterol units (**332**, **338**, and **339**) demonstrated lower cytotoxicity [364,365,395,396,397,398].

The structures of the titanocene steroid conjugates can be observed in Figure 51 and Figure 52, and their corresponding biological activities are summarized in Table 21. To visualize the activity of a specific conjugate, a 3D plot was created, showing the activity of titanocene steroid conjugate (**338**) (Figure 53). Additionally, the percentage distribution of the dominant activity of this specific titanocene steroid conjugate is illustrated in Figure 54.

Furthermore, the use of titanocene steroid conjugates as drug delivery systems has been investigated. By utilizing the unique properties of titanocene, these conjugates can potentially enhance drug delivery, improve bioavailability, and increase therapeutic efficacy [395,396,397,398]. The development of titanocene steroid conjugates holds promise for the design of novel anticancer agents with improved selectivity and reduced toxicity. Further research is needed to optimize the structures, evaluate the pharmacokinetics, and explore the mechanisms of action of these conjugates.

## 9. Conclusions

Heteroatom steroids are a remarkable class of organic compounds that have shown immense potential in the field of medicinal chemistry. The incorporation of heteroatoms into the steroid framework imparts unique chemical and biological properties, making them attractive candidates for therapeutic applications. These compounds have demonstrated diverse biological activities, including antineoplastic, anti-inflammatory, antimicrobial, antiviral, and immunomodulatory effects. 

The potential of heteroatom steroids extends beyond their individual activities. Their incorporation into drug discovery programs and pharmaceutical development offers opportunities for the development of novel therapeutics and the improvement in existing treatment options. Overall, heteroatom steroids represent a rich area of research with significant implications for the fields of medicinal chemistry, pharmacology, and pharmaceutical sciences. Continued investigation into the synthesis, biological activities, and mechanisms of action of these compounds will drive advancements in drug discovery and ultimately benefit patients by providing new and improved treatment options for various diseases.

The steroids containing heteroatoms presented in this review are of great interest for clinical medicine, and we can recommend epithio steroids containing the thiirane group, which demonstrate strong antineoplastic activity, or steroids bearing aluminum atom, which show antiprotozoal (*Plasmodium*) activity.

## Figures and Tables

**Figure 1 biomedicines-11-02698-f001:**
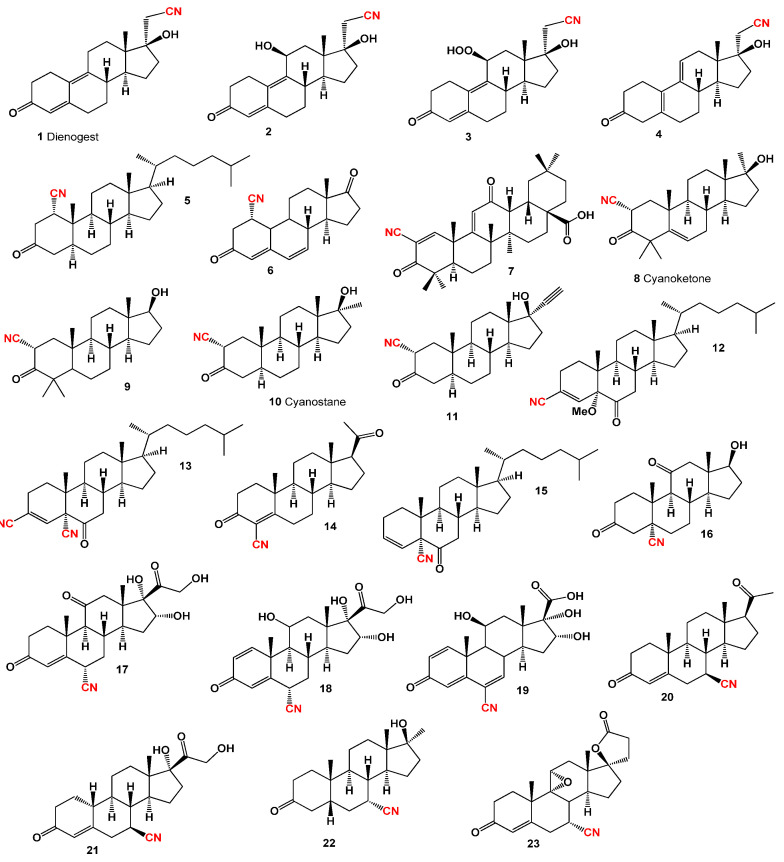
Structures of Bioactive steroids bearing nitrile group(s).

**Figure 2 biomedicines-11-02698-f002:**
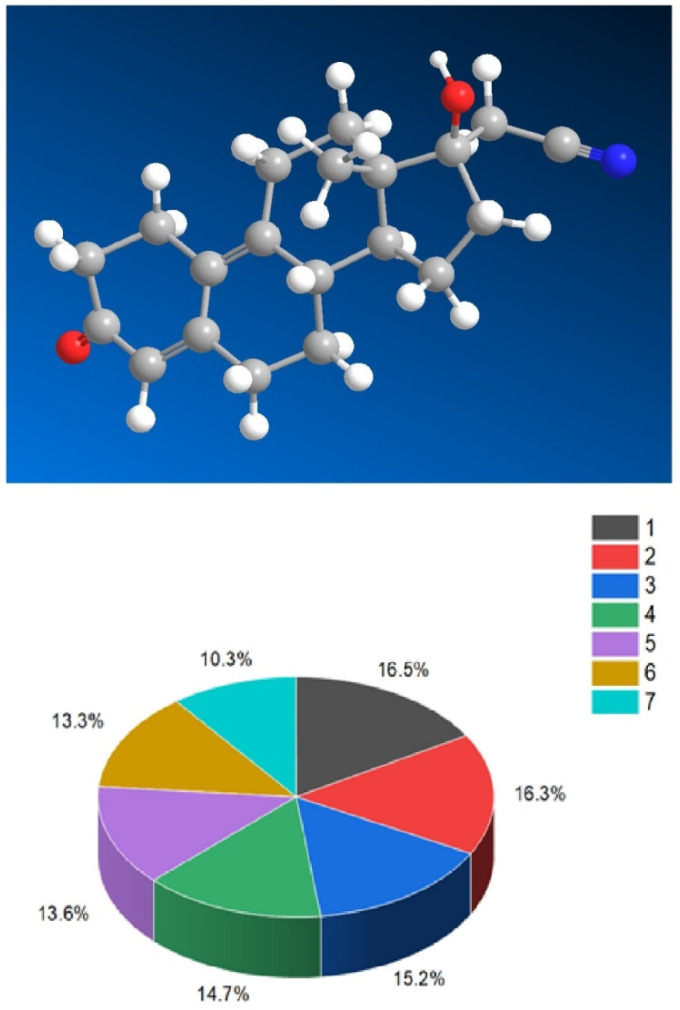
3D model and percentage distribution of dominant and related biological activities illustrated by the example of dienogest (**1**), a widely recognized semi-synthetic 19-nortestosterone derivative with unique pharmacological properties. The activities indicated by the numbers are as follows: 1. *Menstruation disorders treatment* (16.5%), 2. *Menopausal disorders treatment* (16.3%), 3. *Ovulation inhibitor* (15.2%), 4. *Contraceptive* (14.7%), 5. *Anti-inflammatory* (13.6%), 6. *Androgen antagonist* (13.3%), and 7. *Endometriosis treatment* (10.3%). Dienogest is highly selective for the progesterone receptor, exerting a potent progestogenic effect and a moderate anti-gonadotropic effect. It does not possess androgenic, glucocorticoid, or mineralocorticoid activity. Its resemblance to norethisterone is evident in its remarkable endometrial efficacy, which contributes to the stability of the menstrual cycle in women. Dienogest’s robust endometrial efficacy underlies its application in the treatment of endometriosis and provides anti-proliferative and anti-inflammatory effects for the management of endometriotic lesions. Nitrile group is marked in blue.

**Figure 3 biomedicines-11-02698-f003:**
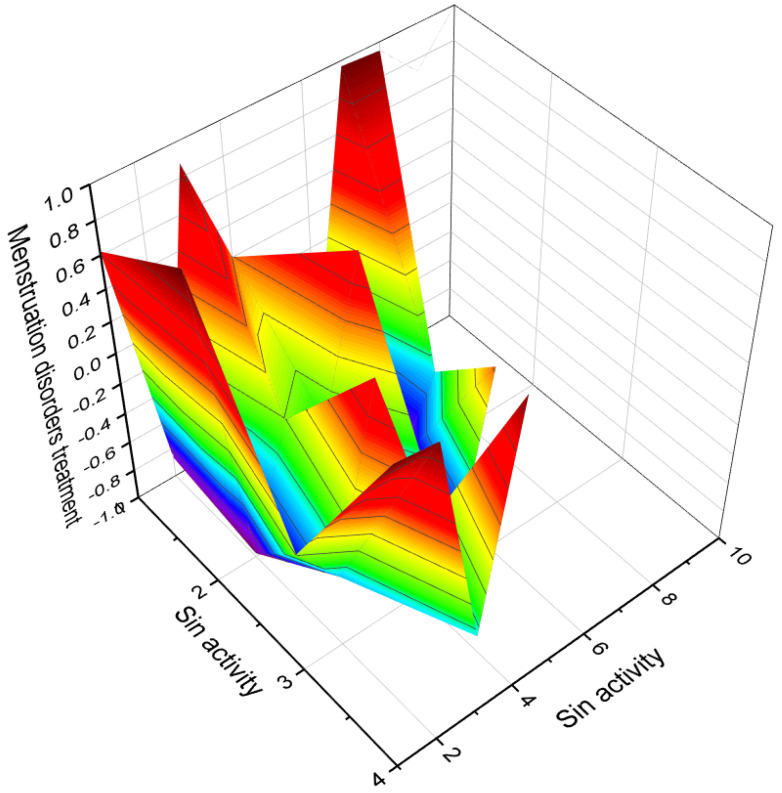
3D Graph illustrating the predicted and calculated activity of cyanosteroids (**1**, **2**, **4**, and **20**) with 92% confidence for the treatment of menstruation disorders and related diseases. These cyanosteroids, featuring a nitrile group, have exhibited noteworthy properties in the treatment of various conditions. They have demonstrated efficacy in addressing menstrual and menopausal disorders, muscular dystrophy, and male reproductive dysfunction and have shown high certainty as potent ovulation inhibitors. These findings have garnered significant attention in the field.

**Figure 4 biomedicines-11-02698-f004:**
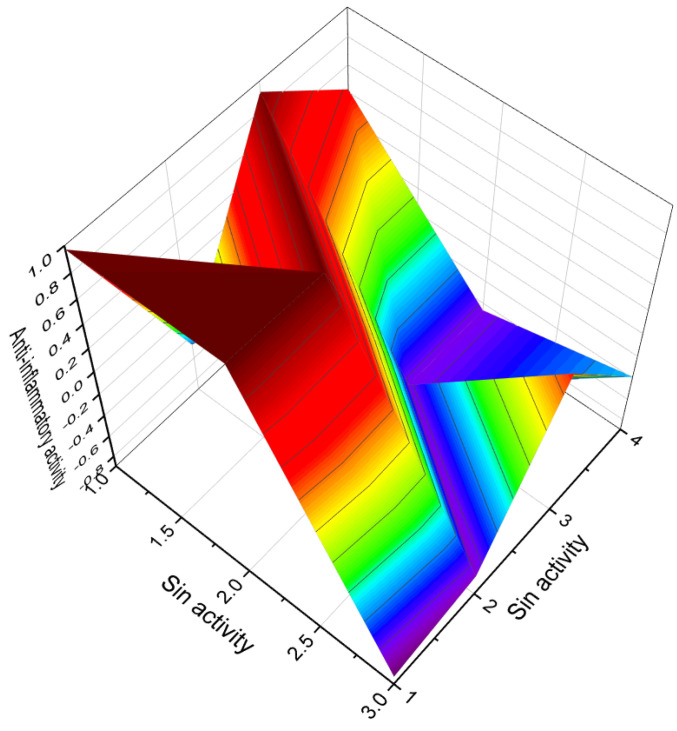
3D Graph depicting the predicted and calculated anti-inflammatory activity of cyanosteroids (**17**, **18**, and **19**) with over 93% confidence. The 6-cyano-steroids (**17**–**19**) exhibit remarkable anti-endocrine properties and display robust anti-inflammatory activity. These cyanosteroids not only possess potent anti-inflammatory effects but also demonstrate additional properties as aromatase inhibitors or estrogen antagonists. The high confidence level of over 93% further emphasizes their potential in addressing inflammatory conditions.

**Figure 5 biomedicines-11-02698-f005:**
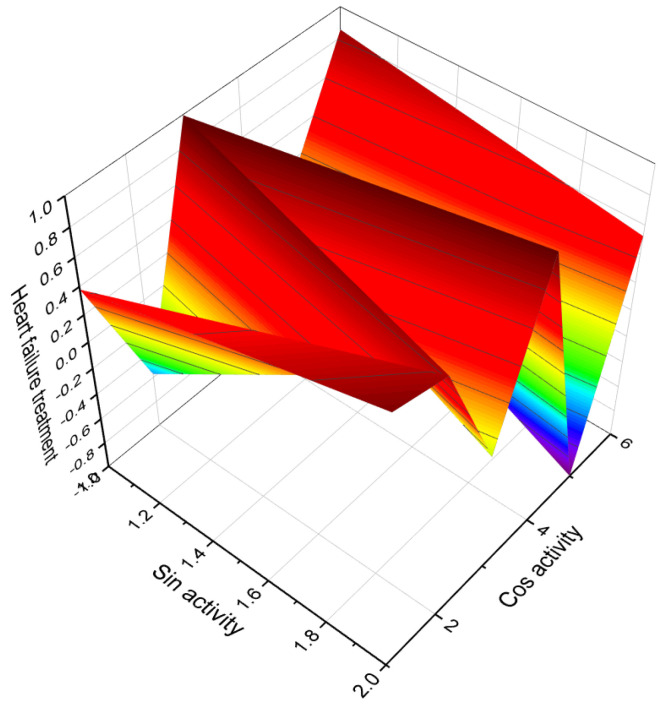
3D Graph illustrating the predicted and calculated activity of cyanosteroid (**23**) with over 97% confidence for the treatment of heart failure and related diseases. This synthetic cyanosteroid demonstrates an exceptional degree of activity, with over 97% confidence, making it a promising candidate for the treatment of heart failure and atherosclerosis. This compound exhibits a wide range of beneficial properties, acting as a potent cardiotonic agent, exerting antihypertensive effects, and displaying an anti-hyperaldosteronism effect. The combination of these positive qualities in a single drug represents a rare occurrence, further highlighting its potential significance in addressing heart-related conditions.

**Figure 6 biomedicines-11-02698-f006:**
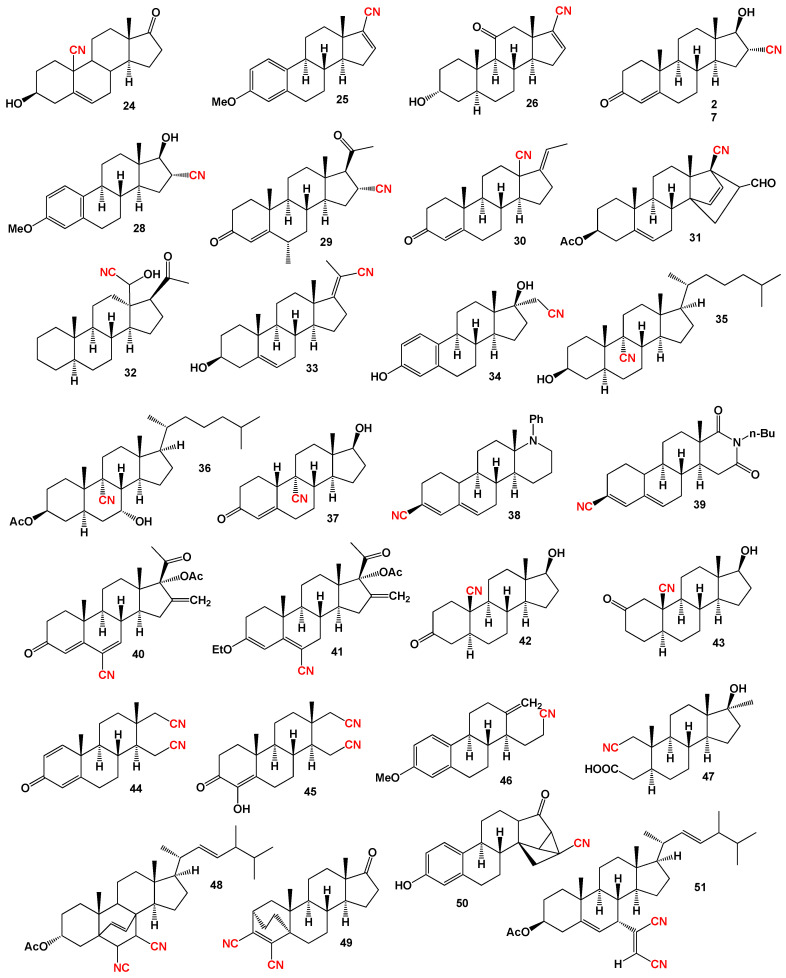
Bioactive steroids bearing nitrile group(s).

**Figure 7 biomedicines-11-02698-f007:**
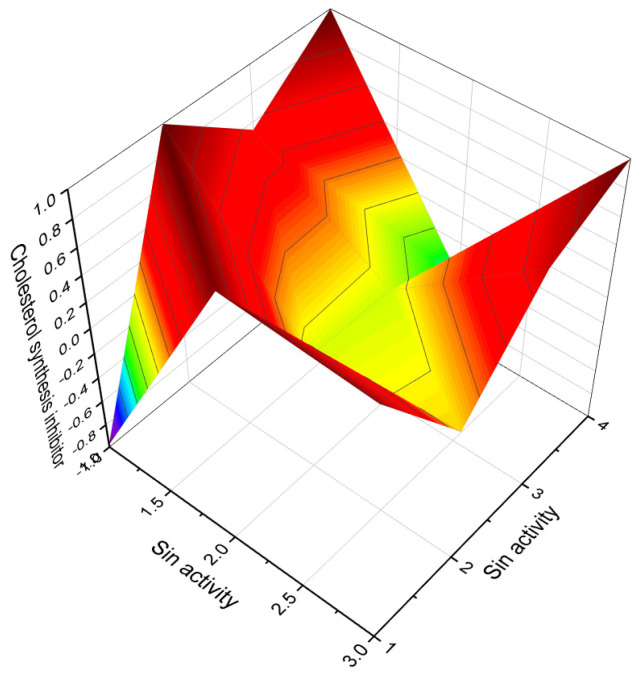
3D Graph depicting the predicted and calculated activity of cyanosteroids (**32**, **35**, and **36**) with over 90% confidence as inhibitors of cholesterol synthesis. Lovastatin and its related metabolites, such as simvastatin, pravastatin, fluvastatin, atorvastatin, and cerivastatin, are well-known inhibitors of cholesterol biosynthesis. These compounds have been isolated from various sources, including *Aspergillus terreus*, *Monascus* species (*M. ruber*, M. *purpureus*, *M. pilosus*, *M. vitreus*, *M. pubigerus*, and *M. anka*), *Paecilomyces viridis*, *Pleurotus ostreatus*, and *Pencillium citrinum*. Cyanosteroids (**32**, **35**, and **36**) exhibit robust activity as potent inhibitors of cholesterol biosynthesis. With their strong inhibitory effects, these compounds hold promise for potential clinical applications in medicine, although further investigation via preliminary trials is warranted.

**Figure 8 biomedicines-11-02698-f008:**
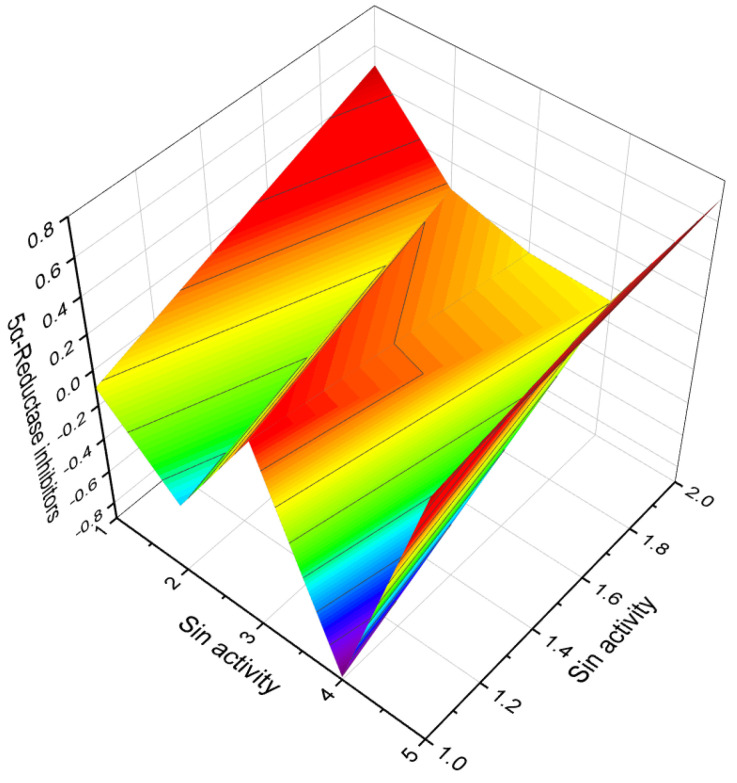
3D Graph illustrating the predicted and calculated activity of cyanosteroids (**38** and **39**) with over 91% confidence as 5α-reductase inhibitors. 5α-Reductase inhibitors are widely used in the treatment of benign prostatic hyperplasia. These inhibitors encompass diverse azasteroids. Cyanosteroids (**38** and **39**) also fall into this category of drugs while additionally featuring a nitrile group. The high confidence level, combined with their classification as 5α-reductase inhibitors, positions cyanosteroids (**38** and **39**) as potential candidates for therapeutic interventions in conditions related to this enzyme’s activity.

**Figure 9 biomedicines-11-02698-f009:**
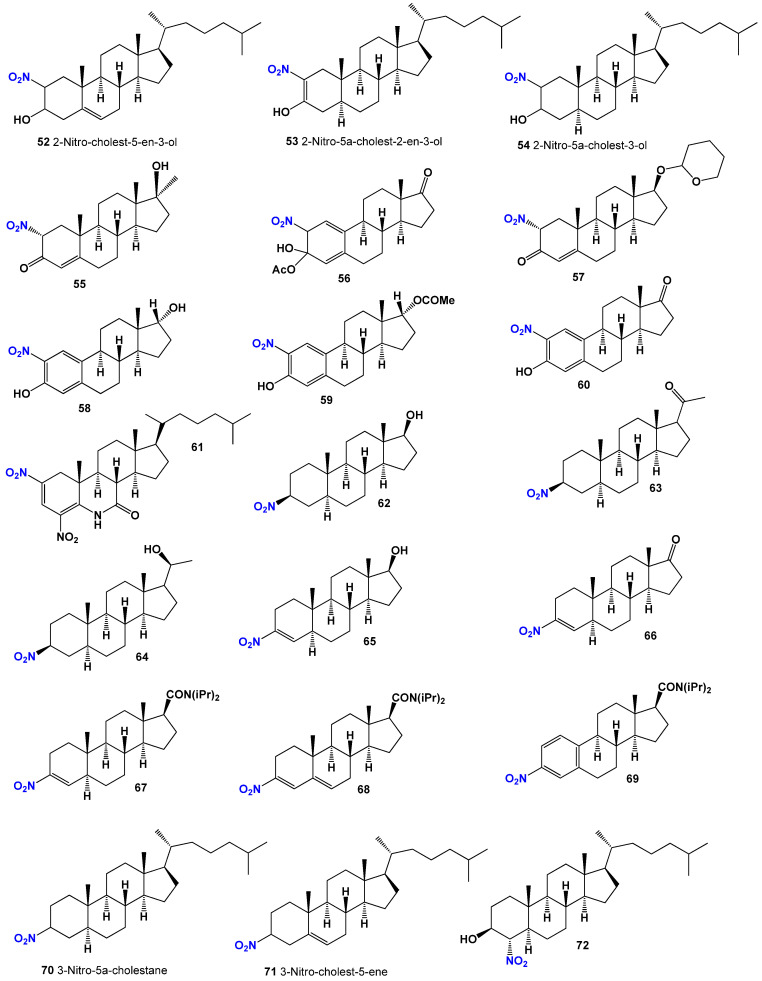
Bioactive steroids bearing nitro group.

**Figure 10 biomedicines-11-02698-f010:**
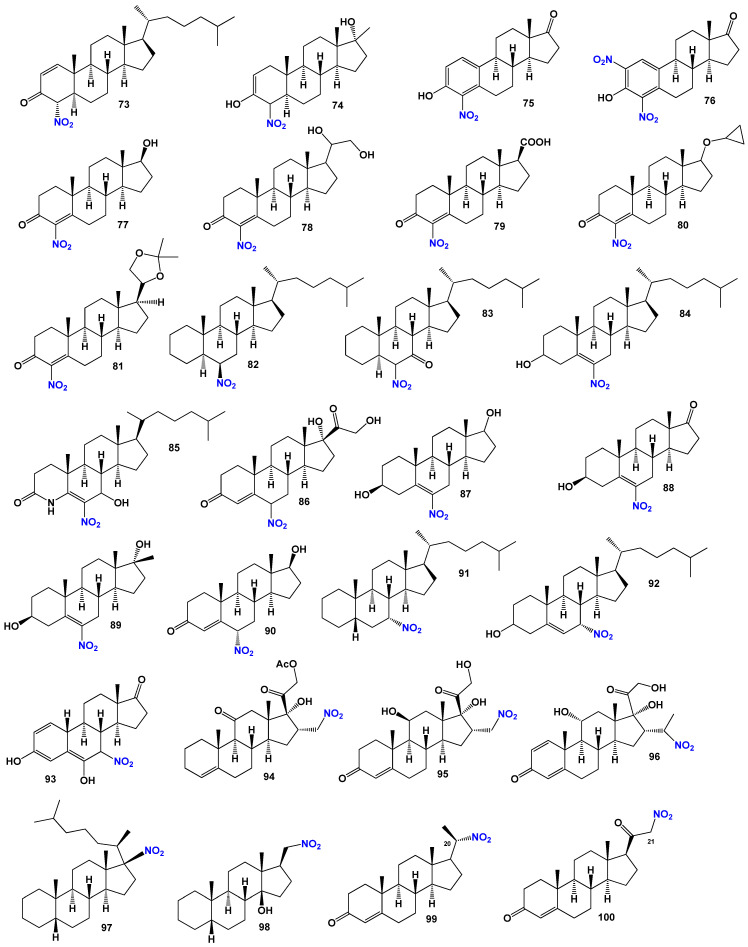
Bioactive steroids bearing nitro group(s).

**Figure 11 biomedicines-11-02698-f011:**
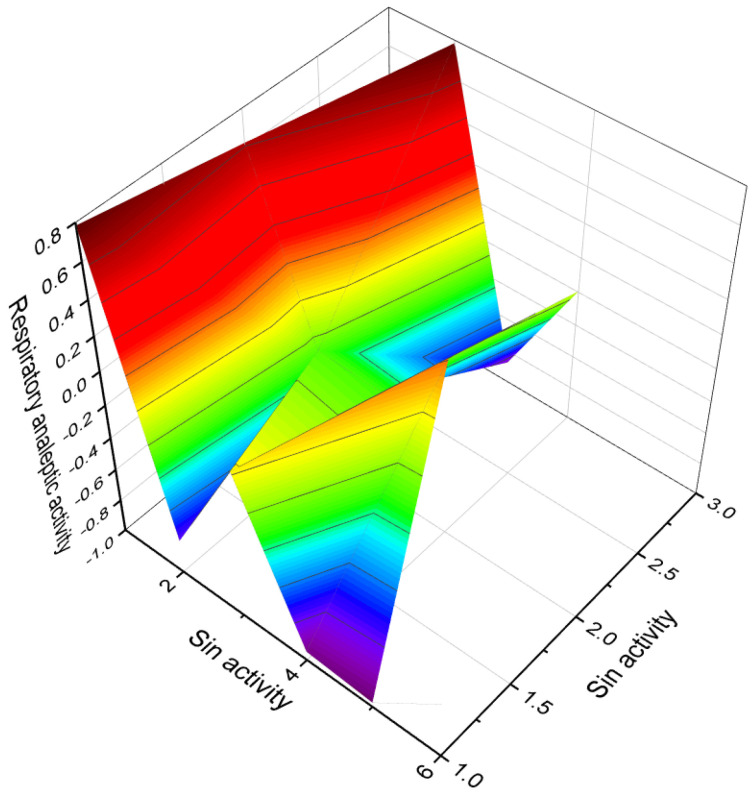
3D Graph illustrating the predicted and calculated activity of nitro-steroids (**52**, **53**, and **54**) with over 95% confidence as respiratory analeptics. The term analeptic typically refers to respiratory analeptics, which are central nervous system stimulants encompassing a wide range of drugs used for the treatment of conditions such as depression, attention deficit hyperactivity disorder, and respiratory depression. Nitrosteroids, in this case, serve as rare representatives demonstrating such properties.

**Figure 12 biomedicines-11-02698-f012:**
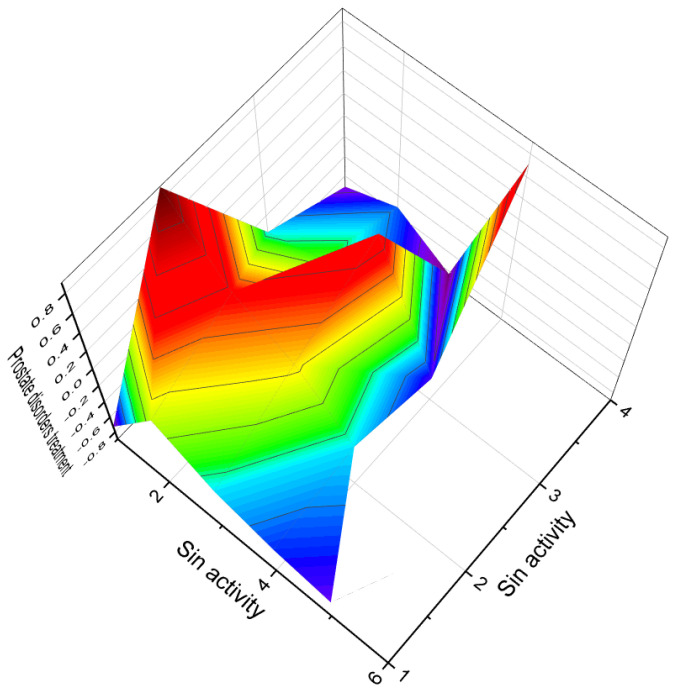
3D Graph presenting the predicted and calculated activity of nitro-steroids (**66**, **67**, **68**, and **69**) with over 92% confidence as treatments for prostate disease and prostatic (benign) hyperplasia. These nitro-steroids demonstrate potential therapeutic effects specifically targeted toward addressing prostate-related conditions and benign prostatic hyperplasia.

**Figure 15 biomedicines-11-02698-f015:**
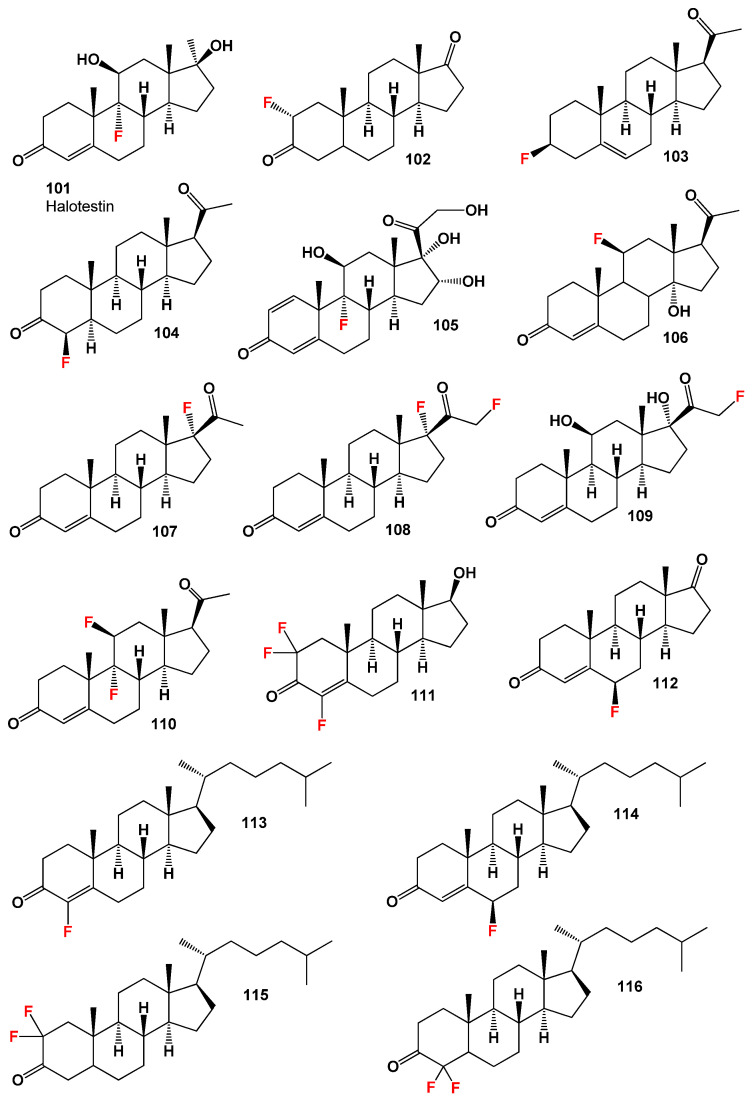
Bioactive steroids bearing fluorine atom(s).

**Figure 16 biomedicines-11-02698-f016:**
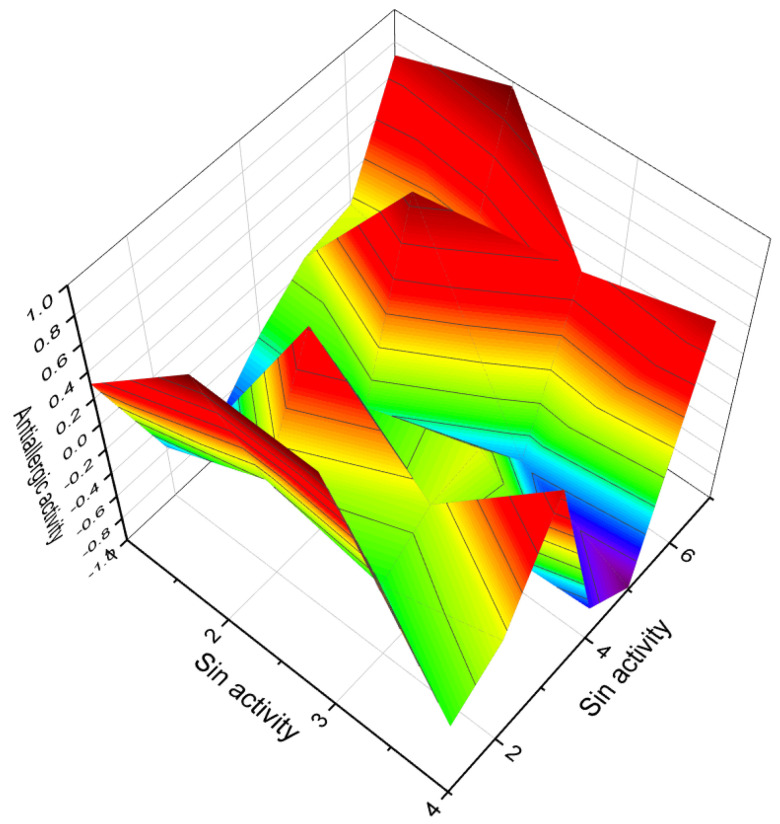
3D graph presenting the predicted and calculated antiallergic and anti-asthmatic activity of fluorinated steroids (**101**, **105**, **110**, and **112**) with over 96% confidence. It is noteworthy that these steroids demonstrate rare and beneficial properties as antiallergic and anti-asthmatic agents. This activity seems to be a dominant characteristic of steroids bearing fluorine atom(s), as multiple steroids within this group exhibit such activity. Further details and specific information can be found in Table 5.

**Figure 17 biomedicines-11-02698-f017:**
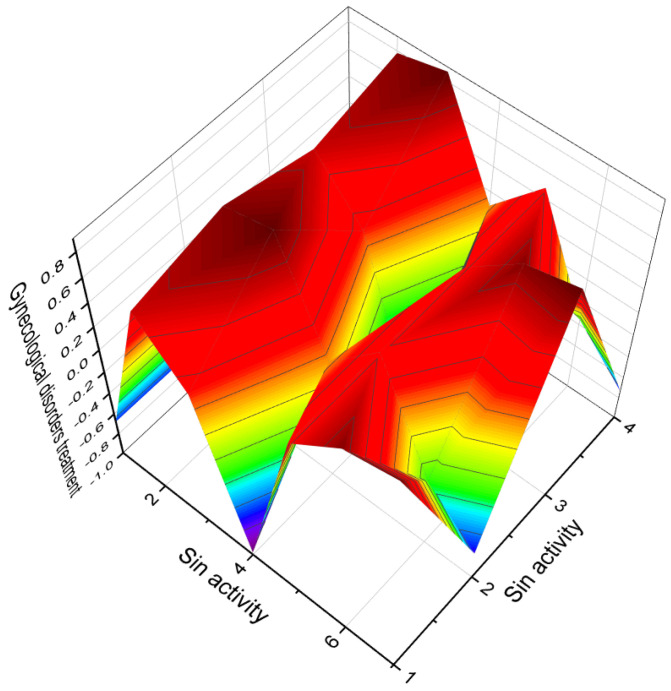
3D graph illustrating the predicted and calculated activities of steroids bearing fluorine atom(s) (**107** and **108**) with over 94% confidence for the treatment of gynecological diseases and menopausal disorders.

**Figure 18 biomedicines-11-02698-f018:**
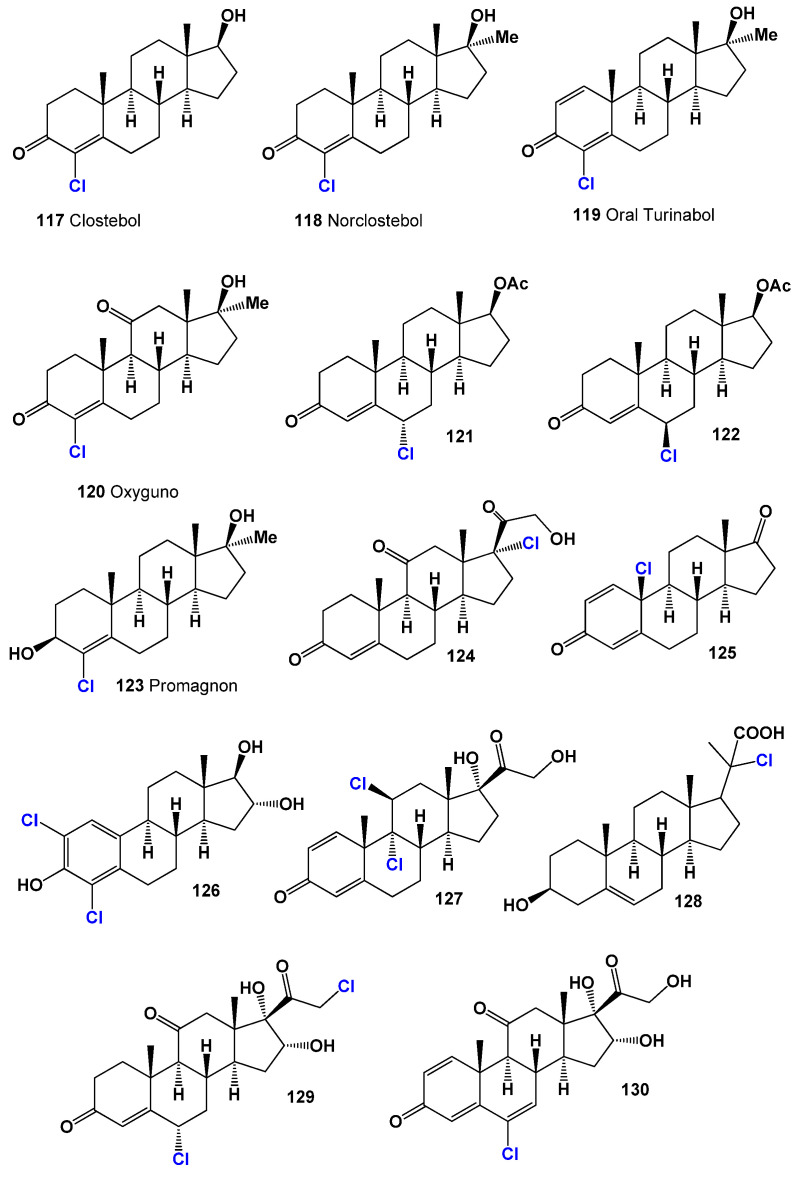
Bioactive steroids bearing chlorine atom(s).

**Figure 19 biomedicines-11-02698-f019:**
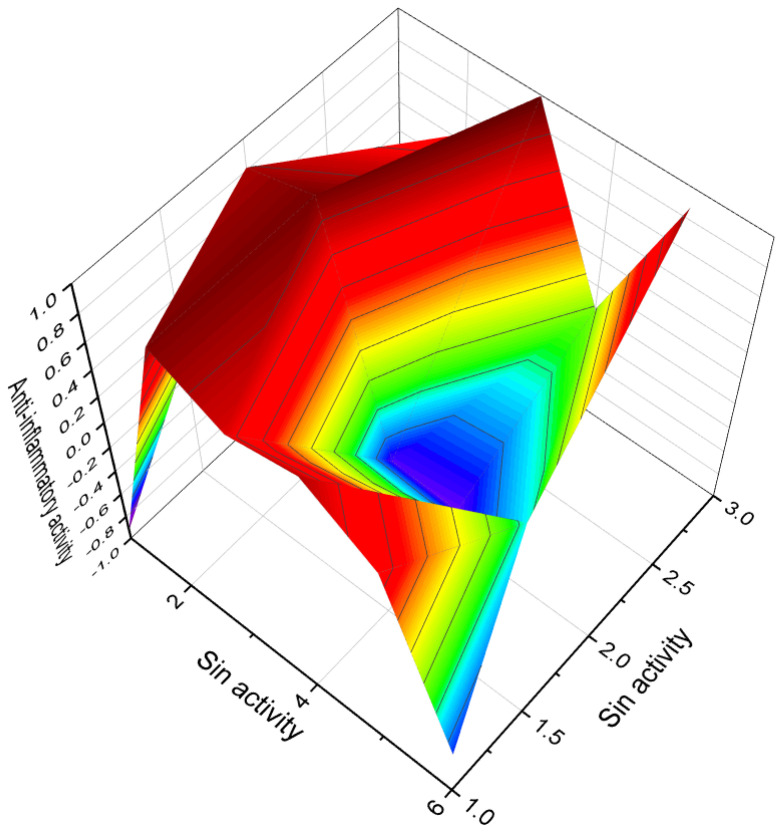
3D Graph depicting the predicted and calculated anti-inflammatory activity of steroids containing chlorine atom(s) (**127**, **129**, and **130**) with over 95% confidence. This demonstrates the diverse and sometimes unexpected characteristics of chlorinated steroids in terms of their biological activities.

**Figure 20 biomedicines-11-02698-f020:**
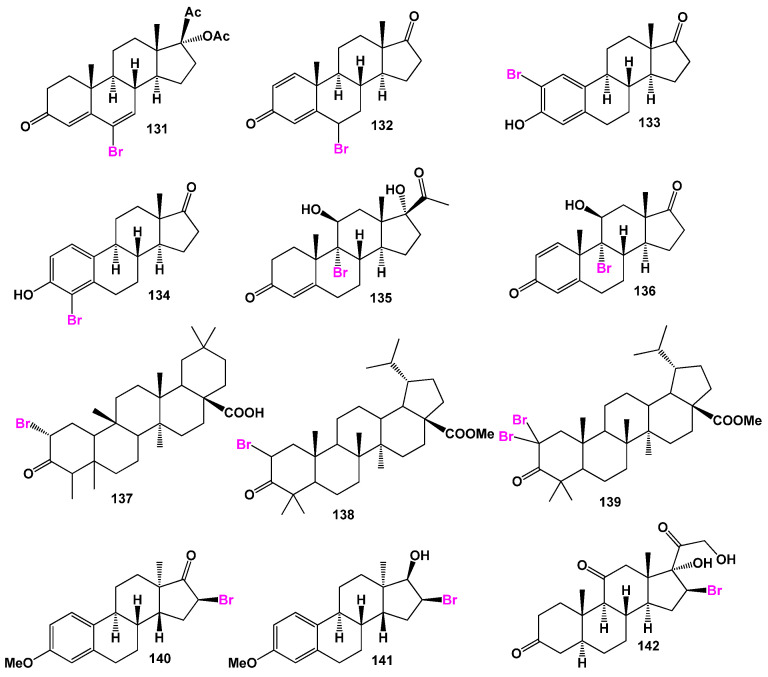
Bioactive steroids bearing bromine atom(s).

**Figure 21 biomedicines-11-02698-f021:**
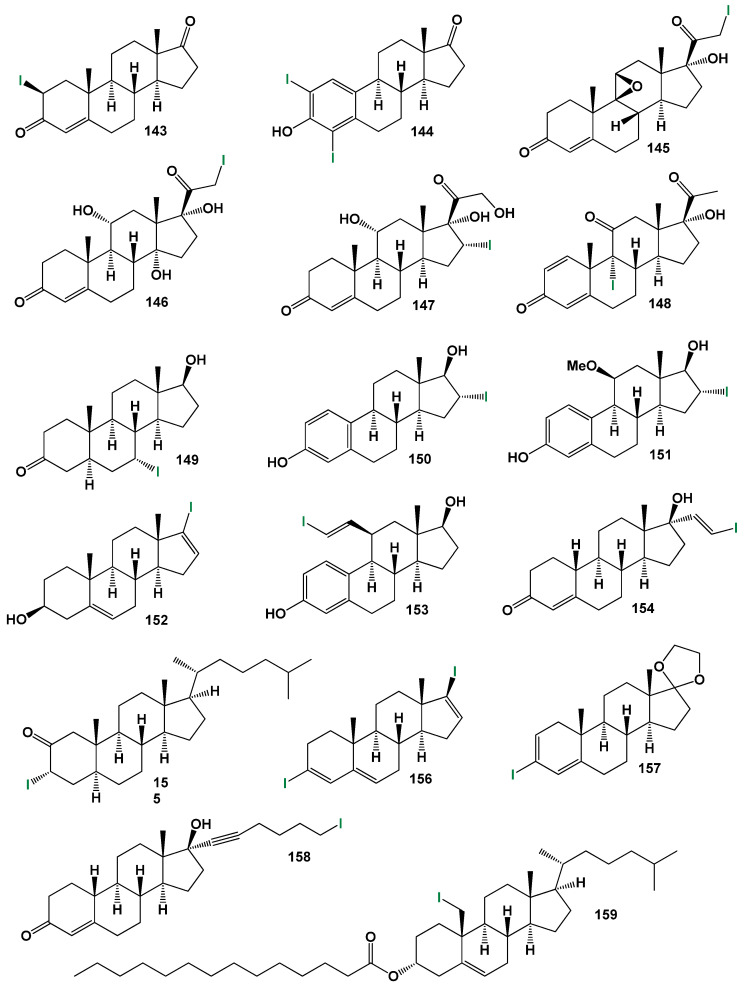
Bioactive steroids bearing iodine atom.

**Figure 22 biomedicines-11-02698-f022:**
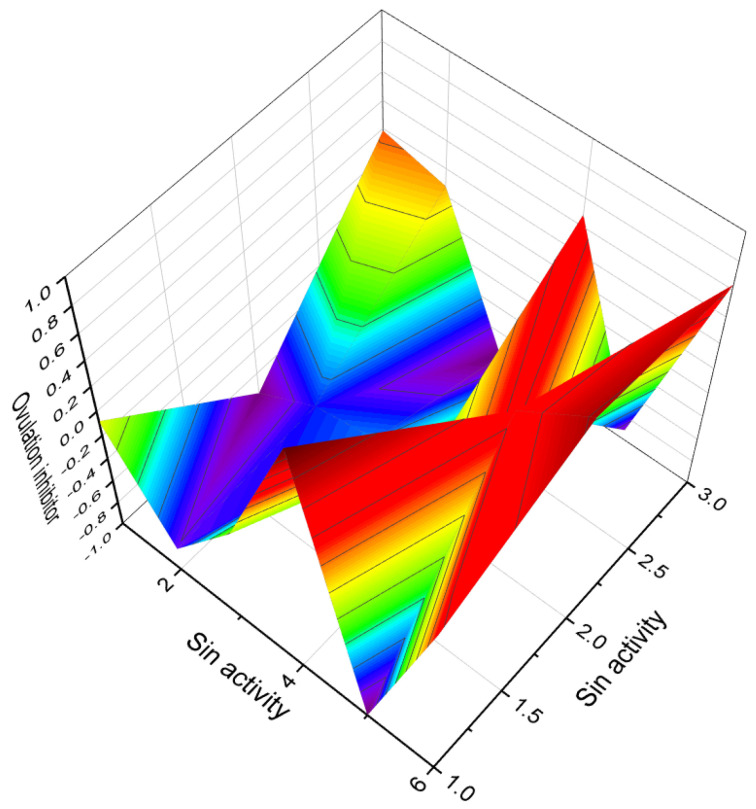
3D Graph showing the predicted and calculated activity of steroids bearing iodine atom (**154**, **158**, and **159**) with a high degree of confidence as ovulation inhibitors, contraceptives, and climacteric treatments. These iodinated steroids exhibit strong pharmacological properties that have significant potential for various applications in medical practice, particularly in the preservation of women’s health.

**Figure 23 biomedicines-11-02698-f023:**
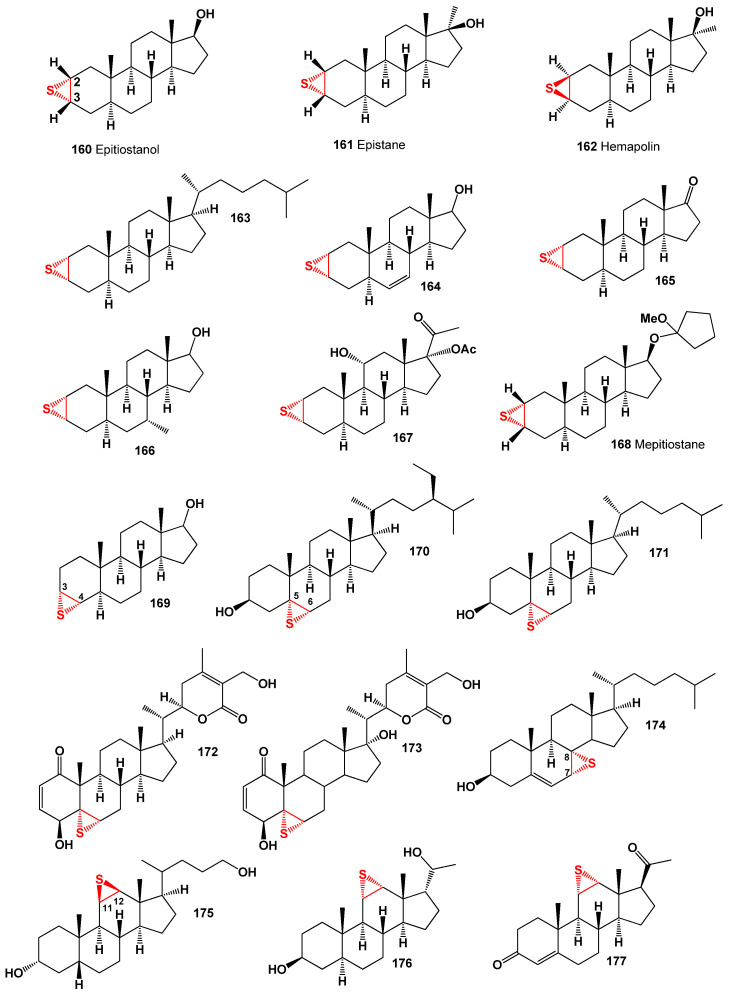
Bioactive steroids bearing the epithio group.

**Figure 24 biomedicines-11-02698-f024:**
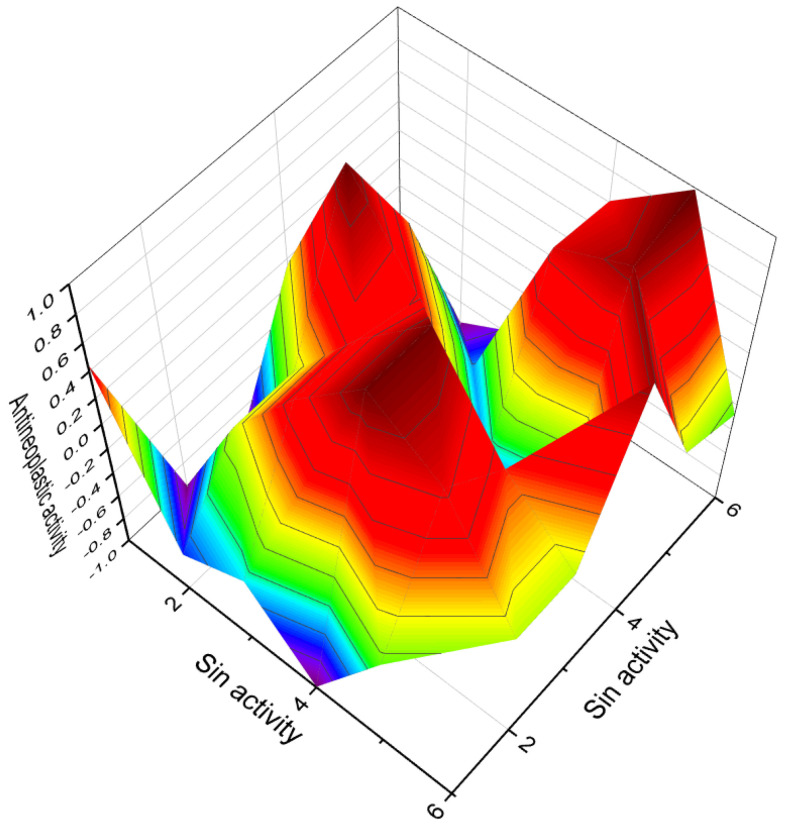
3D graph that depicts the predicted and calculated activity of epithio steroids. Specifically, the graph focuses on compounds numbered **160**, **161**, **162**, **164**, **165**, and **166**. These compounds have been studied for their potential therapeutic applications as antitumor, antitumor breast cancer, and demonstrated estrogen antagonist properties. Based on the graph, these epithio steroids exhibit a high level of confidence, with a maximum prediction accuracy of over 97%. They have shown promising activity in the areas of antitumor effects, specifically in the treatment of breast cancer. Additionally, these steroids have demonstrated properties as estrogen antagonists, suggesting their potential use in blocking the effects of estrogen. The graph provides a visual representation of the relationship between the predicted activity and the calculated activity of these steroids. It allows researchers and scientists to analyze and compare the efficacy of these compounds in different therapeutic areas. This information is valuable in understanding the potential of epithio steroids for their antitumor and estrogen antagonist properties. It can guide further research and development in the field, aiding in the discovery of new treatments and potential drug candidates.

**Figure 25 biomedicines-11-02698-f025:**
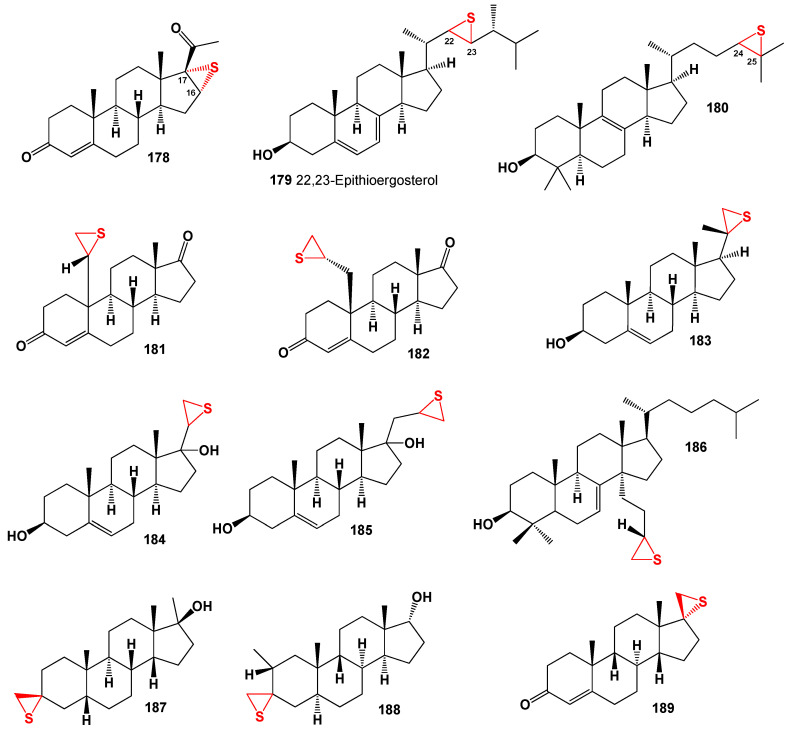
Steroids bearing the epithio group.

**Figure 26 biomedicines-11-02698-f026:**
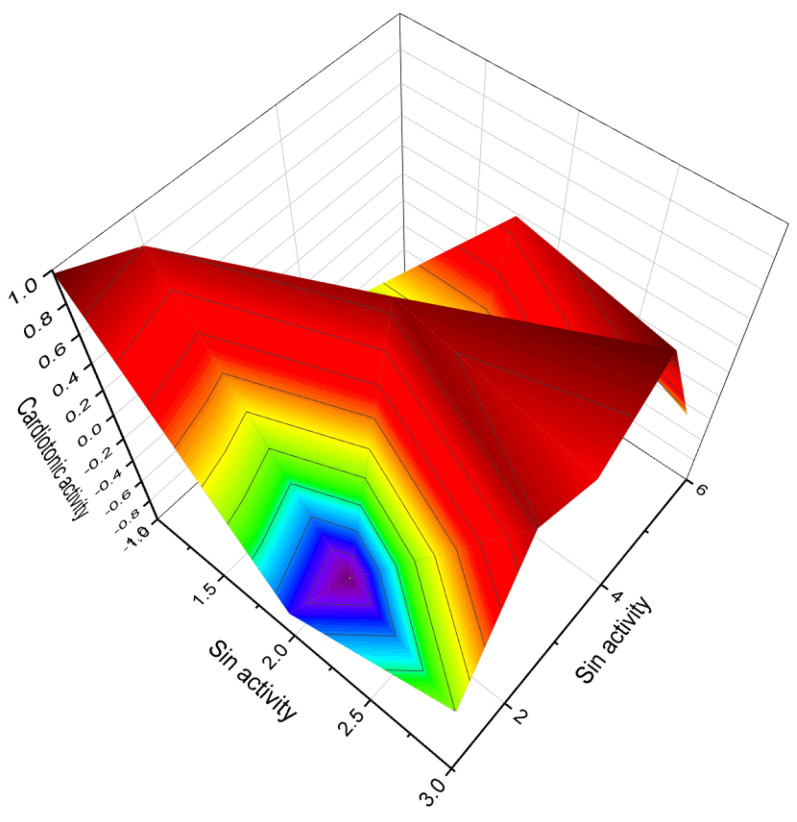
3D graph that illustrates the predicted and calculated activity of epithio steroids. Specifically, the graph focuses on compounds numbered **169**, **176**, and **178**. These compounds have been studied for their potential therapeutic applications. The graph shows that these epithio steroids exhibit a high level of confidence, with a maximum prediction accuracy of over 94%. These compounds possess rare properties as both cardiotonic and antiarrhythmic agents. Cardiotonic drugs are used to enhance the performance and improve the contraction of the heart muscles. This leads to improved blood flow to all body tissues, thereby supporting cardiovascular health. On the other hand, antiarrhythmic drugs are medications that help prevent and treat abnormal or irregular heart rhythms. They help restore and maintain a normal rhythm, ensuring the proper functioning of the heart. The fact that these epithio steroids exhibit both cardiotonic and antiarrhythmic properties is noteworthy. It suggests their potential use as dual-function agents for the treatment of cardiovascular conditions.

**Figure 27 biomedicines-11-02698-f027:**
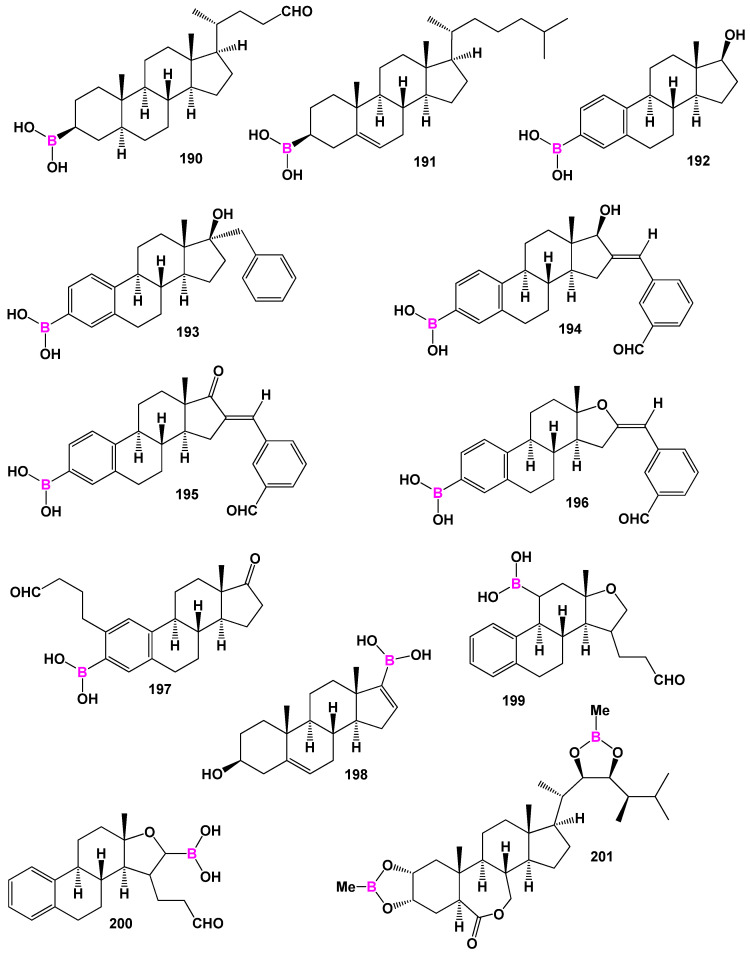
Steroids bearing boron atom(s).

**Figure 28 biomedicines-11-02698-f028:**
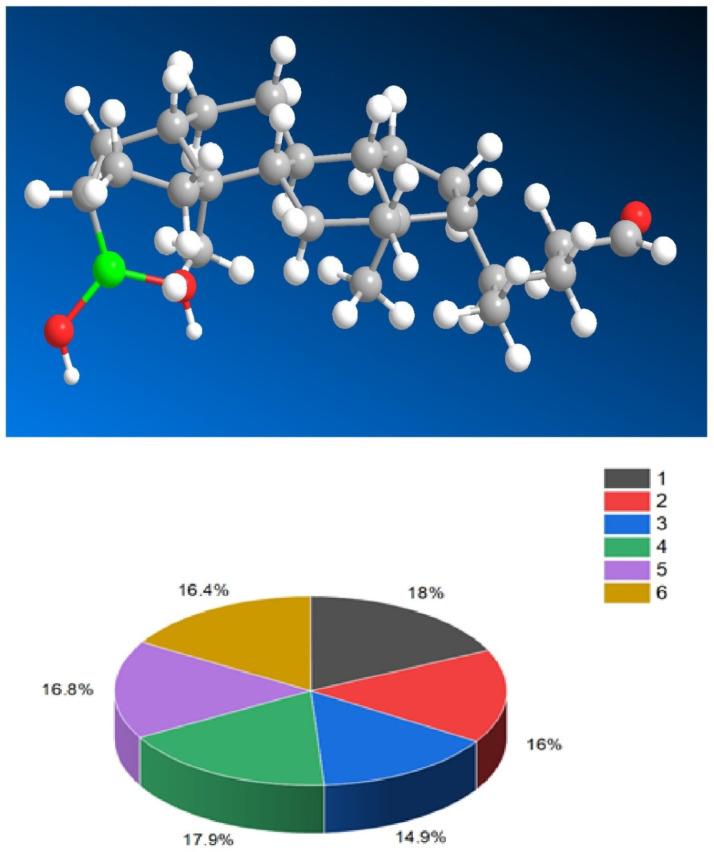
3D model and percentage distribution of the dominant skin diseases activity on the example of steroid-bearing boron atom (**190**), which has a wide range of anticancer properties. Where activities are indicated under the numbers: 1. *Anti-eczematic* (18%), 2. *Dermatologic* (16%). 3, *Anti-psoriatic* (14.9%), 4. *Antihypertensive* (17.9%), 5. *Myocardial ischemia treatment* (16.8%), and 6. *Antineoplastic* (16.4%). The boron atom is highlighted in green.

**Figure 29 biomedicines-11-02698-f029:**
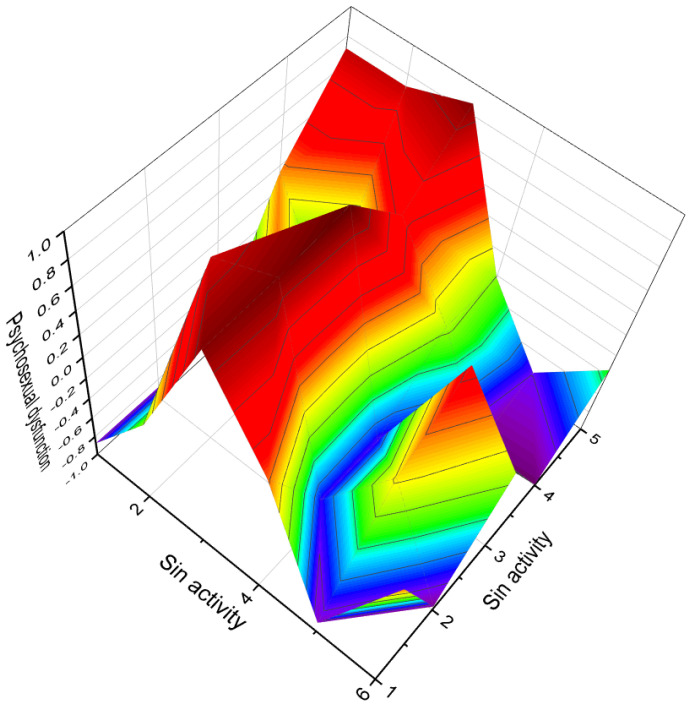
3D graph illustrates the predicted and calculated activity of steroids bearing a boron atom (**192**–**197**). These compounds exhibit a high level of confidence, with a maximum prediction accuracy of over 91%. Notably, these boron-containing steroids demonstrate a unique and rare ability to treat psychosexual dysfunction associated with the treatment of gynecological diseases in women. Psychosexual dysfunction refers to difficulties or disruptions in sexual functioning that may have psychological or emotional origins. The inclusion of boron in these steroids offers a novel approach to addressing psychosexual dysfunction, highlighting their potential in the treatment of gynecological conditions while simultaneously addressing associated sexual issues. This distinctive property sets them apart from other classes of steroids and underscores their significance in providing comprehensive care for women’s health. The 3D graph visually represents the relationship between the predicted activity and the calculated activity of these boron-containing steroids. It offers valuable insights into their potential efficacy in treating psychosexual dysfunction, providing a foundation for further research and clinical exploration in this area. The availability of the 3D plot and the percentage distribution of dominant activity for specific borosteroids offer valuable insights into their efficacy and potential applications. This information guides researchers in exploring the therapeutic potential of these compounds and designing more targeted and effective treatments for various diseases, particularly those related to skin disorders and beyond.

**Figure 30 biomedicines-11-02698-f030:**
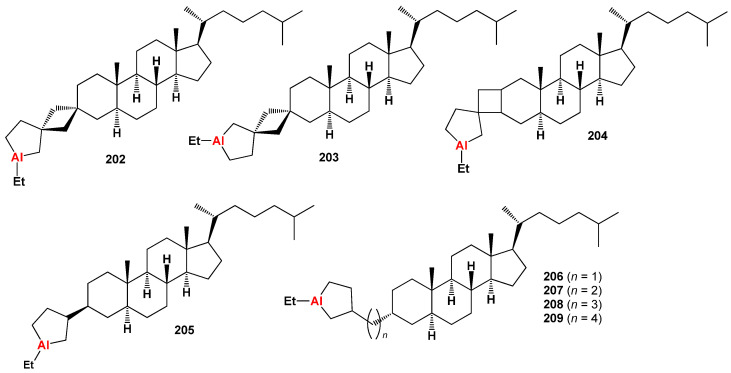
Steroids bearing an aluminum atom.

**Figure 31 biomedicines-11-02698-f031:**
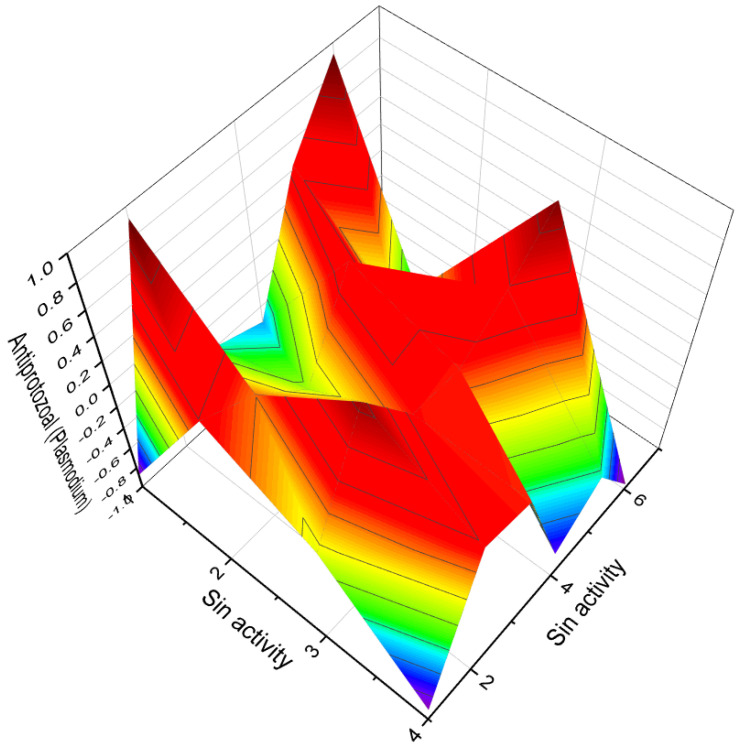
3D graph illustrates the predicted and calculated antiprotozoal activity, specifically against *Plasmodium* sp., of aluminum-containing steroids. The graph focuses on compounds numbered **202**, **203**, **208**, and **209**. These compounds have been investigated for their potential as antiprotozoal agents. The graph demonstrates that these aluminum-containing steroids exhibit a high level of confidence, with a maximum prediction accuracy of over 92%. This indicates a strong likelihood that these compounds possess antiprotozoal activity against *Plasmodium* sp., the protozoan responsible for causing malaria. What makes these aluminum-containing steroids particularly interesting and useful is that they exhibit antiprotozoal activity without the presence of a peroxide group, which is commonly observed in steroids with similar activity. This suggests that the aluminum atom, in combination with the steroid structure, plays a crucial role in their antiprotozoal properties. The 3D graph visually represents the relationship between the predicted activity and the calculated activity of these aluminum-containing steroids. It provides insights into the efficacy of these compounds as antiprotozoal agents against *Plasmodium* sp.

**Figure 32 biomedicines-11-02698-f032:**
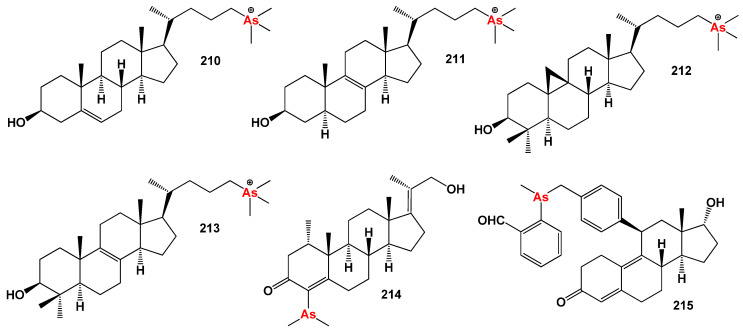
Steroids bearing arsenic atoms or arsenosteroids.

**Figure 33 biomedicines-11-02698-f033:**
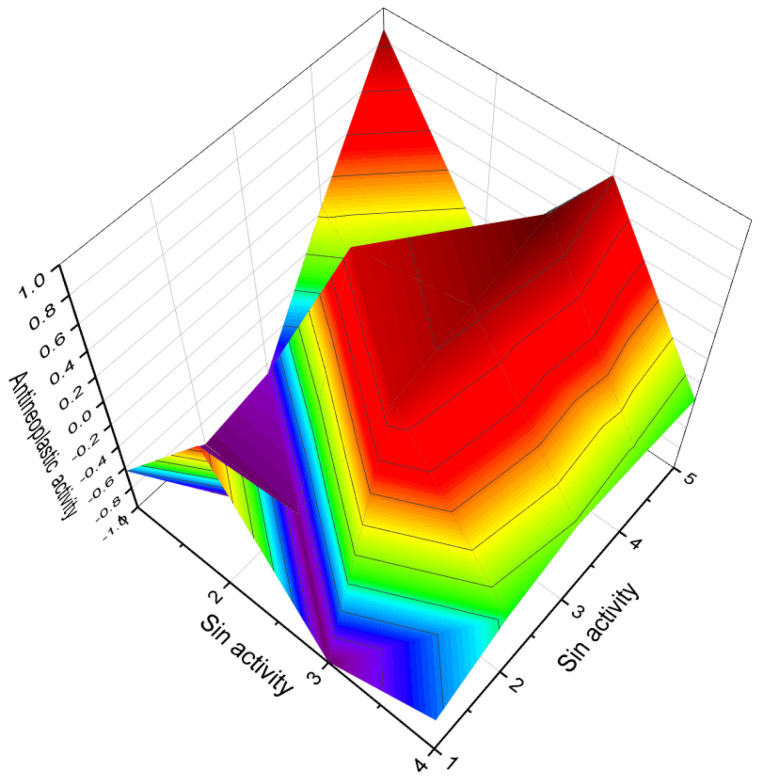
3D graph illustrates the predicted and calculated antineoplastic activity of steroids bearing arsenic atoms (**210**–**213**). These compounds exhibit a high level of confidence, with a maximum prediction accuracy of over 98%.

**Figure 34 biomedicines-11-02698-f034:**
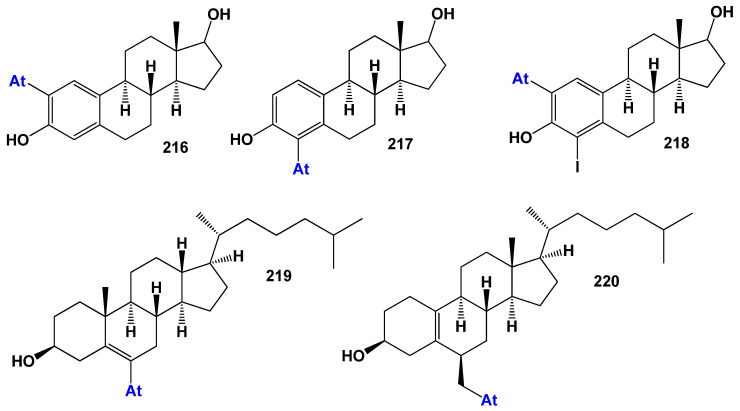
Steroids bearing astatine atoms or astatosteroids.

**Figure 35 biomedicines-11-02698-f035:**
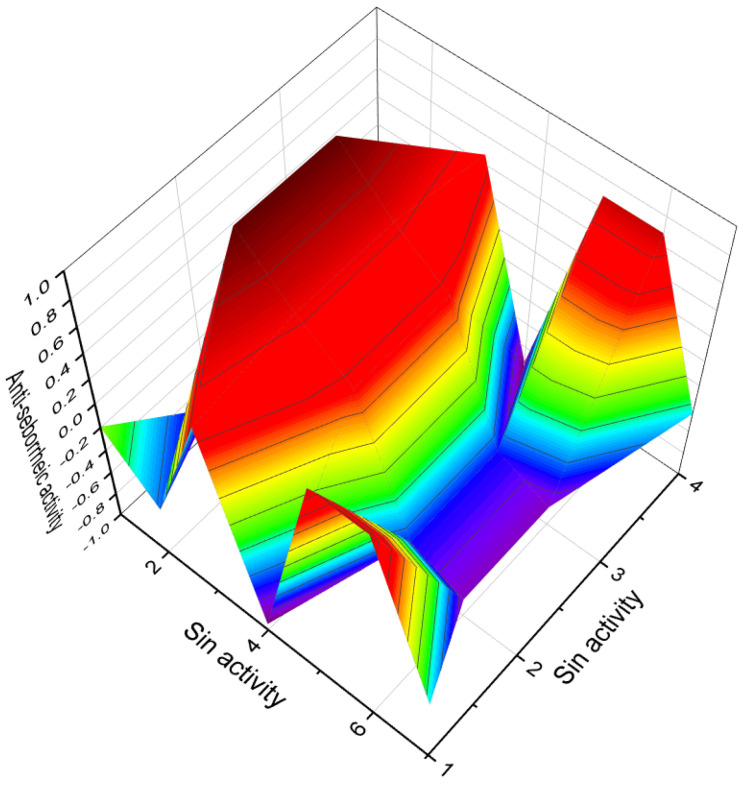
3D graph represents the predicted and calculated anti-seborrheic and antifungal activity of steroids bearing an astatine atom (**216** and **217**). The graph demonstrates a high level of confidence, with a maximum prediction accuracy of over 93%. The focus of this graph is specifically on the anti-seborrheic and antifungal properties of astatine-containing steroids. Anti-seborrheic agents are substances that help control seborrheic dermatitis, a common skin condition characterized by red, itchy, and flaky skin. Antifungal agents, on the other hand, combat fungal infections caused by various fungi. By visualizing the predicted and calculated activity in this 3D graph, valuable insights into the potential efficacy of these steroids as anti-seborrheic and antifungal agents can be obtained. This information serves as a foundation for further research and development in the field of dermatology and related medical disciplines. It is worth noting that the exploration of astatosteroids and their biological activities is an ongoing area of investigation. Further studies are required to elucidate their mechanisms of action, evaluate their stability, and assess their potential for therapeutic applications in treating seborrheic dermatitis, fungal infections, and other related conditions.

**Figure 36 biomedicines-11-02698-f036:**
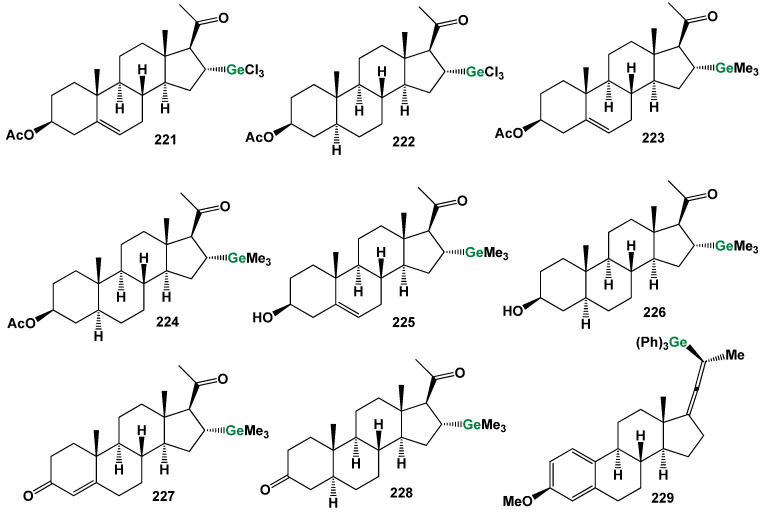
Steroids bearing germanium atoms or germinated steroids.

**Figure 37 biomedicines-11-02698-f037:**
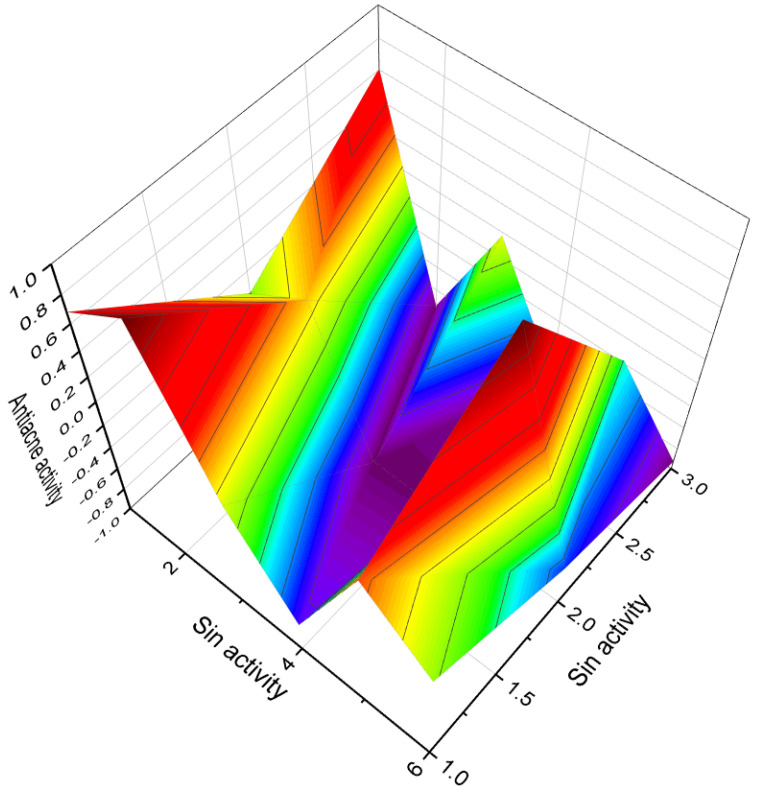
3D graph represents the predicted and calculated antiacne and dermatologic activity of steroids bearing germanium atom (**227**, **228**, and **229**). The graph demonstrates a high level of confidence, with a maximum prediction accuracy of over 98%. The 3D plot presented in Figure 37 provides a visual representation of the potential activity of steroids bearing a germanium atom (**227**–**229**), offering valuable insights into their predicted efficacy. These findings inspire further investigation and potential development of novel therapeutic approaches based on germinated steroids.

**Figure 38 biomedicines-11-02698-f038:**
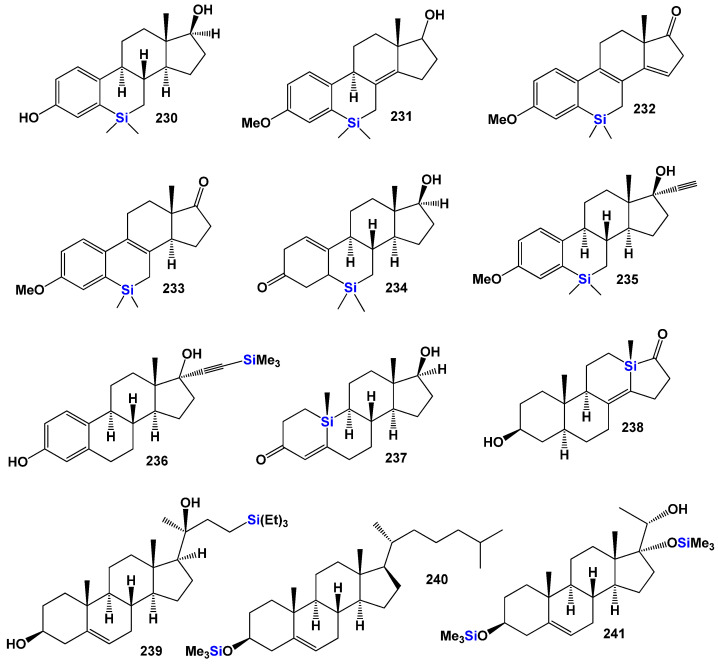
Steroids bearing silicon atom(s) or silasteroids.

**Figure 39 biomedicines-11-02698-f039:**
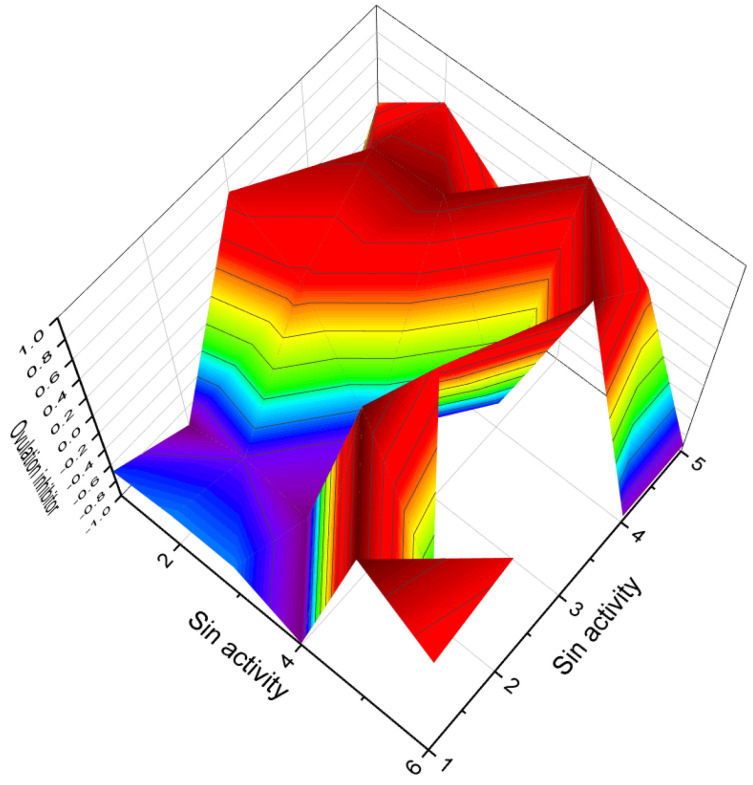
3D graph represents the predicted and calculated activity of steroids bearing silicon atoms (**230**, **231**, **232**, **236**, and **237**) with a maximum of over 99% confidence as ovulation inhibitors, accompanied by contraceptive and antitumor properties. The 3D graph presented in Figure 39 offers a visual representation of the potential activities of steroids bearing a silicon atom. It serves as a valuable tool for researchers in assessing the biological potential of these compounds and guiding future investigations.

**Figure 40 biomedicines-11-02698-f040:**
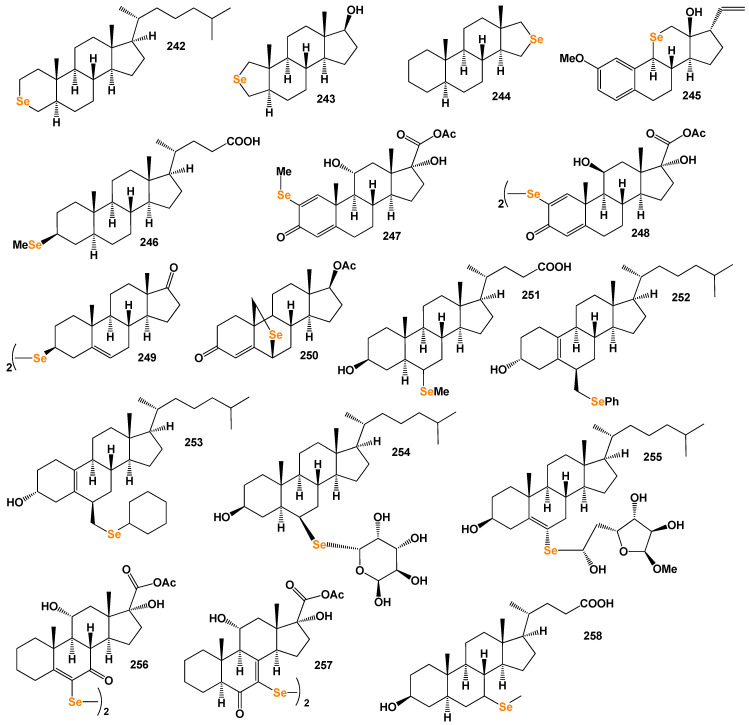
Bioactive Steroids bearing selenium atoms or selenasteroids.

**Figure 41 biomedicines-11-02698-f041:**
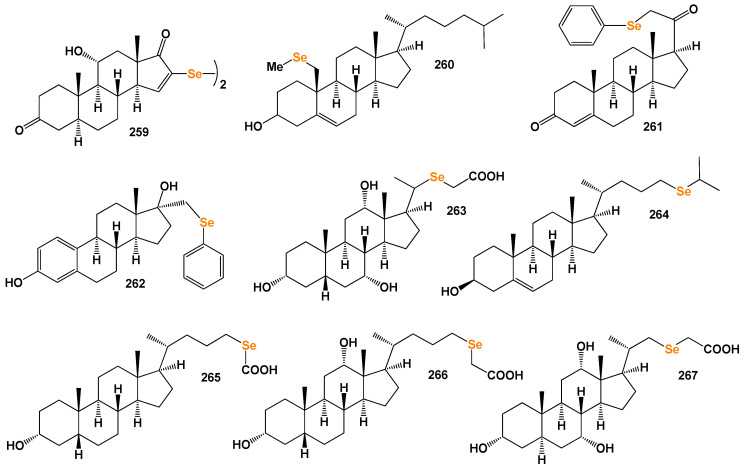
Steroids bearing selenium atoms or selenasteroids.

**Figure 42 biomedicines-11-02698-f042:**
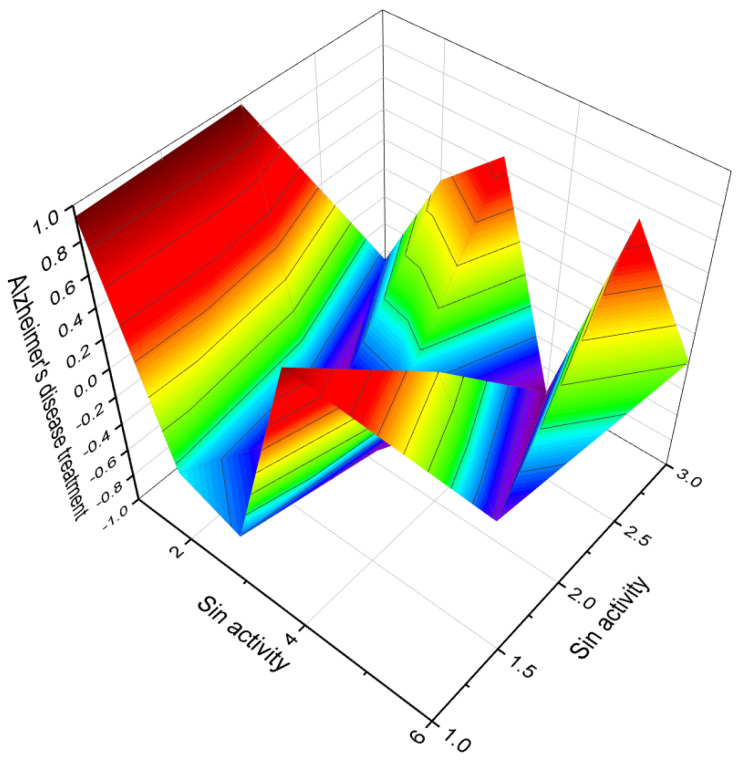
3D graph illustrating the predicted and calculated activity of steroids bearing a selenium atom (**244**, **245**, and **250**) with a maximum confidence level exceeding 93%. These compounds exhibit significant potential for the treatment of Alzheimer’s disease. The graph provides a visual representation of the activity profiles of these selenium-containing steroids, offering valuable insights into their efficacy and potential therapeutic benefits. The high confidence level indicates the robustness of the predictions and underscores the promise of these compounds in addressing Alzheimer’s disease. Further research and validation studies are necessary to fully evaluate the mechanisms of action and optimize the properties of these steroids for the treatment of Alzheimer’s disease. However, the 3D graph serves as an informative tool, aiding researchers in assessing the potential of these compounds and guiding future investigations. The study of selenium-containing steroids for Alzheimer’s disease presents an exciting opportunity for the development of novel therapeutic strategies. Continued research in this area holds promise for advancing our understanding of the disease and potentially leading to the discovery of effective treatments.

**Figure 43 biomedicines-11-02698-f043:**
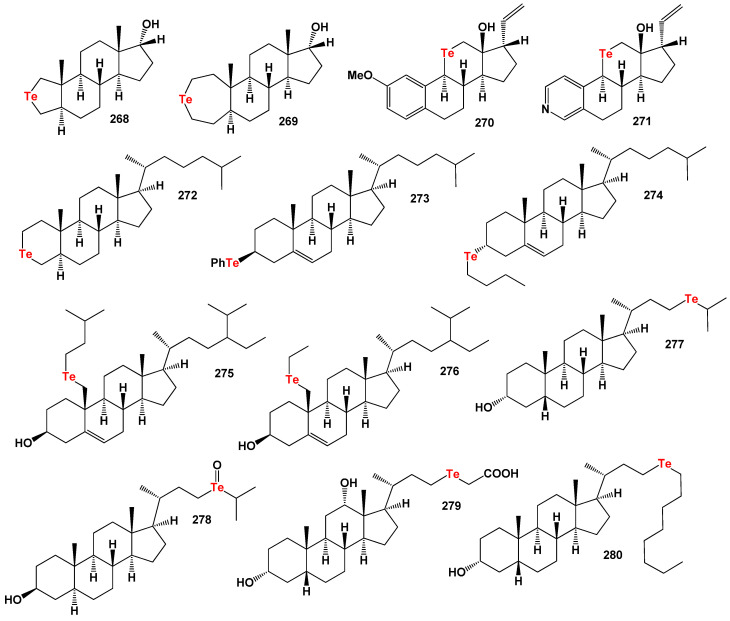
Steroids bearing tellurium atoms or tellurasteroids.

**Figure 44 biomedicines-11-02698-f044:**
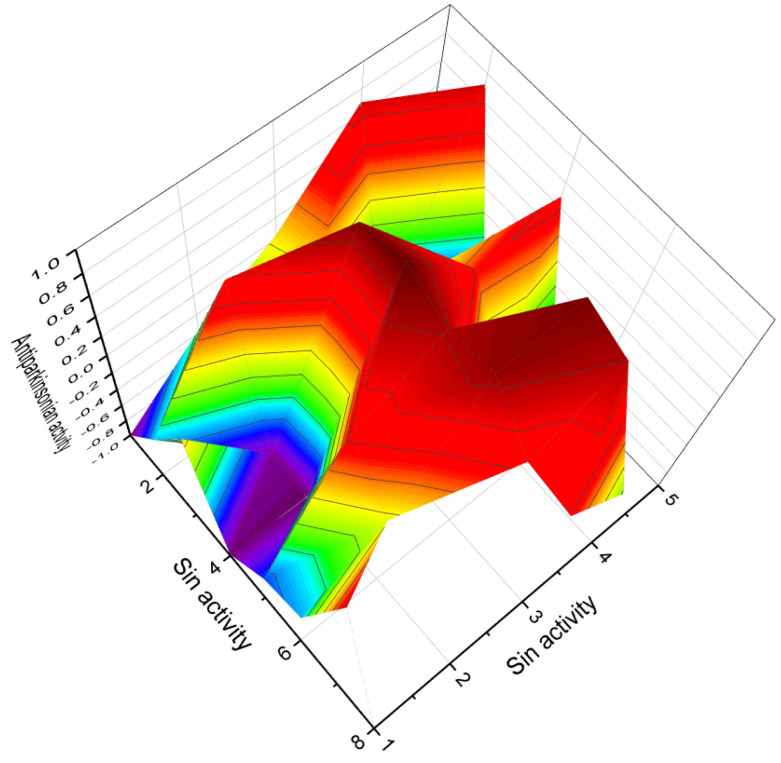
3D graph illustrating the predicted and calculated activity of steroids bearing tellurium atom (**271**, **272**, **274**, **275**, and **294**) with a maximum of over 96% confidence, which can be used to treat Alzheimer’s, Parkinson’s, and other neurodegenerative diseases. The investigation of tellura steroids showcases their diverse range of biological activities and potential applications in various fields, including anti-inflammatory, antioxidant, and anticancer therapies. Further research is warranted to elucidate the underlying mechanisms of action and optimize the properties of these compounds for potential clinical use. It is important to note that the specific structure and activity of each compound may vary, and further studies are needed to fully understand their potential therapeutic applications. The presented data serve as a starting point for the exploration of tellura steroids and their potential roles in addressing various diseases and health conditions.

**Figure 45 biomedicines-11-02698-f045:**
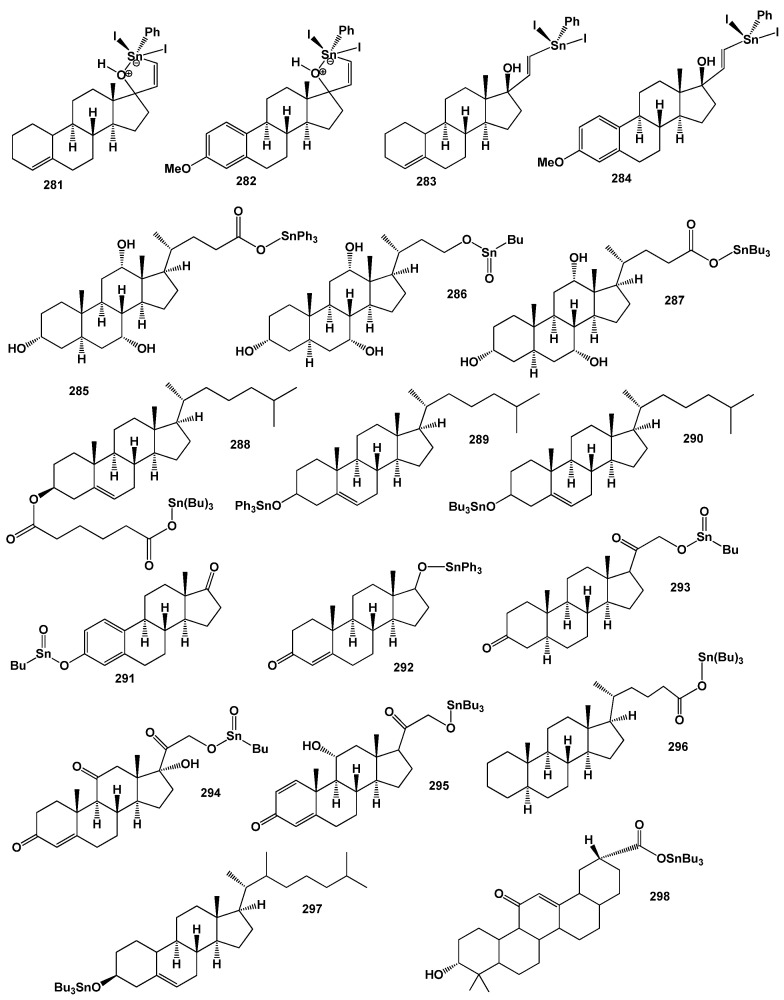
Steroids bearing tin atom or organotin steroids.

**Figure 46 biomedicines-11-02698-f046:**
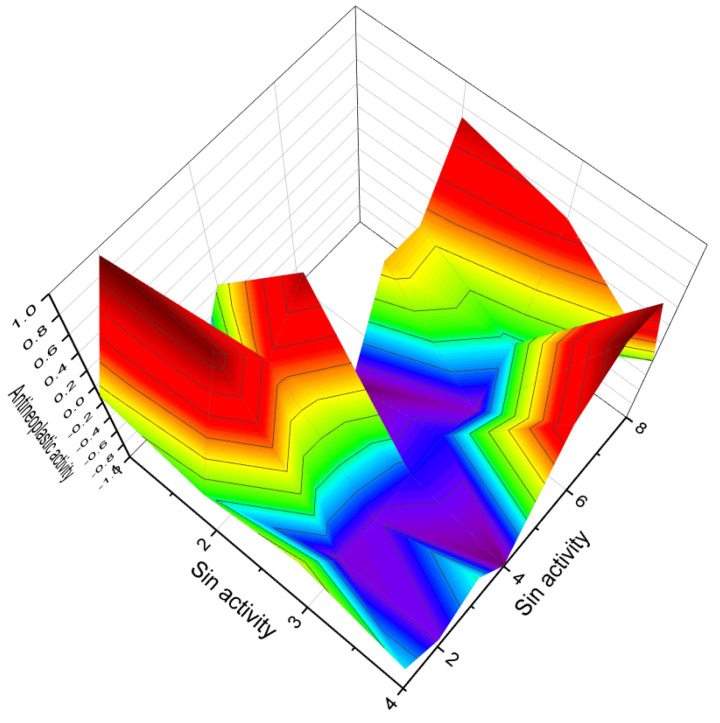
3D graph showing the predicted and calculated activity of steroids bearing tin atom (**282**, **283**, **284**, and **289**) with a maximum of over 99% confidence, demonstrating antitumor activity against pancreatic cancer, prostatic hyperplasia, prostate cancer, and generating anti-metastatic effect. De facto, but all steroids bearing tin atoms show antitumor activity as the dominant effect.

**Figure 47 biomedicines-11-02698-f047:**
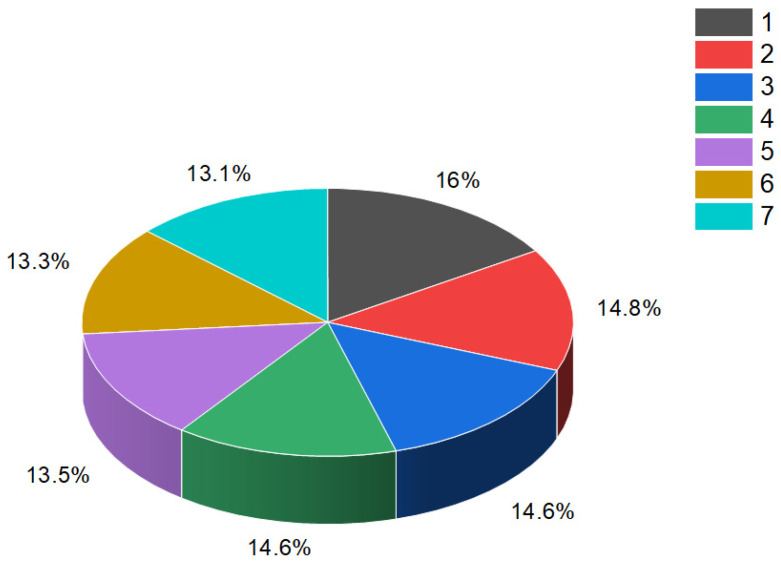
Percentage distribution of the dominant antitumor activity, for example, organotin steroid (**292**), which possesses a broad range of anticancer properties. The activities are indicated by the following numbers: 1. *Antineoplastic* (16%): This activity refers to the compound’s ability to inhibit the growth of various types of tumors. 2. *Anti-breast cancer* (14.8%): This activity specifically targets breast cancer cells, indicating its potential as a therapeutic agent against this malignancy. 3. *Prostatic (benign) hyperplasia treatment* (14.6%): This activity focuses on the treatment of benign prostatic hyperplasia, a non-cancerous enlargement of the prostate gland. 4. *Anti-sarcoma cancer* (14.6%): This activity highlights the compound’s effectiveness against sarcoma, a cancerous tumor arising from connective tissues. 5. *Anti-renal cancer* (13.5%): This activity indicates the compound’s potential in combating renal cancer, which originates in the kidneys. 6. *Prostate cancer treatment* (13.3%): This activity specifically targets prostate cancer cells, suggesting its potential as a therapeutic option for this type of cancer. 7. *Anti-pancreatic cancer* (13.1%): This activity focuses on the compound’s efficacy against pancreatic cancer, a challenging malignancy with limited treatment options. The percentage distribution of these dominant antitumor activities provides insights into the compound’s efficacy against various types of cancer. These findings suggest that organotin steroid (**292**) holds promise as a multifunctional anticancer agent capable of targeting different tumor types. Understanding the specific activities and distribution of anticancer properties is crucial in evaluating the potential of organotin steroids as therapeutic agents. Further research and development in this area may uncover opportunities for the design of novel treatments and therapeutic strategies against cancer.

**Figure 48 biomedicines-11-02698-f048:**
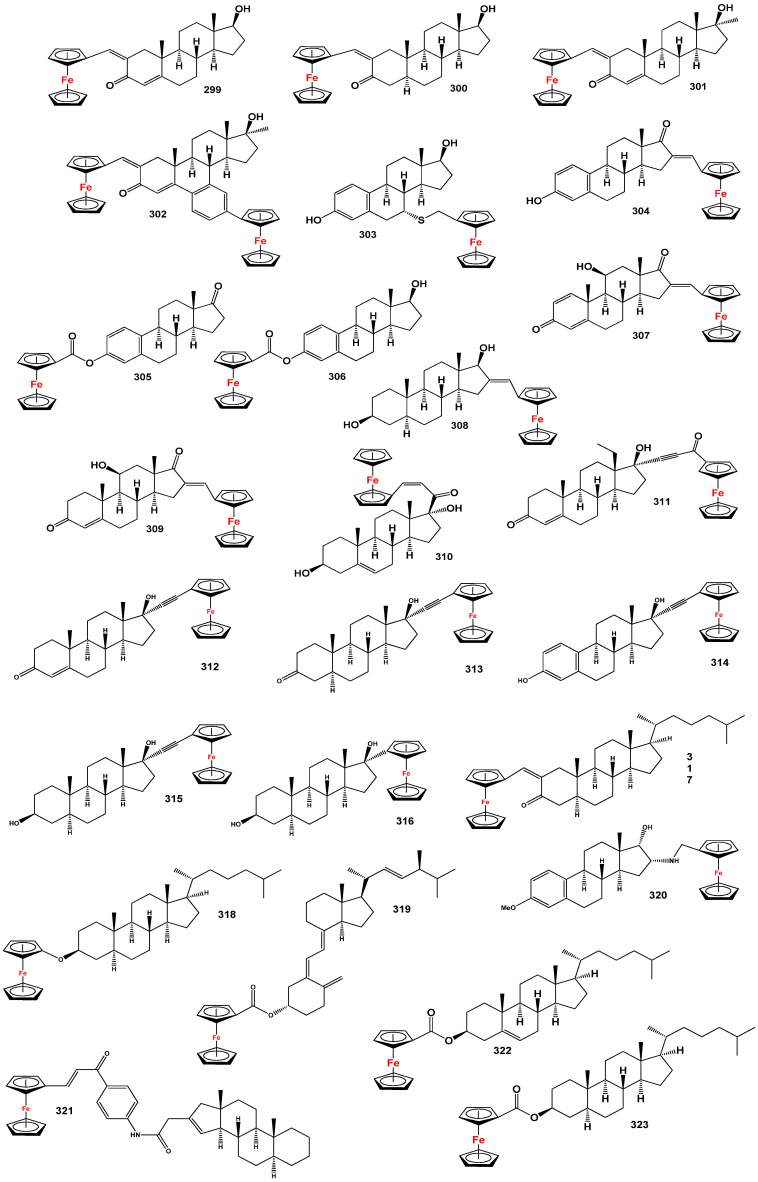
Structures of ferrocene steroid conjugates.

**Figure 49 biomedicines-11-02698-f049:**
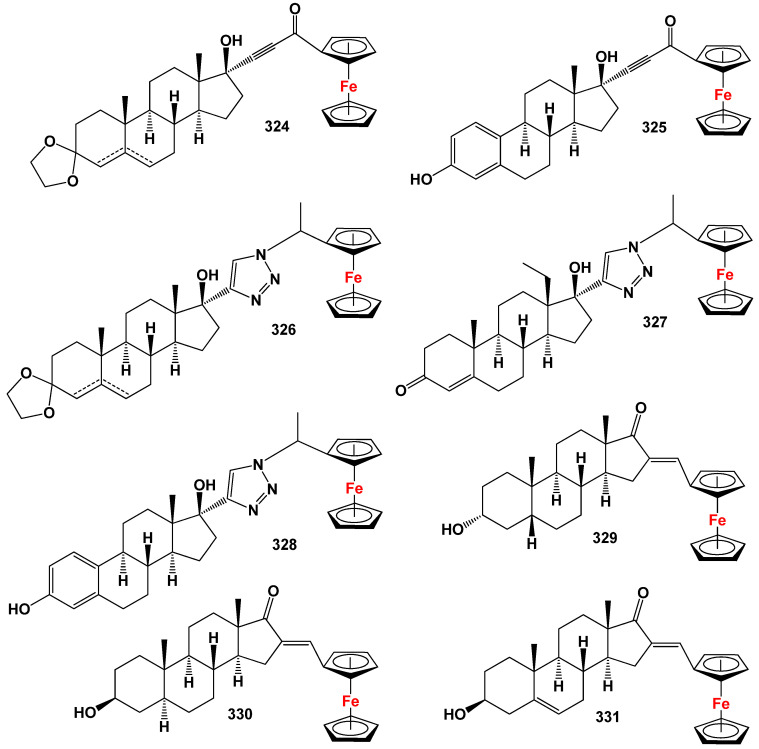
Bioactive ferrocene steroid conjugates.

**Figure 50 biomedicines-11-02698-f050:**
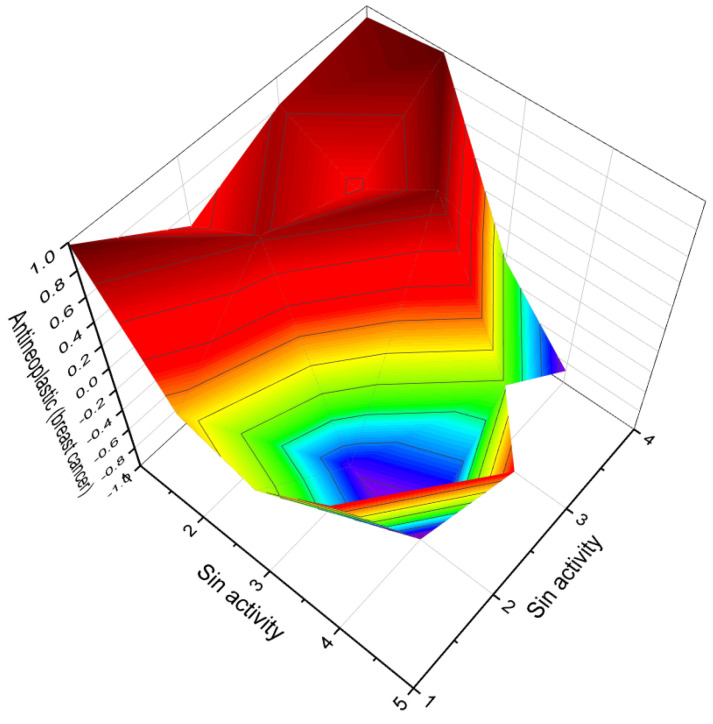
3D graph illustrating the predicted and calculated activity of ferrocene steroid conjugates (**311**, **317**, **320**, and **321**) with a maximum of over 96% confidence as a strong anti-breast cancer drug. In materials science, ferrocene steroid conjugates have been explored for their potential in developing functional materials, such as sensors and molecular switches. The combination of the redox-active ferrocene unit with the structural versatility of steroids opens possibilities for designing materials with tailored properties and responsive behavior. Furthermore, ferrocene steroid conjugates have shown promise in catalysis, where the unique electronic properties of ferrocene can influence the reactivity and selectivity of the conjugates. These conjugates can serve as catalysts or catalytic precursors in various transformations, offering new synthetic pathways and opportunities for sustainable and efficient chemical processes. The study of ferrocene steroid conjugates represents an interdisciplinary field at the intersection of chemistry and biology. The exploration of their properties and applications provides valuable insights into the synergistic effects arising from the combination of ferrocene and steroid motifs.

**Figure 51 biomedicines-11-02698-f051:**
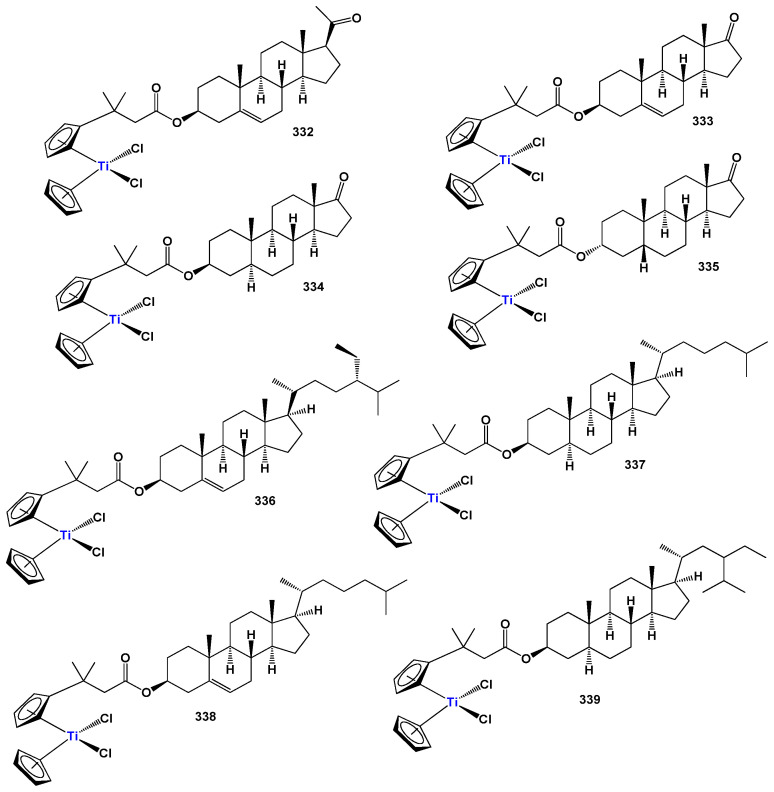
Structures of titanocene steroid conjugates.

**Figure 52 biomedicines-11-02698-f052:**
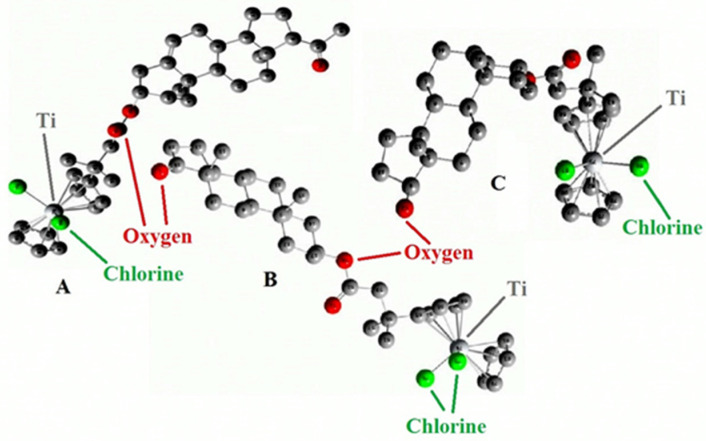
Density functional theory calculated structures of titanocene steroid conjugates. (A = 332, B = 338 and C = 339).

**Figure 53 biomedicines-11-02698-f053:**
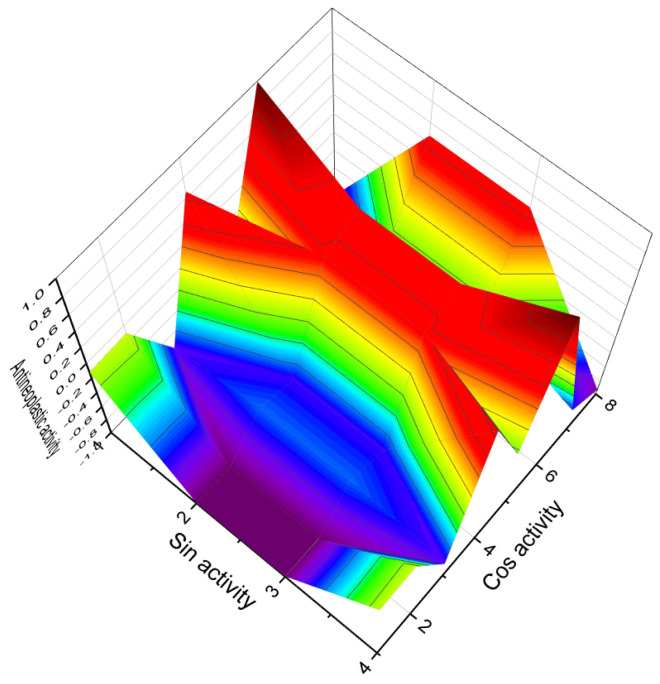
3D graph illustrating the predicted and calculated activity of titanocene steroid conjugate (**338**) with a high confidence level of over 99%. The graph indicates that this conjugate exhibits strong antineoplastic activity and shows potential as a therapeutic agent for the treatment of prostatic hyperplasia and proliferative diseases.

**Figure 54 biomedicines-11-02698-f054:**
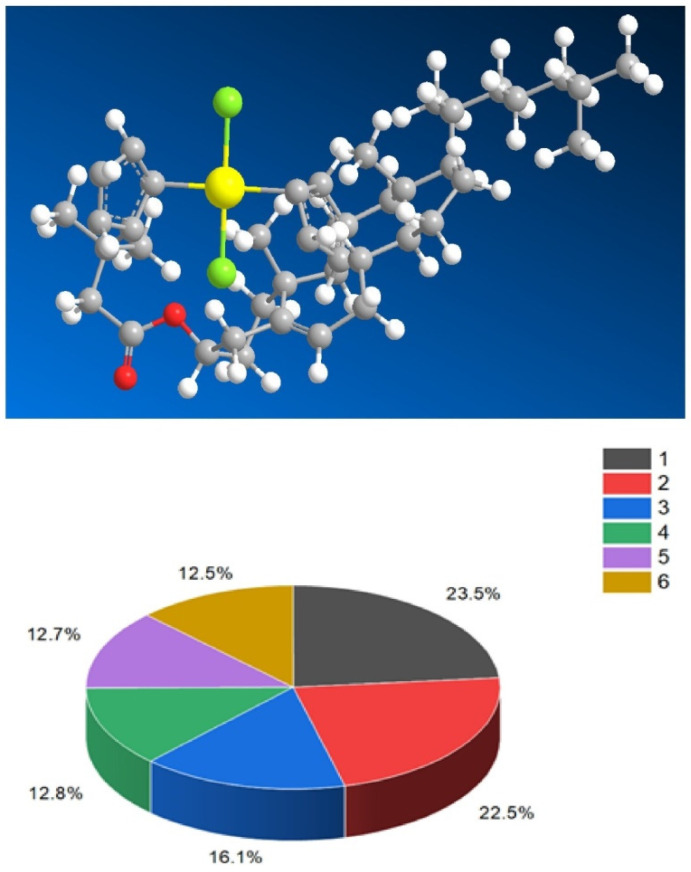
3D model and percentage distribution of the dominant antitumor activity for the titanocene steroid conjugate (**338**). The graph depicts the distribution of activities as follows: 1, *Antineoplastic* (23.5%); 2, *Toxic* (22.5%); 3, *Antineoplastic, alkylator* (16.1%); 4, *Proliferative diseases treatment* (12.8%), 5, *Prostatic (benign) hyperplasia treatment* (12.7%), and 6, *Apoptosis agonist* (12.5%). These percentages represent the relative contribution of each activity to the overall biological profile of the titanocene steroid conjugate (**338**). The titanium atom is highlighted in yellow, and the chlorine atoms in green.

**Table 1 biomedicines-11-02698-t001:** Biological activities of cyanosteroids (**1**–**23**) [41].

No.	Dominated Biological Activity (Pa) *	Additional Predicted Activities (Pa) *
**1**	Menstruation disorders treatment (0.962)	Antineoplastic (0.821)
	Menopausal disorders treatment (0.954)	Anti-inflammatory (0.793)
	Ovulation inhibitor (0.888)	Diuretic (0.781)
	Contraceptive (0.861)	Endometriosis treatment (0.604)
	Androgen antagonist (0.777)	Psychosexual dysfunction treatment (0.567)
**2**	Menopausal disorders treatment (0.929)	Antineoplastic (0.839)
	Contraceptive (0.859)	Respiratory analeptic (0.830)
	Ovulation inhibitor (0.829)	Psychosexual dysfunction treatment (0.514)
	Menstruation disorders treatment (0.776)	Prostate cancer treatment (0.500)
**3**	Menopausal disorders treatment (0.754)	Antineoplastic (0.813)
	Ovulation inhibitor (0.692)	Prostate disorders treatment (0.589)
**4**	Menstruation disorders treatment (0.980)	Antineoplastic (0.852)
	Menopausal disorders treatment (0.965)	Ovulation inhibitor (0.840)
	Contraceptive (0.857)	Anti-inflammatory (0.826)
**5**	Hair growth stimulant (0.875)	Antineoplastic (0.768)
	Male reproductive dysfunction treatment (0.833)	Immunosuppressant (0.739)
**6**	Male reproductive dysfunction treatment (0.875)	Ovulation inhibitor (0.745)
**7**	Cytoprotectant (0.963)	Antineoplastic (melanoma) (0.831)
	Apoptosis agonist (0.937)	Antineoplastic (pancreatic cancer) (0.777)
	Chemopreventive (0.900)	Antineoplastic (solid tumors (0.747)
**8**	Lipid metabolism regulator (0.965)	Anti-secretoric (0.958)
	Steroid synthesis inhibitor (0.808)	Antineoplastic (0.909)
**9**	Antineoplastic (0.913)	Steroid synthesis inhibitor (0.800)
	Apoptosis agonist (0.813)	Lipid metabolism regulator (0.750)
**10**	Androgen antagonist (0.958)	Antineoplastic (0.874)
	Steroid synthesis inhibitor (0.829)	Prostate cancer treatment (0.809)
**11**	Androgen antagonist (0.987)	Ovulation inhibitor (0.909)
	Steroid synthesis inhibitor (0.773)	Contraceptive (0.875)
	Estrogen antagonist (0.750)	Menopausal disorders treatment (0.865)
**12**	Antineoplastic (0.787)	Anti-inflammatory (0.682)
**13**	Antineoplastic (0.811)	Anti-inflammatory (0.731)
**14**	5-Alpha-reductase inhibitor (0.841)	Anesthetic general (0.796)
**15**	Antineoplastic (0.849)	Anti-inflammatory (0.752)
**16**	Male reproductive dysfunction treatment (0.896)	Antineoplastic (0.870)
**17**	Anti-inflammatory (0.938)	Antineoplastic (0.898)
	Aromatase inhibitor (0.818)	Estrogen antagonist (0.723)
**18**	Anti-inflammatory (0.963)	Estrogen antagonist (0.727)
	Antineoplastic (0.894)	Aromatase inhibitor (0.739)
**19**	Anti-inflammatory (0.934)	Estrogen antagonist (0.807)
	Antineoplastic (0.904)	Aromatase inhibitor (0.785)
**20**	Menopausal disorders treatment (0.925)	Anesthetic general (0.825)
	Contraceptive female (0.789)	Ovulation inhibitor (0.756)
**21**	Anti-inflammatory (0.863)	Hair growth stimulant (0.914)
	Estrogen antagonist (0.816)	Contraceptive (0.912)
	Menopausal disorders treatment (0.813)	Ovulation inhibitor (0.844)
**22**	Antineoplastic (0.868)	Hair growth stimulant (0.784)
	Menopausal disorders treatment (0.833)	Respiratory analeptic (0.750)
	Contraceptive (0.770)	Hypogonadism treatment (0.700)
**23**	Heart failure treatment (0.968)	Anti-hyperaldosteronism (0.963)
	Diuretic (0.967)	Antihypertensive (0.930)
	Cardiotonic (0.933)	Atherosclerosis treatment (0.755)

* Only activities with Pa > 0.5 are shown. The primary biological activity of steroids is considered. significant when the value of Pa exceeds 0.5. However, it is worth noting that for certain steroids.

**Table 2 biomedicines-11-02698-t002:** Biological activities of cyanosteroids (**24**–**51**) [41].

No.	Dominated Biological Activity (Pa) *	Additional Predicted Activities (Pa) *
**24**	Ovulation inhibitor (0.942)	Muscular dystrophy treatment (0.927)
	Menopausal disorders treatment (0.876)	Respiratory analeptic (0.907)
	Male reproductive dysfunction treatment (0.854)	Neuroprotector (0.904)
	Contraceptive (0.725)	Oxytocic (0.755)
**25**	Ovulation inhibitor (0.835)	Antineoplastic (0.779)
	Menopausal disorders treatment (0.658)	Dementia treatment (0.624)
	Contraceptive (0.649)	Prostate cancer treatment (0.603)
**26**	Anesthetic general (0.913)	Cholesterol antagonist (0.936)
	Anesthetic (0.792)	Erythropoiesis stimulant (0.879)
**27**	Cholesterol antagonist (0.707)	Ovulation inhibitor (0.843)
	Anti-hypercholesterolemic (0.618)	Menopausal disorders treatment (0.813)
**28**	Anti-hypercholesterolemic (0.826)	Ovulation inhibitor (0.779)
	Cholesterol antagonist (0.786)	Contraceptive (0.719)
**29**	Anti-inflammatory (0.846)	Contraceptive (0.796)
	Antineoplastic (0.833)	Ovulation inhibitor (0.682)
**30**	Cholesterol antagonist (0.840)	Antineoplastic (0.869)
	Anti-hypercholesterolemic (0.706)	Spasmolytic, urinary (0.857)
	Transcription factor NF kappa B inhibitor (0.687)	Anti-inflammatory (0.840)
**31**	Anti-osteoporotic (0.814)	Antineoplastic (0.780)
	Hypolipemic (0.711)	Prostatic (benign) hyperplasia treatment (0.609)
**32**	Cholesterol antagonist (0.903)	Erythropoiesis stimulant (0.700)
	Anti-hypercholesterolemic (0.743)	Cytoprotectant (0.699)
**33**	Antineoplastic (0.878)	Ovulation inhibitor (0.872)
	Apoptosis agonist (0.699)	Menopausal disorders treatment (0.715)
**34**	Menopausal disorders treatment (0.932)	Diuretic (0.788)
**35**	Cholesterol synthesis inhibitor (0.911)	Anesthetic general (0.913)
	Atherosclerosis treatment (0.908)	Respiratory analeptic (0.898)
**36**	Cholesterol synthesis inhibitor (0.932)	Anesthetic general (0.889)
	Anti-hypercholesterolemic (0.838)	Respiratory analeptic (0.831)
**37**	Antineoplastic (0.845)	Ovulation inhibitor (0.835)
	Cytoprotectant (0.764)	Menopausal disorders treatment (0.814)
**38**	Inhibitor 5α-reductase type II (0.921)	Psychotropic (0.952)
	Anxiolytic (0.909)	Antidepressant (0.914)
**39**	Inhibitor 5α-reductase type II (0.933)	Psychotropic (0.943)
	Anxiolytic (0.911)	Antidepressant (0.908)
**40**	Contraceptive (0.977)	Antineoplastic (0.910)
	Menopausal disorders treatment (0.872)	Prostatic (benign) hyperplasia treatment (0.666)
**41**	Anti-inflammatory (0.837)	Contraceptive (0.803)
**42**	Male reproductive dysfunction treatment (0.874)	Antineoplastic (0.864)
**43**	Male reproductive dysfunction treatment (0.886)	Antineoplastic (0.871)
**44**	Antineoplastic (sarcoma) (0.808)	Anti-inflammatory (0.845)
**45**	Antineoplastic (breast cancer) (0.885)	Apoptosis agonist (0.710)
**46**	Antineoplastic (0.894)	Apoptosis agonist (0.788)
**47**	Lipid metabolism regulator (0.748)	Anti-hypercholesterolemic (0.682)
**48**	Antineoplastic (0.847)	Apoptosis agonist (0.810)
**49**	Antineoplastic (0.843)	Male reproductive dysfunction treatment (0.817)
**50**	Antineoplastic (0.879)	Male reproductive dysfunction treatment (0.865)
**51**	Anti-hypercholesterolemic (0.878)	Antineoplastic (0.849)
	Hypolipemic (0.735)	Apoptosis agonist (0.820)
	Atherosclerosis treatment (0.544)	Prostate cancer treatment (0.585)
	Cholesterol synthesis inhibitor (0.524)	Cytoprotectant (0.584)

* Only activities with Pa > 0.5 are shown. The main biological activity has a value. Where Pa is more than 0.5, but for some steroids.

**Table 3 biomedicines-11-02698-t003:** Biological activities of nitro-steroids (**52**–**72**) [110].

No.	Dominated Biological Activity (Pa) *	Additional Predicted Activities (Pa) *
**52**	Respiratory analeptic (0.951)	Anti-hypercholesterolemic (0.908)
	Anesthetic general (0.929)	Analeptic (0.847)
	Anesthetic (0.815)	Dermatologic (0.747)
**53**	Respiratory analeptic (0.962)	Analeptic (0.873)
	Anesthetic general (0.917)	Anti-hypercholesterolemic (0.845)
**54**	Respiratory analeptic (0.953)	Analeptic (0.865)
	Anesthetic general (0.951)	Anti-hypercholesterolemic (0.839)
**55**	Antifungal (0.893)	Ovulation inhibitor (0.841)
	Anti-inflammatory (0.751)	Muscular dystrophy treatment (0.813)
**56**	Ovulation inhibitor (0.872)	Antineoplastic (0.818)
	Respiratory analeptic (0.809)	Prostate disorders treatment (0.660)
**57**	Antineoplastic (0.870)	Contraceptive (0.695)
	Prostatic (benign) hyperplasia treatment (0.651)	Ovulation inhibitor (0.646)
**58**	Antineoplastic (0.792)	Anesthetic general (0.882)
	Acute neurologic disorders treatment (0.761)	Ovulation inhibitor (0.772)
**59**	Neuroprotector (0.810)	Respiratory analeptic (0.789)
	Acute neurologic disorders treatment (0.791)	Anesthetic general (0.753)
**60**	Antineoplastic (0.797)	Ovulation inhibitor (0.885)
	Prostate disorders treatment (0.703)	Anesthetic general (0.750)
**61**	Antineoplastic (0.862)	Dermatologic (0.716)
	Prostate disorders treatment (0.784)	
	Prostatic (benign) hyperplasia treatment (0.675)	
**62**	Respiratory analeptic (0.882)	Antineoplastic (0.790)
	Analeptic (0.768)	Erythropoiesis stimulant (0.740)
**63**	Anesthetic general (0.865)	Prostate disorders treatment (0.750)
	Erythropoiesis stimulant (0.805)	Prostatic (benign) hyperplasia treatment (0.665)
**64**	Inhibitor 5a-reductase (0.971)	Respiratory analeptic (0.964)
	Spasmolytic, Papaverin-like (0.667)	Analeptic (0.884)
**65**	Inhibitor 5a-reductase (0.933)	Prostate disorders treatment (0.920)
	Spasmolytic, Papaverin-like (0.623)	Antineoplastic (0.764)
	Erythropoiesis stimulant (0.704)	Prostatic (benign) hyperplasia treatment (0.714)
**66**	Prostate disorders treatment (0.959)	Inhibitor 5a-reductase (0.911)
	Prostatic (benign) hyperplasia treatment (0.785)	Respiratory analeptic (0.703)
	Antineoplastic (0.764)	Ovulation inhibitor (0.689)
**67**	Prostate disorders treatment (0.976)	Inhibitor 5a-reductase (0.889)
	Prostatic (benign) hyperplasia treatment (0.895)	Dermatologic (0.785)
**68**	Prostate disorders treatment (0.946)	Inhibitor 5a-reductase (0.876)
	Prostatic (benign) hyperplasia treatment (0.883)	Dermatologic (0.791)
**69**	Prostate disorders treatment (0.915)	Dermatologic (0.766)
	Prostatic (benign) hyperplasia treatment (0.753)	
**70**	Respiratory analeptic (0.918)	Dermatologic (0.741)
	Anesthetic general (0.839)	Anti-psoriatic (0.652)
**71**	Respiratory analeptic (0.880)	Anti-eczematic (0.791)
	Anesthetic general (0.812)	Dermatologic (0.753)
**72**	Respiratory analeptic (0.929)	Anesthetic general (0.919)
	Analeptic (0.833)	Anti-hypercholesterolemic (0.819)

* Only activities with Pa > 0.5 are shown.

**Table 4 biomedicines-11-02698-t004:** Biological activities of nitro-steroids (**73**–**100**) [110].

No.	Dominated Biological Activity (Pa) *	Additional Predicted Activities (Pa) *
**73**	Anti-eczematic (0.814)	Antineoplastic (0.709)
	Dermatologic (0.736)	Prostate disorders treatment (0.708)
**74**	Antineoplastic (0.799)	Respiratory analeptic (0.748)
	Prostate disorders treatment (0.715)	Antiallergic (0.681)
	Prostatic (benign) hyperplasia treatment (0.645)	Dermatologic (0.646)
**75**	Ovulation inhibitor (0.819)	Neuroprotector (0.807)
	Analeptic (0.706)	Acute neurologic disorders treatment (0.727)
**76**	Ovulation inhibitor (0.811)	Neuroprotector (0.810)
	Analeptic (0.723)	Acute neurologic disorders treatment (0.707)
**77**	Antineoplastic (0.906)	Respiratory analeptic (0.860) Analeptic (0.726)
	Prostate disorders treatment (0.883)	
	Prostatic (benign) hyperplasia treatment (0.736)	
**78**	4-Methyl sterol oxidase (0.925)	Prostate disorders treatment (0.887)
	Neuroprotector (0.708)	Antineoplastic (0.870)
	Immunosuppressant (0.648)	Prostatic (benign) hyperplasia treatment (0.768)
**79**	4-Methyl sterol oxidase (0.914)	Prostate disorders treatment (0.892)
	Neuroprotector (0.742)	Antineoplastic (0.843)
**80**	Prostate disorders treatment (0.976)	4-Methyl sterol oxidase (0.909)
	Antineoplastic (0.922)	Neuroprotector (0.723)
**81**	Antineoplastic (0.890)	Anti-inflammatory (0.755)
	Prostate disorders treatment (0.809)	Antifungal (0.753)
**82**	Respiratory analeptic (0.883)	Anti-eczematic (0.831)
**83**	Respiratory analeptic (0.877)	Anti-eczematic (0.829)
**84**	Prostate disorders treatment (0.812)	Dermatologic (0.793)
**85**	Respiratory analeptic (0.905)	Anti-eczematic (0.836)
	Analeptic (0.838)	Dermatologic (0.756)
**86**	Respiratory analeptic (0.976)	Antineoplastic (0.898)
	Analeptic (0.900)	Anti-secretoric (0.880)
**87**	Antineoplastic (0.903)	Respiratory analeptic (0.881)
	Prostate disorders treatment (0.868)	Male reproductive dysfunction treatment (0.871)
	Prostatic (benign) hyperplasia treatment (0.726)	Ovulation inhibitor (0.721)
**88**	Respiratory analeptic (0.928)	Male reproductive dysfunction treatment (0.889)
	Neuroprotector (0.916)	Muscular dystrophy treatment (0.882)
	Antineoplastic (0.904)	Ovulation inhibitor (0.866)
**89**	Muscular dystrophy treatment (0.940)	Respiratory analeptic (0.921)
	Ovulation inhibitor (0.812)	Antineoplastic (0.909)
	Hypogonadism treatment (0.746)	Prostate disorders treatment (0.863)
**90**	Antineoplastic (0.905)	Respiratory analeptic (0.886)
	Prostate disorders treatment (0.702)	Analeptic (0.765)
**91**	Anti-eczematic (0.831)	Dermatologic (0.709)
**92**	Respiratory analeptic (0.953)	Anesthetic general (0.919)
	Analeptic (0.858)	Anti-hypercholesterolemic (0.892)
**94**	Gonadotropin inhibitor (0.932)	Muscular dystrophy treatment (0.891)
	Ovulation inhibitor (0.886)	Male reproductive dysfunction treatment (0.879)
**95**	Respiratory analeptic (0.980)	Anti-inflammatory (0.930)
	Anti-secretoric (0.937)	Antiallergic (0.827)
**96**	Respiratory analeptic (0.965)	Anti-inflammatory (0.944)
	Anti-secretoric (0.934)	Antiallergic (0.864)
**97**	Anesthetic general (0.905)	Respiratory analeptic (0.782)
**98**	Cardiotonic (0.890)	Anesthetic general (0.707)
**99**	Anesthetic general (0.880)	Respiratory analeptic (0.868)
**100**	Anesthetic general (0.869)	Respiratory analeptic (0.851)

* Only activities with Pa > 0.5 are shown.

**Table 6 biomedicines-11-02698-t006:** Biological activities of steroids bearing chlorine atom (**117**–**130**) [151,175].

No.	Dominated Biological Activity (Pa) *	Additional Predicted Activities (Pa) *
**117**	Anti-seborrheic (0.931)	Respiratory analeptic (0.796)
	Growth stimulant (0.923)	Endometriosis treatment (0.770)
**118**	Anti-seborrheic (0.936)	Antineoplastic (0.888)
	Growth stimulant (0.900)	Respiratory analeptic (0.828)
**119**	Anti-seborrheic (0.941)	Antineoplastic (0.893)
	Growth stimulant (0.873)	Anti-inflammatory (0.868)
**120**	Anti-seborrheic (0.909)	Antineoplastic (0.888)
	Growth stimulant (0.886)	Anti-inflammatory (0.806)
**121**	Respiratory analeptic (0.925)	Anti-inflammatory (0.871)
	Anti-hypercholesterolemic (0.914)	Antiallergic (0.747)
**122**	Respiratory analeptic (0.925)	Anti-inflammatory (0.871)
	Anti-hypercholesterolemic (0.914)	Antiallergic (0.747)
**123**	Gynecological disorders treatment (0.959)	Anti-hypercholesterolemic (0.887)
	Antineoplastic (0.947)	Lipid metabolism regulator (0.828)
	Growth stimulant (0.848)	Antidepressant (0.833)
**124**	Anesthetic general (0.951)	Anti-inflammatory (0.863)
	Respiratory analeptic (0.764)	Antineoplastic (0.821)
	Antiallergic (0.734)	Antipruritic (0.797)
**125**	Anti-inflammatory (0.888)	Ovulation inhibitor (0.844)
	Anti-asthmatic (0.847)	Menopausal disorders treatment (0.614)
**126**	Antineoplastic (0.818)	Anesthetic general (0.769)
**127**	Anti-inflammatory (0.979)	Antiarthritic (0.926)
	Anti-asthmatic (0.957)	Antipruritic, allergic (0.832)
	Antiallergic (0.907)	Rheumatoid arthritis treatment (0.607)
**128**	Anti-hypercholesterolemic (0.926)	Neuroprotector (0.872)
	Anesthetic general (0.913)	Erythropoiesis stimulant (0.817)
	Respiratory analeptic (0.897)	Immunosuppressant (0.755)
**129**	Anti-inflammatory (0.956)	Antiallergic (0.904)
	Antineoplastic (0.906)	Antipruritic, allergic (0.741)
	Respiratory analeptic (0.816)	
**130**	Anti-inflammatory (0.955)	Immunosuppressant (0.812)
	Antiallergic (0.920)	Anti-asthmatic (0.795)

* Only activities with Pa > 0.5 are shown.

**Table 7 biomedicines-11-02698-t007:** Biological activities of steroids bearing bromine atom(s) (**131**–**142**) [175].

No.	Dominated Biological Activity (Pa) *	Additional Predicted Activities (Pa) *
**131**	Antineoplastic (0.886)	Anti-inflammatory (0.819)
	Prostate cancer treatment (0.674)	Antibacterial (0.786)
**132**	Antineoplastic (0.902)	Anti-inflammatory (0.710)
	Prostatic (benign) hyperplasia treatment (0.673)	Cystic fibrosis treatment (0.555)
**133**	Ovulation inhibitor (0.887)	Antineoplastic (0.833)
	Contraceptive (0.609)	Prostate cancer treatment (0.694)
**134**	Ovulation inhibitor (0.859)	Antineoplastic (0.848)
	Menopausal disorders treatment (0.691)	Prostate cancer treatment (0.561)
**135**	Anti-inflammatory (0.895)	Ovulation inhibitor (0.801)
	Respiratory analeptic (0.875)	Contraceptive (0.745)
**136**	Anti-inflammatory (0.879)	Ovulation inhibitor (0.783)
	Antineoplastic (0.816)	Menopausal disorders treatment (0.655)
**137**	Apoptosis agonist (0.752)	Antinociceptive (0.734)
	Antineoplastic (0.751)	Cytoprotectant (0.682)
**138**	Antineoplastic (0.891)	Angiogenesis inhibitor (0.945)
	Apoptosis agonist (0.785)	Antiviral (Influenza) (0.567)
**139**	Antiviral (Influenza) (0.888)	Apoptosis agonist (0.775)
	Antibacterial (0.856)	Antineoplastic (0.757)
**140**	Ovulation inhibitor (0.793)	Antineoplastic (0.751)
	Menopausal disorders treatment (0.579)	Cystic fibrosis treatment (0.740)
**141**	Antineoplastic (0.785)	Ovulation inhibitor (0.742)
	Genital warts treatment (0.759)	Menopausal disorders treatment (0.618)
**142**	Apoptosis agonist (0.869)	Anti-inflammatory (0.853)
	Antineoplastic (0.852)	Antiallergic (0.665)

* Only activities with Pa > 0.5 are shown.

**Table 8 biomedicines-11-02698-t008:** Biological activities of steroids bearing iodine atom (**143**–**159**) [175].

No.	Dominated Biological Activity (Pa) *	Additional Predicted Activities (Pa) *
**143**	Antineoplastic (0.906)	Ovulation inhibitor (0.840)
	Prostate cancer treatment (0.712)	Male reproductive dysfunction treatment (0.757)
**144**	Anti-hypercholesterolemic (0.877)	Ovulation inhibitor (0.814)
	Antineoplastic (0.823)	Male reproductive dysfunction treatment (0.754)
**145**	Respiratory analeptic (0.947)	Antineoplastic (0.855)
	Diuretic (0.813)	Apoptosis agonist (0.780)
**146**	Respiratory analeptic (0.895)	Contraceptive (0.796)
	Antineoplastic (0.820)	Ovulation inhibitor (0.651)
**147**	Respiratory analeptic (0.980)	Anti-inflammatory (0.924)
	Anti-secretoric (0.932)	Antineoplastic (0.915)
**148**	Anti-inflammatory (0.910)	Ovulation inhibitor (0.724)
	Anti-secretoric (0.828)	Contraceptive (0.656)
**149**	Antineoplastic (0.924)	Respiratory analeptic (0.791)
	Prostate cancer treatment (0.710)	Erythropoiesis stimulant (0.738)
**150**	Antineoplastic (0.905)	Ovulation inhibitor (0.726)
	Prostate cancer treatment (0.608)	Menopausal disorders treatment (0.643)
**151**	Antineoplastic (0.883)	Antileukemic (0.599)
**152**	Respiratory analeptic (0.952)	Ovulation inhibitor (0.913)
	Anti-hypercholesterolemic (0.927)	Menopausal disorders treatment (0.721)
**153**	Alopecia treatment (0.864)	Anti-osteoporotic (0.793)
	Antineoplastic (0.847)	Anti-secretoric (0.729)
**154**	Contraceptive (0.908)	Antineoplastic (0.883)
	Ovulation inhibitor (0.885)	Prostate disorders treatment (0.712)
	Menopausal disorders treatment (0.677)	Apoptosis agonist (0.606)
**155**	Antineoplastic (0.846)	Anti-eczematic (0.817)
	Prostate cancer treatment (0.675)	Anti-psoriatic (0.621)
**156**	Ovulation inhibitor (0.833)	Antineoplastic (0.812)
	Respiratory analeptic (0.803)	Apoptosis agonist (0.647)
**157**	Antineoplastic (0.827)	Cardiotonic (0.723)
	Prostate cancer treatment (0.775)	Antiarrhythmic (0.568)
**158**	Ovulation inhibitor (0.903)	Antineoplastic (0.860)
	Contraceptive (0.835)	Prostate disorders treatment (0.804)
	Menopausal disorders treatment (0.692)	Prostate cancer treatment (0.719)
**159**	Ovulation inhibitor (0.958)	Antineoplastic (0.889)
	Contraceptive (0.878)	Prostate disorders treatment (0.876)
	Menopausal disorders treatment (0.752)	Prostate cancer treatment (0.769)

* Only activities with Pa > 0.5 are shown.

**Table 10 biomedicines-11-02698-t010:** Biological activities of steroids bearing epithio group (**178**–**189**) [110].

No.	Dominated Biological Activity (Pa) *	Additional Predicted Activities (Pa) *
**178**	Cardiotonic (0.936)	Antineoplastic (0.912)
	Antipruritic (0.787)	Anti-secretoric (0.857)
**179**	Anti-hypercholesterolemic (0.865)	Hepatic disorders treatment (0.792)
	Cholesterol antagonist (0.858)	Hepatoprotectant (0.762)
**180**	Hypolipemic (0.805)	Antineoplastic (0.828)
	Cholesterol antagonist (0.754)	Apoptosis agonist (0.785)
	Atherosclerosis treatment (0.657)	
**181**	Aromatase inhibitor (0.884)	Male reproductive dysfunction treatment (0.896)
	Antineoplastic (0.806)	
**182**	Aromatase inhibitor (0.854)	Ovulation inhibitor (0.750)
	Antineoplastic (0.746)	Male reproductive dysfunction treatment (0.720)
**183**	Cholesterol antagonist (0.916)	Respiratory analeptic (0.903)
	Anti-hypercholesterolemic (0.836)	Anesthetic general (0.858)
**184**	Neuroprotector (0.916)	Cholesterol antagonist (0.893)
	Cardiotonic (0.788)	Anti-hypercholesterolemic (0.771)
**185**	Muscular dystrophy treatment (0.873)	Anti-hypercholesterolemic (0.872)
	Acute neurologic disorders treatment (0.849)	Cholesterol antagonist (0.817)
**186**	Cholesterol synthesis inhibitor (0.834)	Hepatoprotectant (0.859)
	Cholesterol antagonist (0.824)	Hepatic disorders treatment (0.762)
**187**	Anti-secretoric (0.823)	Antineoplastic (0.781)
	Anti-osteoporotic (0.665)	Prostatic (benign) hyperplasia treatment (0.666)
**188**	Anti-seborrheic (0.924)	Anti-hypercholesterolemic (0.775)
	Anti-eczematic (0.797)	Estrogen antagonist (0.670)
**189**	Ovulation inhibitor (0.830)	Cholesterol antagonist (0.734)

* Only activities with Pa > 0.5 are shown.

**Table 11 biomedicines-11-02698-t011:** Biological activities of steroids bearing boron atom(s) (**190**–**201**) [260].

No.	Dominated Biological Activity (Pa) *	Additional Predicted Activities (Pa) *
**190**	Anti-eczematic (0.804)	Antihypertensive (0.798)
	Dermatologic (0.716)	Myocardial ischemia treatment (0.751)
	Anti-psoriatic (0.665)	Antineoplastic (0.735)
**191**	Anti-eczematic (0.796)	Antineoplastic (0.796)
	Dermatologic (0.767)	Anti-hypercholesterolemic (0.756)
**192**	Antineoplastic (0.910)	Gynecological disorders treatment (0.866)
	Prostatic (benign) hyperplasia treatment (0.713)	Psychosexual dysfunction treatment (0.821)
**193**	Antineoplastic (0.883)	Gynecological disorders treatment (0.849)
	Prostatic (benign) hyperplasia treatment (0.695)	Psychosexual dysfunction treatment (0.817)
**194**	Antineoplastic (0.844)	Psychosexual dysfunction treatment (0.756)
	Prostatic (benign) hyperplasia treatment (0.631)	Gynecological disorders treatment (0.693)
**195**	Antineoplastic (0.870)	Psychosexual dysfunction treatment (0.772)
	Prostatic (benign) hyperplasia treatment (0.647)	Gynecological disorders treatment (0.714)
**196**	Antineoplastic (0.866)	Psychosexual dysfunction treatment (0.754)
	Antimetastatic (0.614)	Gynecological disorders treatment (0.631)
**197**	Antineoplastic (0.817)	Psychosexual dysfunction treatment (0.759)
	Prostatic (benign) hyperplasia treatment (0.652)	Gynecological disorders treatment (0.707)
**198**	Antineoplastic (0.682)	Radiosensitizer (0.521)
	Antineoplastic (lymphocytic leukemia) (0.526)	
**199**	Antineoplastic (0.881)	Neuroprotector (0.870)
	Apoptosis agonist (0.742)	Chemosensitizer (0.755)
	Prostate disorders treatment (0.712)	Radiosensitizer (0.738)
**200**	Antineoplastic (0.762)	Radiosensitizer (0.587) Chemosensitizer (0.525)
	Antineoplastic (renal cancer) (0.574)	
	Antineoplastic (breast cancer) (0.524)	
	Prostate cancer treatment (0.513)	
	Antineoplastic (pancreatic cancer) (0.512)	
**201**	Genital warts treatment (0.894)	Antineoplastic (0.849)
	Rhinitis treatment (0.662)	Apoptosis agonist (0.639)
	Macular degeneration treatment (0.597)	Prostate disorders treatment (0.555)

*Only activities with Pa > 0.5 are shown.

**Table 12 biomedicines-11-02698-t012:** Biological activities of steroids bearing aluminum atom (**202**–**209**) [284].

No.	Dominated Biological Activity (Pa) *	Additional Predicted Activities (Pa) *
**202**	Antiprotozoal (Plasmodium) (0.906)	Anti-hypercholesterolemic (0.787)
	Antifungal (0.825)	Hypolipemic (0.753)
	Antibacterial (0.758)	Atherosclerosis treatment (0.602)
**203**	Antiprotozoal (Plasmodium) (0.906)	Anti-hypercholesterolemic (0.787)
	Antifungal (0.825)	Hypolipemic (0.753)
	Antibacterial (0.758)	Atherosclerosis treatment (0.602)
**204**	Antiprotozoal (Plasmodium) (0.910)	Anti-hypercholesterolemic (0.741)
	Antifungal (0.818)	Hypolipemic (0.675)
	Antibacterial (0.722)	Atherosclerosis treatment (0.580)
**205**	Antiprotozoal (Plasmodium) (0.924)	Anti-hypercholesterolemic (0.817)
	Cytoprotectant (0.751)	Hypolipemic (0.735)
	Biliary tract disorders treatment (0.744)	Cholesterol synthesis inhibitor (0.601)
**206**	Antiprotozoal (Plasmodium) (0.908)	Anti-hypercholesterolemic (0.881)
	Biliary tract disorders treatment (0.713)	Hypolipemic (0.691)
	Anti-psoriatic (0.695)	Myasthenia Gravis treatment (0.584)
**207**	Antiprotozoal (Plasmodium) (0.909)	Anti-hypercholesterolemic (0.899)
	Biliary tract disorders treatment (0.726)	Hypolipemic (0.776)
	Anti-psoriatic (0.717)	Atherosclerosis treatment (0.584)
**208**	Antiprotozoal (Plasmodium) (0.911)	Anti-hypercholesterolemic (0.907)
	Cytoprotectant (0.704)	Hypolipemic (0.748)
	Anti-psoriatic (0.700)	Atherosclerosis treatment (0.599)
**209**	Antiprotozoal (Plasmodium) (0.910)	Anti-hypercholesterolemic (0.895)
	Cytoprotectant (0.728)	Hypolipemic (0.738)
	Anti-psoriatic (0.697)	Atherosclerosis treatment (0.595)

* Only activities with Pa > 0.5 are shown.

**Table 13 biomedicines-11-02698-t013:** Biological activities of steroids bearing arsenic atom (**210**–**215**) [260].

No.	Dominated Biological Activity (Pa) *	Additional Predicted Activities (Pa) *
**210**	Antineoplastic (0.983)	Antiviral (0.866)
	Antiprotozoal (0.941)	
**211**	Antineoplastic (0.982)	Antiviral (0.882)
	Antiprotozoal (0.947)	
**212**	Antineoplastic (0.985)	Antiviral (0.939)
	Apoptosis agonist (0.755)	Anti-inflammatory (0.629)
**213**	Antineoplastic (0.984)	Antiprotozoal (0.946)
	Apoptosis agonist (0.716)	Antiviral (0.927)
**214**	Antineoplastic (0.870)	Dermatologic (0.818)
	Prostate disorders treatment (0.641)	Anti-inflammatory (0.517)
**215**	Antineoplastic (0.844)	Contraceptive (0.625)
	Apoptosis agonist (0.574)	Gynecological disorders treatment (0.604)

* Only activities with Pa > 0.5 are shown.

**Table 14 biomedicines-11-02698-t014:** Biological activities of steroids bearing astatine atom (**216**–**220**) [260].

No.	Dominated Biological Activity (Pa) *	Additional Predicted Activities (Pa) *
**216**	Anti-seborrheic (0.936)	Alopecia treatment (0.923)
	Growth stimulant (0.805)	Anti-hypercholesterolemic (0.920)
	Antifungal (0.789)	Anti-secretoric (0.884)
**217**	Anti-seborrheic (0.934)	Alopecia treatment (0.913)
	Growth stimulant (0.703)	Anti-hypercholesterolemic (0.901)
	Antifungal (0.689)	Anti-secretoric (0.836)
**218**	Anti-hypercholesterolemic (0.912)	Antineoplastic (0.839)
	Anti-secretoric (0.837)	Bone diseases treatment (0.777)
**219**	Respiratory analeptic (0.976)	Anti-hypercholesterolemic (0.967)
	Anesthetic general (0.924)	Hypolipemic (0.785)
	Antineoplastic (0.824)	Bone diseases treatment (0.796)
**220**	Anti-hypercholesterolemic (0.927)	Anti-eczematic (0.825)
	Antineoplastic (0.834)	Dermatologic (0.805)
	Hypolipemic (0.740)	Anti-psoriatic (0.685)

* Only activities with Pa > 0.5 are shown.

**Table 15 biomedicines-11-02698-t015:** Biological activities of steroids bearing germanium atom (**221**–**229**) [260].

No.	Dominated Biological Activity (Pa) *	Additional Predicted Activities (Pa) *
**221**	Respiratory analeptic (0.870)	Antineoplastic (0.860)
	Anesthetic (0.738)	Anti-hypercholesterolemic (0.806)
**222**	Respiratory analeptic (0.874)	Antineoplastic (0.852)
	Anesthetic general (0.856)	Cytoprotectant (0.714)
**223**	Antineoplastic (0.957)	Antiacne (0.939)
	Respiratory analeptic (0.779)	Dermatologic (0.928)
**224**	Antineoplastic (0.961)	Antiacne (0.952)
	Erythropoiesis stimulant (0.793)	Dermatologic (0.935)
	Respiratory analeptic (0.785)	Anti-seborrheic (0.830)
**225**	Antineoplastic (0.960)	Antiacne (0.951)
	Anesthetic general (0.841)	Dermatologic (0.940)
	Erythropoiesis stimulant (0.770)	Anti-eczematic (0.695)
**226**	Antineoplastic (0.964)	Antiacne (0.960)
	Anesthetic general (0.915)	Dermatologic (0.948)
	Erythropoiesis stimulant (0.896)	Anti-eczematic (0.735)
**227**	Antiacne (0.970)	Antineoplastic (0.969)
	Dermatologic (0.955)	Anesthetic general (0.816)
	Anti-seborrheic (0.691)	Prostate disorders treatment (0.738)
**228**	Antiacne (0.980)	Antineoplastic (0.974)
	Dermatologic (0.972)	Erythropoiesis stimulant (0.768)
	Anti-seborrheic (0.831)	Prostatic (benign) hyperplasia treatment (0.651)
**229**	Anti-seborrheic (0.918)	Ovulation inhibitor (0.847)
	Dermatologic (0.776)	Antineoplastic (0.821)
	Antiacne (0.613)	Prostatic (benign) hyperplasia treatment (0.606)

* Only activities with Pa > 0.5 are shown.

**Table 16 biomedicines-11-02698-t016:** Biological activities of steroids bearing silicone atom(s) (**230**–**241**) [260].

No.	Dominated Biological Activity (Pa) *	Additional Predicted Activities (Pa) *
**230**	Antineoplastic (0.992)	Psychotropic (0.879)
	Contraceptive (0.879)	Anxiolytic (0.762)
	Ovulation inhibitor (0.872)	Neurodegenerative diseases treatment (0.587)
**231**	Antineoplastic (0.985)	Psychotropic (0.815)
	Contraceptive (0.841)	Anxiolytic (0.754)
	Ovulation inhibitor (0.812)	Neurodegenerative diseases treatment (0.725)
**232**	Antineoplastic (0.995)	Psychotropic (0.827)
	5 Hydroxytryptamine 2A antagonist (0.765)	Anxiolytic (0.806)
	5 Hydroxytryptamine 2 antagonist (0.588)	
**233**	Antineoplastic (0.994)	Psychotropic (0.834)
	5 Hydroxytryptamine 2A antagonist (0.740)	Anxiolytic (0.823)
**234**	Antineoplastic (0.973)	Psychotropic (0.702)
	Apoptosis agonist (0.887)	Erythropoiesis stimulant (0.618)
**235**	Ovulation inhibitor (0.926)	Psychotropic (0.846)
	Contraceptive (0.880)	Anxiolytic (0.742)
**236**	Ovulation inhibitor (0.859)	Antineoplastic (0.871)
	Contraceptive (0.799)	Hypolipemic (0.818)
	Menopausal disorders treatment (0.686)	Prostate disorders treatment (0.690)
**237**	Antineoplastic (0.977)	Hypolipemic (0.634)
	Apoptosis agonist (0.616)	Ovulation inhibitor (0.606)
	Prostate disorders treatment (0.613)	Contraceptive (0.575)
**238**	Antineoplastic (0.943)	Antiarthritic (0.927)
	Erythropoiesis stimulant (0.728)	Hypolipemic (0.751)
	Skeletal muscle relaxant (0.673)	Atherosclerosis treatment (0.688)
**239**	Antineoplastic (0.905)	Anti-inflammatory (0.755)
	Contraceptive (0.813)	Antifungal (0.748)
	Dermatologic (0.773)	Hypolipemic (0.752)
**240**	Anti-hypercholesterolemic (0.917)	Antineoplastic (0.883)
	Immunosuppressant (0.776)	Antiprotozoal (Leishmania) (0.858)
	Antipruritic (0.771)	Respiratory analeptic (0.800)
**241**	Antineoplastic (0.899)	Anti-hypercholesterolemic (0.796)
	Anabolic (0.878)	Anti-inflammatory (0.750)
	Prostatic (benign) hyperplasia treatment (0.605)	Respiratory analeptic (0.728)

* Only activities with Pa > 0.5 are shown.

**Table 17 biomedicines-11-02698-t017:** Biological activities of steroids bearing selenium atom (**242**–**267**) [260].

No.	Dominated Biological Activity (Pa) *	Additional Predicted Activities (Pa) *
**242**	Antineoplastic (0.894)	Alzheimer’s disease treatment (0.702)
	Chemoprotective (0.775)	Anti-psoriatic (0.691)
**243**	Antineoplastic (0.913)	Alzheimer’s disease treatment (0.794)
	Chemoprotective (0.816)	Cerebrovascular disorders treatment (0.774)
**244**	Cerebrovascular disorders treatment (0.869)	Psychosexual dysfunction treatment (0.791)
	Alzheimer’s disease treatment (0.833)	Neurodegenerative diseases treatment (0.775)
**245**	Antineoplastic (0.938)	Erythropoiesis stimulant (0.756)
	Chemoprotective (0.809)	Cerebrovascular disorders treatment (0.732)
	Alzheimer’s disease treatment (0.778)	Vascular (peripheral) disease treatment (0.689)
**246**	Antineoplastic (0.846)	Anti-hypercholesterolemic (0.841)
	Chemoprotective (0.786)	Erythropoiesis stimulant (0.791)
**247**	Anti-inflammatory (0.940)	Antineoplastic (0.897)
	Antiallergic (0.790)	Anticarcinogenic (0.778)
**248**	Anti-inflammatory (0.959)	Antiallergic (0.843)
	Respiratory analeptic (0.870)	Anti-asthmatic (0.757)
**249**	Ovulation inhibitor (0.918)	Anti-inflammatory (0.738)
	Antineoplastic (0.876)	Anti-hypercholesterolemic (0.726)
**250**	Alzheimer’s disease treatment (0.783)	Antineoplastic (0.899)
	Ovulation inhibitor (0.757)	Membrane permeability inhibitor (0.788)
	Chemoprotective (0.740)	Cytochrome P450 inhibitor (0.559)
**251**	Anti-hypercholesterolemic (0.905)	Antineoplastic (0.858)
	Anesthetic general (0.901)	Erythropoiesis stimulant (0.855)
**252**	Antineoplastic (0.926)	Anti-hypercholesterolemic (0.877)
	Chemoprotective (0.780)	Anti-inflammatory (0,865)
**253**	Anti-hypercholesterolemic (0.908)	Anti-eczematic (0.825)
	Antineoplastic (0.882)	Dermatologic (0.799)
**254**	Respiratory analeptic (0.963)	Chemopreventive (0.916)
	Anesthetic general (0.946)	Antineoplastic (0.884)
	Wound healing agent (0.880)	Anticarcinogenic (0.753)
**255**	Respiratory analeptic (0.971)	Chemopreventive (0.923)
	Anesthetic general (0.910)	Antineoplastic (0.856)
	Wound healing agent (0.817)	Anticarcinogenic (0.804)
**256**	Anti-inflammatory (0.950)	Antineoplastic (0.818)
	Antiallergic (0.793)	Cell adhesion molecule inhibitor (0.745)
**257**	Anti-inflammatory (0.936)	Antineoplastic (0.813)
	Antiallergic (0.682)	Cell adhesion molecule inhibitor (0.752)
**258**	Choleretic (0.909)	Anti-hypercholesterolemic (0.905)
	Erythropoiesis stimulant (0.856)	Anesthetic general (0.903)
	Hepatoprotectant (0.829)	Antineoplastic (0.849)
**259**	Anti-inflammatory (0.907)	Respiratory analeptic (0.861)
	Inflammatory Bowel disease treatment (0.750)	Antineoplastic (0.855)
**260**	Chemopreventive (0.936)	Anticarcinogenic (0.815)
	Antineoplastic (0.918)	Prostate cancer treatment (0.795)
**261**	Antineoplastic (0.951)	Ovulation inhibitor (0.704)
	Chemoprotective (0.844)	Antihypertensive (0.649)
**262**	Antineoplastic (0.935)	Anti-inflammatory (0.918)
	Chemoprotective (0.804)	Ovulation inhibitor (0.826)
**263**	Hypolipemic (0.995)	Antioxidant (0.973)
	Atherosclerosis treatment (0.991)	Antineoplastic (0.852)
	Lipoprotein disorders treatment (0.982)	Chemoprotective (0.748)
**264**	Antineoplastic (0.922)	Anti-hypercholesterolemic (0.902)
	Chemopreventive (0.914)	Hypolipemic (0.899)
**265**	Antineoplastic (0.918)	Hypolipemic (0.913)
	Chemopreventive (0.894)	Anti-hypercholesterolemic (0.884)
	Anticarcinogenic (0.801)	Atherosclerosis treatment (0.822)
**266**	Hypolipemic (0.996)	Antioxidant (0.978)
	Atherosclerosis treatment (0.995)	Antineoplastic (0.904)
	Lipoprotein disorders treatment (0.991)	Chemoprotective (0.823)
	Antiarthritic (0.979)	Anticarcinogenic (0.795)
**267**	Hypolipemic (0.995)	Antioxidant (0.970)
	Atherosclerosis treatment (0.989)	Antineoplastic (0.850)
	Lipoprotein disorders treatment (0.980)	Chemoprotective (0.785)

* Only activities with Pa > 0.5 are shown.

**Table 18 biomedicines-11-02698-t018:** Biological activities of steroids bearing tellurium atom (**268**–**280**) [260].

No.	Dominated Biological Activity (Pa) *	Additional Predicted Activities (Pa) *
**268**	Antioxidant (0.956)	Neurodegenerative disease treatment (0.897)
	Anti-inflammatory (0.936)	Alzheimer’s disease treatment (0.877)
	Antineoplastic (0.935)	Atherosclerosis treatment (0.873)
	Erythropoiesis stimulant (0.815)	Antiparkinsonian (0.868)
**269**	Antioxidant (0.946)	Neurodegenerative diseases treatment (0.871)
	Anti-inflammatory (0.930)	Alzheimer’s disease treatment (0.863)
	Antineoplastic (0.930)	Atherosclerosis treatment (0.861)
	Anesthetic general (0.879)	Antiparkinsonian (0.833)
**270**	Antiprotozoal (Plasmodium) (0.818)	Alzheimer’s disease treatment (0.757)
	Antiprotozoal (0.801)	Neurodegenerative diseases treatment (0.735)
	Antioxidant (0.774)	Antiparkinsonian (0.730)
	Ovulation inhibitor (0.625)	Atherosclerosis treatment (0.670)
**271**	Alzheimer’s disease treatment (0.966)	Antineoplastic (0.828)
	Antioxidant (0.965)	Prostatic (benign) hyperplasia treatment (0.678)
	Anti-hypercholesterolemic (0.889)	Antineoplastic (pancreatic cancer) (0.522)
	Atherosclerosis treatment (0.703)	Prostate cancer treatment (0.501)
**272**	Antiparkinsonian (0.946)	Antioxidant (0.966)
	Neurodegenerative diseases treatment (0.945)	Atherosclerosis treatment (0.964)
	Alzheimer’s disease treatment (0.928)	Anti-inflammatory (0.955)
**273**	Anti-hypercholesterolemic (0.958)	Antineoplastic (0.871)
	Anti-inflammatory (0.913)	Alzheimer’s disease treatment (0.838)
	Atherosclerosis treatment (0.890)	Antioxidant (0.807)
	Respiratory analeptic (0.885)	Neurodegenerative diseases treatment (0.801)
**274**	Antiparkinsonian (0.955)	Atherosclerosis treatment (0.977)
	Neurodegenerative diseases treatment (0.954)	Anti-hypercholesterolemic (0.956)
	Alzheimer’s disease treatment (0.940)	Antioxidant (0.963)
	Antineoplastic (0.932)	Anti-hyperlipoproteinemic (0.811)
**275**	Anti-inflammatory (0.926)	Neurodegenerative diseases treatment (0.877)
	Antioxidant (0.922)	Anti-hypercholesterolemic (0.869)
	Atherosclerosis treatment (0.908)	Alzheimer’s disease treatment (0.868)
	Antineoplastic (0.895)	Antiparkinsonian (0.848)
**276**	Respiratory analeptic (0.959)	Atherosclerosis treatment (0.876)
	Anesthetic general (0.937)	Alzheimer’s disease treatment (0.828)
	Anti-hypercholesterolemic (0.909)	Neurodegenerative diseases treatment (0.808)
**277**	Anti-inflammatory (0.911)	Antioxidant (0.878)
	Anti-hypercholesterolemic (0.886)	Anesthetic general (0.863)
	Atherosclerosis treatment (0.883)	Antineoplastic (0.862)
**278**	Respiratory analeptic (0.933)	Atherosclerosis treatment (0.838)
	Anesthetic general (0.921)	Alzheimer’s disease treatment (0.821)
	Anti-hypercholesterolemic (0.902)	Neurodegenerative diseases treatment (0.796)
**279**	Anti-inflammatory (0.918)	Antioxidant (0.894)
	Anti-hypercholesterolemic (0.892)	Anesthetic general (0.859)
	Atherosclerosis treatment (0.843)	Antineoplastic (0.834)
**280**	Anti-inflammatory (0.928)	Neurodegenerative diseases treatment (0.888)
	Antioxidant (0.926)	Antineoplastic (0.884)
	Atherosclerosis treatment (0.910)	Alzheimer’s disease treatment (0.877)
	Anesthetic general (0.894)	Antiparkinsonian (0.872)

* Only activities with Pa > 0.5 are shown.

**Table 19 biomedicines-11-02698-t019:** Biological activities of steroids bearing tin atom (**281**–**298**) [358].

No.	Dominated Biological Activity (Pa) *	Additional Predicted Activities (Pa) *
**281**	Antineoplastic (0.905)	Anti-inflammatory (0.640)
	Antineoplastic (breast cancer) (0.866)	Dementia treatment (0.519)
	Prostatic (benign) hyperplasia treatment (0.832)	
	Prostate disorders treatment (0.766)	
**282**	Antineoplastic (0.996)	Ovulation inhibitor (0.721)
	Antineoplastic (breast cancer) (0.825)	
	Prostate disorders treatment (0.647)	
**283**	Antineoplastic (0.946)	Postmenopausal disorders treatment (0.677)
	Prostatic (benign) hyperplasia treatment (0.843)	Gynecological disorders treatment (0.553)
	Antineoplastic (breast cancer) (0.780)	Contraceptive (0.509)
	Prostate disorders treatment (0.748)	Erythropoiesis stimulant (0.505)
**284**	Antineoplastic (0.996)	Ovulation inhibitor (0.689)
	Antineoplastic (breast cancer) (0.777)	Genital warts treatment (0.538)
	Prostate disorders treatment (0.582)	Menopausal disorders treatment (0.532)
**285**	Antineoplastic (0.887)	Choleretic (0.895)
	Antimetastatic (0.776)	Biliary tract disorders treatment (0.825)
	Antibacterial (0.677)	Hepatic disorders treatment (0.716)
**286**	Antineoplastic (0.890)	Choleretic (0.894)
	Antimetastatic (0.781)	Hepatic disorders treatment (0.734)
**287**	Antineoplastic (0.823)	Choleretic (0.895)
	Antimetastatic (0.775)	Hepatic disorders treatment (0.707)
**288**	Antineoplastic (0.848)	Anti-hypercholesterolemic (0. 895)
	Antimetastatic (0.710)	Cholesterol synthesis inhibitor (0.645)
	Antibacterial (0.685)	Antifungal (0.649)
**289**	Antineoplastic (0.921)	Anti-hypercholesterolemic (0.907)
	Prostatic (benign) hyperplasia treatment (0.753)	Respiratory analeptic (0.847)
	Antimetastatic (0.748)	Anti-eczematic (0.822)
	Antineoplastic (pancreatic cancer) (0.702)	Anesthetic general (0.804)
	Prostate cancer treatment (0.700)	Dermatologic (0.747)
	Antibacterial (0.573)	Cholesterol synthesis inhibitor (0.627)
**290**	Antineoplastic (0.909)	Anti-hypercholesterolemic (0.906)
	Prostatic (benign) hyperplasia treatment (0.833)	Anti-eczematic (0.830)
	Antimetastatic (0.740)	Dermatologic (0.736)
**291**	Antineoplastic (0.904)	Ovulation inhibitor (0.901)
	Antineoplastic (breast cancer) (0.881)	Male reproductive dysfunction treatment (0.716)
	Prostatic (benign) hyperplasia treatment (0.780)	Menopausal disorders treatment (0.663)
**292**	Antineoplastic (0.906)	Ovulation inhibitor (0.840)
	Antineoplastic (breast cancer) (0.837)	Dermatologic (0.674)
	Prostatic (benign) hyperplasia treatment (0.827)	Menopausal disorders treatment (0.666)
	Antineoplastic (sarcoma) (0.824)	Male reproductive dysfunction treatment (0.650)
	Antineoplastic (renal cancer) (0.762)	Respiratory analeptic (0.651)
	Prostate cancer treatment (0.750)	Bone diseases treatment (0.641)
	Antineoplastic (pancreatic cancer) (0.741)	
**293**	Antineoplastic (0.841)	Anesthetic general (0.850)
	Prostatic (benign) hyperplasia treatment (0.841)	Immunosuppressant (0.725)
**294**	Anti-inflammatory (0.917)	Antineoplastic (0.828)
	Anesthetic general (0.872)	Muscular dystrophy treatment (0.784)
**295**	Anti-inflammatory (0.920)	Allergic conjunctivitis treatment (0.765)
	Antineoplastic (0.761)	Antiallergic (0.760)
	Prostate disorders treatment (0.701)	Antipruritic, allergic (0.710)
**296**	Anti-eczematic (0.807)	Prostate disorders treatment (0.683)
	Anti-osteoporotic (0.612)	Cytoprotectant (0.683)
**297**	Antineoplastic (0.874)	Anti-eczematic (0.707)
	Prostatic (benign) hyperplasia treatment (0.734)	Anti-inflammatory (0.622)
	Antineoplastic (breast cancer) (0.712)	Anti-secretoric (0.580)
	Prostate disorders treatment (0.718)	Antibacterial (0.544)
**298**	Antineoplastic (0.704)	Anti-eczematic (0.630)
	Antibacterial (0.566)	Dermatologic (0.532)

* Only activities with Pa > 0.5 are shown.

**Table 20 biomedicines-11-02698-t020:** Biological activities of ferrocene steroid conjugates (**299**–**332**) [370].

No.	Dominated Biological Activity (Pa) *	Additional Predicted Activities (Pa) *
**299**	Erythropoiesis stimulant (0.847)	Ovulation inhibitor (0.656)
**300**	Erythropoiesis stimulant (0.909)	Antineoplastic (breast cancer) (0.909)
	Prostate disorders treatment (0.626)	Antineoplastic (0.880)
**301**	Erythropoiesis stimulant (0.831)	Ovulation inhibitor (0.714)
**302**	Erythropoiesis stimulant (0.794)	Ovulation inhibitor (0.596)
**303**	Ovulation inhibitor (0.748)	Erythropoiesis stimulant (0.703)
**304**	Ovulation inhibitor (0.748)	Erythropoiesis stimulant (0.703)
**305**	Prostate disorders treatment (0.755)	Erythropoiesis stimulant (0.588)
**306**	Anti-inflammatory (0.876)	Erythropoiesis stimulant (0.725) Growth stimulant (0.581)
	Allergic conjunctivitis treatment (0.765)	
	Antiallergic (0.690)	
**307**	Erythropoiesis stimulant (0.731)	Ovulation inhibitor (0.648)
**308**	Erythropoiesis stimulant (0.826)	Ovulation inhibitor (0.712)
**309**	Anti-inflammatory (0.788)	Erythropoiesis stimulant (0.763)
	Allergic conjunctivitis treatment (0.680)	Ovulation inhibitor (0.639)
**310**	Erythropoiesis stimulant (0.795)	Muscular dystrophy treatment (0.719)
	Ovulation inhibitor (0.770)	Menopausal disorders treatment (0.629)
**311**	Antineoplastic (breast cancer) (0.969)	Ovulation inhibitor (0.867)
	Antineoplastic (0.930)	Contraceptive (0.635)
**312**	Ovulation inhibitor (0.880)	Prostate disorders treatment (0.697)
	Erythropoiesis stimulant (0.714)	Menopausal disorders treatment (0.668)
**313**	Erythropoiesis stimulant (0.809)	Ovulation inhibitor (0.680)
**314**	Ovulation inhibitor (0.909)	Prostate disorders treatment (0.653)
	Menopausal disorders treatment (0.651)	Erythropoiesis stimulant (0.625)
**315**	Erythropoiesis stimulant (0.900)	Prostate disorders treatment (0.662)
	Ovulation inhibitor (0.702)	Menopausal disorders treatment (0.648)
**316**	Erythropoiesis stimulant (0.911)	Dementia treatment (0.647)
	Cardiotonic (0.845)	Respiratory analeptic (0.661)
**317**	Antineoplastic (0.832)	Erythropoiesis stimulant (0.758) Choleretic (0.677)
	Antineoplastic (breast cancer) (0.819)	
	Prostatic (benign) hyperplasia treatment (0.518)	
**318**	Antineoplastic (0.837)	Anesthetic general (0.831) Respiratory analeptic (0.785)
	Cytoprotectant (0.684)	
	Prostatic (benign) hyperplasia treatment (0.585)	
**319**	Antineoplastic (0.944)	Anti-psoriatic (0.910)
	Apoptosis agonist (0.879)	Anti-osteoporotic (0.899)
	Proliferative disease treatment (0.637)	Antiparkinsonian, rigidity relieving (0.568)
**320**	Antineoplastic (breast cancer) (0.819)	Genital warts treatment (0.656)
	Antineoplastic (0.774)	Erythropoiesis stimulant (0.594)
**321**	Antineoplastic (breast cancer) (0.818)	Erythropoiesis stimulant (0.638)
	Antineoplastic (0.782)	Steroid synthesis inhibitor (0.533)
**322**	Anti-hypercholesterolemic (0.920)	Respiratory analeptic (0.854)
	Antineoplastic (0.837);	Anesthetic general (0.694)
	Proliferative disease treatment (0.638)	Cholesterol synthesis inhibitor (0.636)
**323**	Anti-hypercholesterolemic (0.885)	Respiratory analeptic (0.860)
	Erythropoiesis stimulant (0.819)	Anesthetic general (0.846)
**324**	Anti-inflammatory (0.718)	HCV IRES inhibitor (0.669)
**325**	Ovulation inhibitor (0.728)	HCV IRES inhibitor (0.712)
	Menopausal disorders treatment (0.625)	Anti-osteoporotic (0.619)
**326**	Ovulation inhibitor (0.775)	HCV IRES inhibitor (0.766)
	Menopausal disorders treatment (0.715)	Anti-osteoporotic (0.708)
**327**	Glutaconyl-CoA decarboxylase inhibitor (0.637)	Glyceryl-ether monooxygenase inhibitor (0.545)
**328**	Gestagen antagonist (0.598)	Ovulation inhibitor (0.526)
**329**	Glutaconyl-CoA decarboxylase inhibitor (0.655)	Estrogen alpha receptor agonist (0.626)
**330**	Erythropoiesis stimulant (0.942)	Ovulation inhibitor (0.862)
**331**	Erythropoiesis stimulant (0.942)	Ovulation inhibitor (0.862)
**332**	Erythropoiesis stimulant (0.938)	Ovulation inhibitor (0.859)

* Only activities with Pa > 0.5 are shown.

**Table 21 biomedicines-11-02698-t021:** Biological activities of titanocene steroid conjugates (**332**–**339**) [370].

No.	Dominated Biological Activity (Pa) *	Additional Predicted Activities (Pa) *
**332**	Antineoplastic (0.996)	Dermatologic (0.638)
	Antineoplastic, alkylator (0.878)	Anti-eczematic (0.553)
	Prostatic (benign) hyperplasia treatment (0.740)	Erythropoiesis stimulant (0.527)
**333**	Antineoplastic (0.996)	Ovulation inhibitor (0.668)
	Antineoplastic, alkylator (0.823)	Dermatologic (0.583)
	Antineoplastic (breast cancer) (0.740)	Menopausal disorders treatment (0.525)
**334**	Antineoplastic (0.996)	Dermatologic (0.531)
	Alkylator (0.895)	Erythropoiesis stimulant (0.527)
	Antineoplastic, alkylator (0.775)	Male reproductive dysfunction treatment (0.500)
**335**	Antineoplastic (0.996)	Dermatologic (0.531)
	Alkylator (0.895)	Erythropoiesis stimulant (0.527)
	Antineoplastic, alkylator (0.775)	Male reproductive dysfunction treatment (0.500)
**336**	Antineoplastic (0.995)	Anti-hypercholesterolemic (0.881)
	Toxic (0.948)	Hypolipemic (0.643)
	Antineoplastic, alkylator (0.583)	Cholesterol synthesis inhibitor (0.622)
	Apoptosis agonist (0.529)	Immunosuppressant (0.593)
**337**	Antineoplastic (0.995)	Anti-hypercholesterolemic (0.881)
	Toxic (0.948)	Hypolipemic (0.643)
	Antineoplastic, alkylator (0.583)	Cholesterol synthesis inhibitor (0.622)
	Apoptosis agonist (0.529)	Immunosuppressant (0.593)
**338**	Antineoplastic (0.996)	Anti-hypercholesterolemic (0.802) Cholesterol synthesis inhibitor (0.594) Hypolipemic (0.558)
	Toxic (0.952)	
	Antineoplastic, alkylator (0.681)	
	Proliferative disease treatment (0.542)	
	Prostatic (benign) hyperplasia treatment (0.537)	
	Apoptosis agonist (0.529)	
**339**	Antineoplastic (0.996)	Anti-hypercholesterolemic (0.643)
	Proliferative disease treatment (0.579)	Cholesterol synthesis inhibitor (0.550)

* Only activities with Pa > 0.5 are shown.

## Data Availability

Not applicable.

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
