# Peer review of "Steroids Bearing Heteroatom as Potential Drugs for Medicine"

_biomedicines, 2023, doi:10.3390/biomedicines11102698_

Round 1

Reviewer 1 Report

The author reports an overview of hetero-steroids as potential therapeutic agents. The reported issues have relevance in drug discovery. I believe that this manuscript can be considered for publication in this Journal after some revisions.

-          The Abstract section should be revised; it should report only a brief and comprehensive summary of the reported contents. General issues about the covered issues should be discussed in the Introduction section.

-          The Introduction section must be deeply improved by better discussing the background in steroids bearing heteroatoms and their potential in medicine field. The novelty of the work with respect to that reported by other authors in similar review must be also discussed in this section. Moreover, a brief discussion related to the used classification (steroids containing nitrile, nitro, halogen atoms, etc.) must be inserted.  

-          The references (literature data) related to the biological activities of steroids, must be inserted in all the Tables.

-          The Conclusions section should highlight the most important advances in the field also discussing the classes of heterosteroids with the greatest potential in the medical field.

Minor editing of English language is required.

Author Response

Reviewer 1

I thank you very much for your feedback on my review, which presents steroids with a heteroatom.

According to your recommendations, abstract has been completely rewritten. Introduction and conclusions have also been rewritten.

Finally, a discussion of heterosteroids with the greatest potential in the medical field is added.

Added some literary sources from where the activities of steroids were taken.

To improve my English, I asked a friend from the English Philology department to remove some inaccuracies.

This review appears to be rare in the field of heteroatomic steroids, and, it seems to me, it turned out to be interesting and useful for readers. I would like to hope that my additions and corrections turned out to be correct.

In addition, all corrections and additions to the text are highlighted in blue.

Reviewer 2 Report

Compounds studied in this ample review have shown promise as potential therapeutic agents in the treatment of various diseases, such as cancer, infectious diseases, cardiovascular disorders, and neurodegenerative conditions. Moreover, the incorporation of heteroatoms has led to the development of targeted drug delivery systems, prodrugs, and other innovative pharmaceutical approaches. 

It is important to note that the specific biological activities and properties of heteroatom steroids can vary depending on the structure of the steroid, the presence, and position of heteroatoms, and other factors.

It is a very extensive review that brings together very useful information in relation to structure-activity. The bibliographic references are updated and therefore, they are insurance when it comes to relying on the latest research.

The work has required a notable effort on the part of the author, and consequently, I give my approval for its publication in its current state.

Author Response

Reviewer 2

I thank you very much for your feedback on my review, which presents steroids with a heteroatom.

I would like to note that the specific biological activity and properties of heteroatomic steroids may vary depending on the structure of the steroid, the presence and position of heteroatoms and other factors.

This review appears to be rare in the field of heteroatomic steroids, and, it seems to me, it turned out to be interesting and useful for readers.

With kind regards

Valery M Dembitsky

Round 2

Reviewer 1 Report

I believe that this revised version of the manuscript can be accepted for publication in this Journal.